# CircRREB1 mediates lipid metabolism related senescent phenotypes in chondrocytes through FASN post-translational modifications

Zhe Gong[1,3], Jinjin Zhu[1,3], Junxin Chen[1,3], Fan Feng[2,3], Haitao Zhang[1], Zheyuan Zhang [1], Chenxin Song[1], Kaiyu Liang [1], Shuhui Yang[1], Shunwu Fan [1] ✉, Xiangqian Fang [1] ✉ & Shuying Shen [1] ✉

Osteoarthritis is a prevalent age-related disease characterized by dysregulation of extracellular matrix metabolism, lipid metabolism, and upregulation of senescence-associated secretory phenotypes. Herein, we clarify that CircRREB1 is highly expressed in secondary generation chondrocytes and its deficiency can alleviate FASN related senescent phenotypes and osteoarthritis progression. CircRREB1 impedes proteasome-mediated degradation of FASN by inhibiting acetylation-mediated ubiquitination. Meanwhile, CircRREB1 induces RanBP2-mediated SUMOylation of FASN and enhances its protein stability. CircRREB1-FASN axis inhibits FGF18 and FGFR3 mediated PI3K-AKT signal transduction, then increased p21 expression. Intra-articular injection of adenovirus−CircRreb1 reverses the protective effects in CircRreb1 deficiency mice. Further therapeutic interventions could have beneficial effects in identifying CircRREB1 as a potential prognostic and therapeutic target for age-related OA.

Osteoarthritis (OA) is the most prevalent degenerative disease worldwide and is more prominent in older individuals[1]. OA manifests as joint pain, affecting the knee, hand, hips, and spine, and occurs due to the wearing down of the protective cartilage around the joints. Even though numerous treatment measures to relieve OA symptoms have been developed, including anti-inflammatory therapy[2] and other non-surgical treatment[3], a comprehensive and effective treatment that prevents OA progression does not exist. Therefore, many patients must receive a total knee replacement and ultimately undergo total knee arthroplasty (TKA). Cartilage is one of the essential components of knee joints, and chondrocytes maintain cartilage integrity by extracellular matrix (ECM) homeostasis[4]. Disruption of ECM homeostasis

during OA pathogenesis leads to excessive upregulation of matrix-degrading enzymes, including matrix metalloproteinase 13 (MMP13) and A disintegrin and metalloproteinase with thrombospondin motifs (ADAMTS5) and downregulation of matrix components, aggrecan and type II collagen (Col2)[5]. Cartilage destruction, therefore, gets activated, resulting in OA phenotype presentation, such as synovial inflammation and subchondral osteosclerosis[6]. Aging, obesity, sex, and chronic inflammation are the major risk factors for OA pathogenesis[7,8]. An in-depth understanding of the molecular mechanisms underlying age and OA is essential to identify therapeutic targets for treating OA.

Even though OA is considered highly related to ECM metabolic disorders, other metabolic alterations are also involved in OA

[1]Department of Orthopaedic Surgery, Sir Run Run Shaw Hospital, Medical College of Zhejiang University & Key Laboratory of Musculoskeletal System Degeneration and Regeneration Translational Research of Zhejiang Province Sir Run Run Shaw Institute of Clinical Medicine of Zhejiang University, 3 East Qingchun Road, Hangzhou 310016 Zhejiang Province, China. [2]Obstetrics and Gynecology Hospital, Kunpeng Road, Hangzhou 310016 Zhejiang Province, China. [3]These authors contributed equally: Zhe Gong, Jinjin Zhu, Junxin Chen, Fan Feng. ✉e-mail: 0099203@zju.edu.cn; orthofxq@zju.edu.cn; 11207057@zju.edu.cn

pathogenesis. Under inflammatory microenvironment and stress stimuli, chondrocytes undergo a metabolic shift from oxidative phosphorylation to anaerobic glycolysis, aggravating OA severity[9]. Mitochondrial dysfunction and reactive oxygen species (ROS) production also occur in chondrocytes under inflammatory microenvironments, resulting in an imbalance between ROS production and scavenging, contributing to OA development[9]. Hence, determining the type of metabolic pathway in aging chondrocytes is essential for developing effective treatment for age-related OA. The relationship between lipid metabolism and OA has been the primary research focus till recently. Chun et al.[10] identified an increased cholesterol level in OA chondrocytes, regulated by the CH25H−CYP7B1−RORα axis and demonstrated that the CH25H−CYP7B1−RORα axis of cholesterol metabolism triggers a catabolic environment during OA pathogenesis. A lipid scarcity is reported to induce chondrogenesis in skeletal progenitor cells[11], while lipid accumulation in chondrocytes triggers OA development, and dysregulation of lipid metabolism accelerates OA progression[12]. Increasing clinical evidence has also shown that dysregulation of lipid metabolism is highly associated with OA development[13]. Individuals with Older age and obesity are considered the most prominent risk factors for OA[14], with an increased susceptibility for obese populations[15,16]. In addition, increased levels of lipogenesis have been observed in OA mice models[17] and patients with OA[18]. De novo lipogenesis (DNL) is a fundamental biosynthetic process that plays an essential role in lipid accumulation during the development of many diseases, including OA[19,20]. ATP citrate lyase (ACLY), acetyl-CoA carboxylase (ACC), and fatty acid synthase (FASN) are the primary lipogenic regulators of the DNL process[21] and are considered promising therapeutic targets against various diseases[20]. FASN activation is a major metabolic event in tumor cells[22–24]. Although FASN is reported as a therapeutic target in many diseases, the relationship between FASN and age-related OA is still not well studied. Herein, investigating molecular mechanisms underlying the FASN and age-related OA is considered to find out therapeutic targets or age-related OA. This study investigated the molecular mechanisms underlying FASN and age-related OA to identify beneficial therapeutic targets.

Circular RNAs (CircRNAs) are non-coding RNAs characterized by a closed ring formed by the back splicing of the free 3' or 5' ends[25]. Our previous work, and those of others, have demonstrated circRNAs as sponges of micro RNAs (miRNAs) that prevent or promote OA development[26,27]. Recently, our study showed that circRNAs acted as a scaffold for RNA-binding proteins to regulate OA pathogenesis[28]. However, the mechanism by which circRNAs regulate lipid metabolism in age-related OA progression remains poorly understood.

Here, we show that circRNA (hsa_circ_0001573, termed CircRREB1), originating from the RREB1 gene transcript, is a highly expressed P2 generation chondrocyte. We demonstrate and clarify that CircRREB1 participates in chondrocyte senescence and age related OA by regulating FASN associated lipid metabolism. CircRREB1 promotes FASN stability and regulates lipid metabolism-related senescence-associated secretory phenotypes (SASPs) through two mechanisms. CircRREB1 inhibits FASN acetylation-mediated ubiquitination and decreases FASN degradation. Meanwhile, CircRREB1 promotes RanBP2-mediated SUMOylation of FASN, thereby enhancing FASN stability. Hence, we demonstrate the roles of CircRREB1-FASN axis in lipogenesis in P2 generation chondrocytes and believe that targeting CircRREB1 is a promising potential treatment for age-related OA.

## Result

### Senescent phenotype in P0 and P2 generation chondrocytes

In the knee joint, low oxygen tension promotes chondrocyte growth and function under normal physiological conditions. Primary chondrocytes cultured under normoxia conditions showed higher oxidative stress than chondrocytes exposed to hypoxia conditions, as indicated by the upregulation of 4-hydroxynonenal (4-HNE), a lipid peroxidation product (Supplementary Fig. 1a). We examined the senescent phenotype of chondrocytes at passage 0 (P0) and subjected them to two passages (P2) under normoxia or hypoxia conditions. The P2 chondrocytes showed higher p16 expression than P0 chondrocytes under normoxia conditions (Supplementary Fig. 1b). However, P2 chondrocytes under hypoxia showed downregulation of p16 expression (Supplementary Fig. 1b). In addition, P2 chondrocytes expressed more senescence-associated β-galactosidase (SA-β-Gal) (Fig. 1a). Quantification of SA-β-Gal positive staining showed the same result (Fig. 1b).

### CircRREB1 is highly expressed in P2 generation chondrocytes

To generate a circRNA profiling database, we performed RNA-seq analyses of ribosomal RNA-depleted total RNA from three P0 generation chondrocytes and three P2 chondrocytes. A total of 3800 circRNAs were identified using CIRI2 and CIRC explorer. A total of 340 circRNAs differentially expressed in P2 and P0 generation chondrocyte was identified by RNA-seq mapping to the reference genome (hg38, human genome), with | log2FC(P2/P0)|>1 and FDR ≤ 0.05, among which 274 circRNAs were significantly upregulated and 66 circRNAs were significantly downregulated (Fig. 1c). A heat map showed the top 50 upregulated and downregulated circRNAs (Fig. 1d). The top 15 upregulated and downregulated circRNAs are listed in Supplementary Table 1. We selected the top 10 upregulated CircRNAs and the top 10 downregulated CircRNAs. We performed real-time quantitative polymerase chain reaction (RT-qPCR) to confirm the top 10 upregulated and downregulated circRNAs. RT-qPCR analysis indicated that CircRREB1 was upregulated circRNA in P2 senescent chondrocytes (Fig. 1f). We further selected the top 5 upregulated CircRNAs and top5 downregulated CircRNAs to perform RNA knockdown to detect the expression of senescent marker p21 and SASPs factor MMP13 via RT-qPCR analysis. The result showed that CircRREB1 knockdown significantly decreased the expression of p21 and MMP13 (Supplementary Fig. 1d and e). Hence, we selected CircRREB1 as a potential target.

### CircRREB1 expression and characterizations in chondrocyte senescent progression

To examine CircRREB1 expression levels in human chondrocytes, chondrocytes at P0 and P2 were collected. RNA fluorescence in situ hybridization (FISH) staining showed that the P2 generation had higher CircRREB1 expression than P0 chondrocytes (Supplementary Fig. 1f). Next, we collected human cartilage samples from 12 younger and older adults to detect the effect of age on CircRREB1 expression. FISH staining showed increased CircRREB1 expression in older cartilage tissues (Supplementary Fig. 1h). Quantifying CircRREB1 expression in chondrocytes and cartilage tissues showed similar results (Supplementary Fig. 1g and i). To further confirm the relationship between CircRREB1 expression and age in human cartilage samples, we performed CircRREB1 FISH staining in sections with age (n = 24). The result showed a positive correlation between CircRREB1 expression and age (Supplementary Fig. 1c). A previous study has reported doxorubicin (Doxo) to induce chondrocyte senescence[29]. Herein, chondrocytes were treated with Doxo (100 nM), and more CircRREB1 expressing chondrocytes were observed in the Doxo-treated group, as indicated by FISH staining (Supplementary Fig. 1j and k). We also verified CircRreb1 expression in aging mice and Terc+/− mice and identified articular cartilage of 18-month mice to have higher CircRreb1 expression compared to that in the three-month-old mice (Fig. 1e). CircRreb1 was also found upregulated in the articular cartilage of Terc +/− mice (Fig. 1e).

Sanger sequencing was performed to confirm the head-to-tail formation of CircRREB1 (Fig. 1i). To verify that exons 2 to 6 of *RREB1* form endogenous circRNAs, we designed convergent primers to

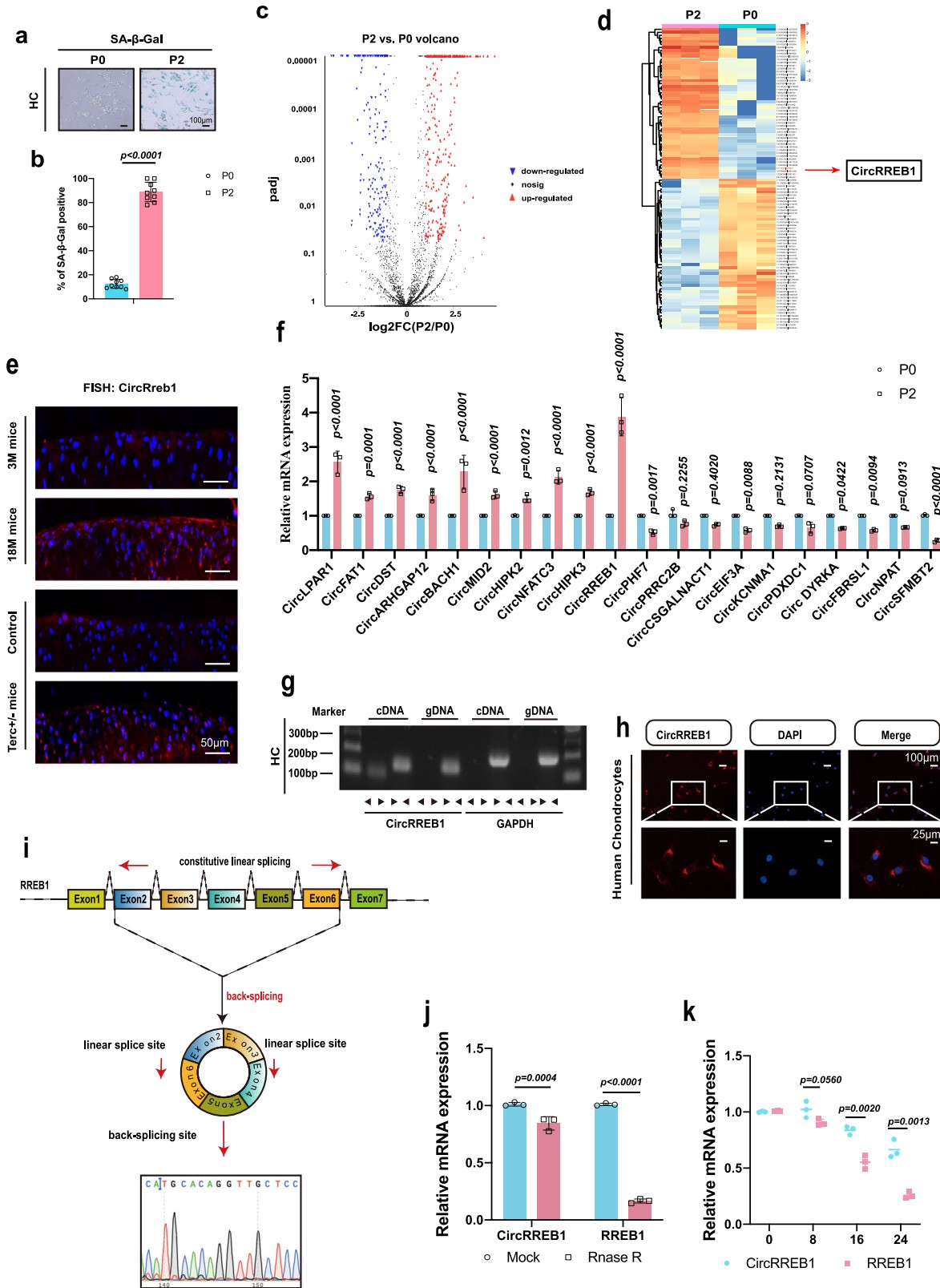

amplify the *RREB1* host gene and divergent primers to amplify *CircRREB1*. Genomic DNA (gDNA) and cDNA from chondrocytes were used as templates, and *CircRREB1* could only be amplified using cDNA instead of gDNA, using divergent primers (Fig. 1g). FISH staining revealed abundant expression of CircRREB1 in the cytoplasm of human chondrocytes (Fig. 1h). Furthermore, we selected two steps to detect

the constancy of CircRREB1. RT-qPCR confirmed that *CircRREB1* was resistant to RNase R digestion, whereas *RREB1* mRNA decreased significantly after RNase R treatment (Fig. 1j). Chondrocytes were treated with actinomycin D, a transcription inhibitor, for 8, 16, and 24 h. RT-qPCR results indicated that the half-life period of *CircRREB1* was longer than that of *RREB1* mRNA (Fig. 1k).

**Fig. 1 | CircRREB1 expression and characterization in P0 and P2 generation chondrocytes. a** SA-β-Gal expression in P0 and P2 generation chondrocytes. **b** Quantification of SA-β-Gal positive chondrocytes (*n* = 9, biologically independent samples). **c** CircRNA identification in P0 and P2 generation chondrocytes. Red labels are upregulated CircRNAs and blue labels are downregulated CircRNAs. **d** A heat map showing differentially expressed CircRNAs in P0 and P2 generation chondrocytes. **e** RNA fluorescence in situ hybridization (FISH) staining of CircRreb1 in aging mice and Terc +/− mice (CircRreb1 labeled with Cy3). **f** The top 10 upregulated and downregulated CircRNAs in P0 and P2 generation chondrocytes (*n* = 3, biologically independent samples). **g** Divergent primers amplified CircRREB1 from cDNA, but not from genomic DNA. GAPDH was used as a negative control. **h** RNA FISH showing the predominant localization of CircRREB1 in the cytoplasm.

**i** Schematic illustration showing *RREB1* exons 2–6 circularization to form CircRREB1. CircRREB1 presence was validated by real-time quantitative polymerase chain reaction (RT-qPCR), followed by Sanger sequencing. **j** *CircRREB1* and *RREB1* mRNA expression in chondrocytes treated with or without Rnase R detected by RT-qPCR (*n* = 3, biologically independent samples). **k** The half-life period of *CircRREB1* and *RREB1* in chondrocytes treated with Actinomycin D for 8, 16, and 24 h, analyzed via RT-qPCR (*n* = 3, biologically independent samples). Two-sided Student's *t* test is used for statistical analysis (**b**, **j**, and **k**). Quantitative data are shown as mean ± s.d. Exact *p* values are shown in the figures. Scar bar for (**a**): 100 μm. Scar bar for (**e**): 50 μm. Scar bar for (**h**): 100 μm, amplification: 25 μm. Source data are provided as a Source Data file.

## CircRREB1 is regulated by RNA-binding protein DExH-Box Helicase 9 (DHX9)

We next explore the upstream regulation of CircRREB1. First, the host gene *RREB1* showed no obvious difference in P0 generation and P2 generation chondrocyte, as well as Doxo-induced senescence model (Supplementary Fig. 2a, b). Hence, we suggested that CircRREB1 was regulated by RNA-binding protein. RNA-binding protein DExH-Box Helicase 9 (DHX9), which was reported governing the CircRNAs biogenesis broadly[30]. After DHX9 knockdown in chondrocyte, *CircRREB1* expression was upregulated, while host gene *RREB1* did not show significant changes (Supplementary Fig. 2c). Notably, we also found that *DHX9* mRNA was downregulated in P2 generation chondrocytes (Supplementary Fig. 2d). DHX9 knockdown in chondrocytes also increased *p16* and *p21* mRNA expression indicated by RT-qPCR (Supplementary Fig. 2e). Together, the downregulation of DHX9 in P2 generation chondrocytes is at least partially responsible for the overexpression of CircRREB1.

## CircRREB1 regulates ECM metabolism and senescent phenotypes in chondrocytes

Transfecting CircRREB1 siRNAs into human chondrocytes (HCs) significantly decreased CircRREB1 expression in HCs, whereas CircRREB1 knockdown did not affect the *RREB1* host gene level (Fig. 2b). We examined the effects of CircRREB1 knockdown on ECM metabolism and senescence phenotypes in HCs. CircRREB1 inhibition in HCs significantly decreased the expression of ECM-degrading genes *MMP3, MMP13*, and *ADAMTS5*, and senescence-associated genes *p16, p21*, and *p53*, and increased SRY-Box Transcription Factor 9 (*Sox9*), *Aggrecan*, and *Col2* expression, as indicated by RT-qPCR (Fig. 2c and d). Consistent with the RT-qPCR results, CircRREB1 knockdown decreased MMP3, ADAMTS5, p16, p21, and p53 protein levels and enhanced Col2 and Aggrecan protein expression levels (Fig. 2a). The SA-β-Gal staining kit was used to evaluate senescent phenotypes[29]. We also have noticed that positive staining of β-Gal staining is caused by β-galactosidase activity in lysosomes. Then, we suggested whether CircRREB1 regulates senescence phenotypes by affecting the autophagy pathway and lysosome activity. Autophagy associated LC3B and p62 protein expression did not changed after CircRREB1 knockdown in chondrocyte (Supplementary Fig. 3c), as well as quantifications of LC3B II and p62 expression (Supplementary Fig. 3d, e). To further detect the effect of CircRREB1 on lysosomes activity, chondrocytes transfected with NC and CircRREB1 SiRNAs were treated with or without chloroquine (CQ). Inhibition of LC3B degradation in autolysosomes with CQ showed that autophagy flux was similar in Si-NC group and Si-CircRREB1 group (Supplementary Fig. 3f, g), which suggested that CircRREB1 did not affect lysosomes activity in chondrocytes. SA-β-Gal positive staining decreased in CircRREB1 knockdown HCs after Doxo stimulation (Fig. 2f), however, lysosome acid β-Galactosidase showed no obvious difference after CircRREB1 knockdown in chondrocytes, which could be a negative control for SA-β-Gal staining (Supplementary Fig. 3b). Moreover, representative immunofluorescence images showed that CircRREB1 decreased MMP13 and p16 expression and

enhanced Sox9 expression (Fig. 2f), as well as p21 and gama-H2AX nuclear foci expression (Supplementary Fig. 3a). CircRREB1 inhibition also increased Alcian blue positive and toluidine blue positive staining, which was more prominent in Doxo induced senescent chondrocytes (Fig. 2e).

Subsequently, a gain-of-function test was performed. 100 multiplicity of infection (MOI), 200 MOI, 400 MOI of Ad-CircRREB1, and 400 MOI control viruses (Ad-Control) were used to infect primary HCs. RT-qPCR analysis indicated that 400 MOI Ad-CircRREB1 significantly upregulated *CircRREB1* expression in the HCs (Fig. 2g). CircRREB1 overexpression increased *MMP13, ADAMTS5, p16*, and *p21* expression and decreased *Col2, Aggrecan*, and *Sox9* expression, as indicated by RT-qPCR (Supplementary Fig. 4a). Consistent with the RT-qPCR analysis, CircRREB1 overexpression also upregulated ADAMTS5, ADAMTS4, MMP13, p16, p21, and p53 expression, while downregulating Col2 and Sox9 expression, as indicated by the western blot analysis (Fig. 2h). More SA-β-Gal-positive chondrocytes were observed following CircRREB1 overexpression (Supplementary Fig. 4b). In addition, representative immunofluorescence images showed that CircRREB1 overexpression decreased Aggrecan and Col2 expression and enhanced p16 expression (Supplementary Fig. 4c).

Next, we examined the effects of CircRreb1 knockdown on ECM metabolism and senescence phenotypes in mouse chondrocytes (MCs). Three effective CircRreb1 siRNAs were transfected into MCs, which decreased *CircRreb1* expression but did not affect Rreb1 host gene expression (Supplementary Fig. 4d). RT-qPCR analysis indicated that CircRreb1 knockdown downregulated senescence-associated genes and *ADAMTS5* expression while upregulating *Aggrecan* and *Sox9* expression (Supplementary Fig. 4e and f). Western blot analysis showed the same results (Supplementary Fig. 4g). Representative immunofluorescence images showed that CircRreb1 knockdown significantly reduced p16 and MMP13 staining (Supplementary Fig. 4i). Knockdown of CircRreb1 in MCs also decreased SA-β-Gal positive staining (Supplementary Fig. 4h). These data indicate that CircRREB1/CircRreb1 is prominent in regulating ECM metabolism and senescence phenotypes in human and mouse chondrocytes.

## Alleviation of OA and senescent phenotypes in CircRreb1-deficiency mice

Based on the effects of CircRREB1/CircRreb1 knockdown in HC/MCs, we investigated whether CircRreb1 loss affects ECM and senescent phenotypes in mice. We successfully generated CircRreb1 gKO mice (Fig. 2i). Significant CircRreb1 knockdown efficiency was observed in knee joint chondrocytes, as indicated by FISH staining (Fig. 2j and k). Knee joints of 18-month-old mice were stained using safranin O/fast green staining to detect cartilage (Fig. 2l). Fewer cartilage losses were observed in CircRreb1 gKO mice than in wide type (WT) mice, as demonstrated by a decrease in the Osteoarthritis Research Society International (OARSI) score (Fig. 2n) and larger cartilage thickness (Fig. 2o). At the molecular level, we observed that MMP13, p16, and p21 positive cells were decreased in CircRreb1 gKO mice compared to those in WT mice, and Aggrecan-positive cells were enhanced

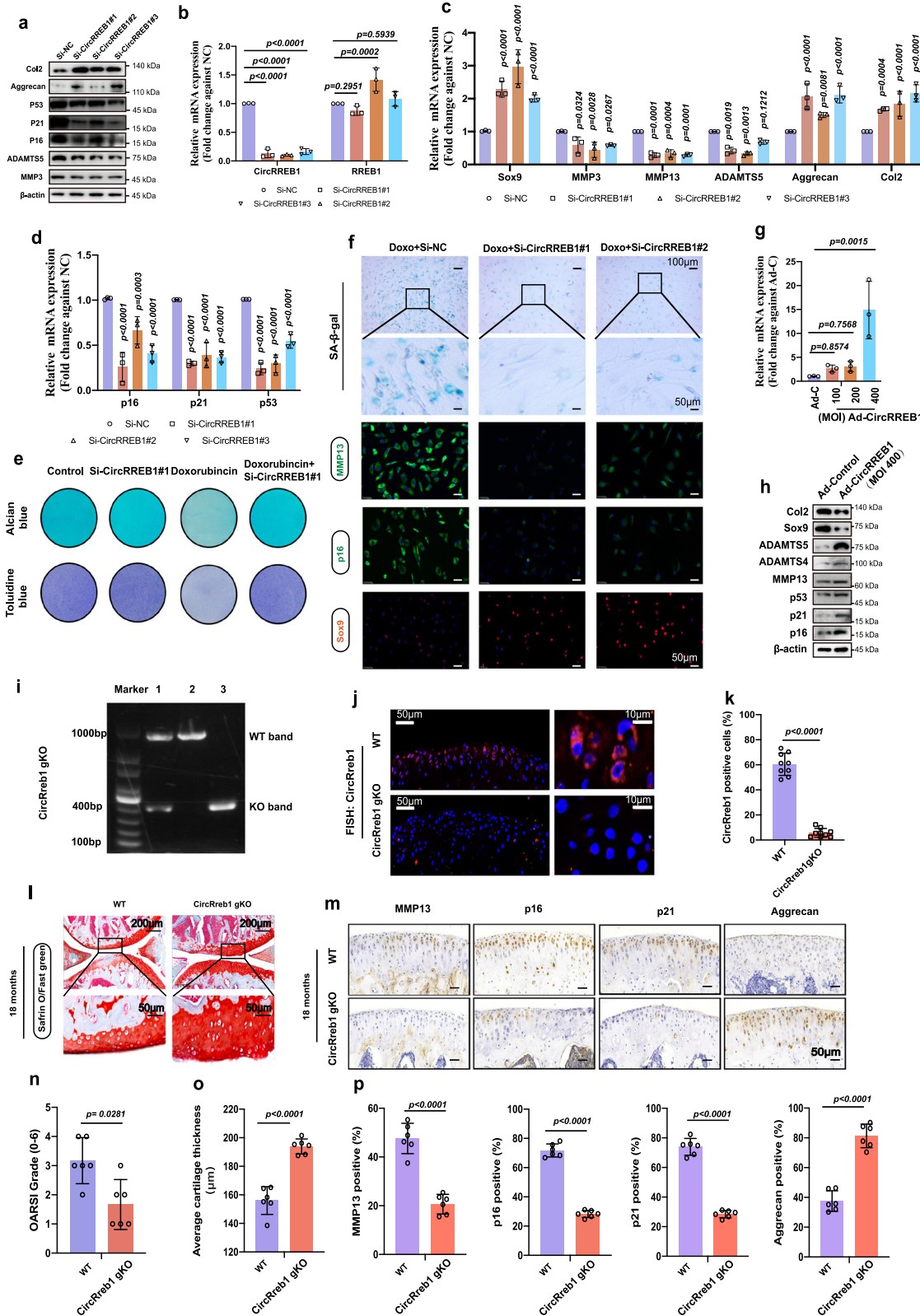

(Fig. 2m). Quantifying MMP13-, p16-, p21-, and Aggrecan-positive cells showed the same results (Fig. 2p).

## CircRREB1 aggravates senescence related OA phenotypes in aging mice

We compared 3-month-old mice and 18-month-old mice to investigate the role of CircRreb1 in promoting age-related OA progression.

Destabilization of the medial meniscus (DMM) was performed in 3-month and 18-month-old mice, followed by CircRreb1 specific adeno-associated virus (AAV) treatment 2 weeks post-DMM operation. All animals were euthanized eight weeks post-injury (Fig. 3a). CircRreb1 AAV infection in 3-month and 18-month-old mice could enhance Cir-cRreb1 overexpression in chondrocytes respectively indicated by Cir-cRreb1 FISH staining (Supplementary Fig. 5a), however, the fold

**Fig. 2 | The role of CircRREB1 in ECM and senescence phenotypes in vitro and in vivo. a** Col2, Aggrecan, p53, p21, p16, ADAMTS5, and MMP3 protein expression in P2 generation HCs infected with CircRREB1 SiRNAs or a negative control.
**b** Knockdown efficiencies of CircRREB1 in P2 generation HCs treated with three CircRREB1 SiRNAs ($n = 3$, biologically independent samples). CircRREB1 knockdown did not decrease RREB1 mRNA expression level. **c, d** *Sox9, MMP3, MMP13, ADAMTS5, Aggrecan, Col2, p16, p21,* and *p53* mRNA expression in P2 generation HCs treated with three CircRREB1 SiRNAs or a negative control ($n = 3$, biologically independent samples). **e** Representative Alcian blue and toluidine blue staining in HCs treated with or without doxo stimulation. **f** SA-β-Gal expression and representative immunofluorescence images of MMP13, p16, and Sox9 in HCs with CircRREB1 SiRNA#1, SiRNA#2, and negative control after doxo stimulation.
**g** Overexpression of CircRREB1 efficiencies in HCs infected with varying MOIs of the CircRREB1 virus (Ad-CircRREB1) or 400 MOI of the control virus (Ad-Control) [$n = 3$, biologically independent samples]. **h** Col2, Sox9, ADAMTS5, ADAMTS4, MMP13, p53, p21, and p16 expression in HCs infected with Ad-Control and Ad-CircRREB1 (400 MOI). Blots are representative of three independent experiments.
**i** Genotyping of CircRreb1-deficient mice. Blots are representative of three independent experiments. **j** RNA FISH in 8-week-old wide type male mice and CircRreb1

gKO male mice (CircRreb1 probe labeled with Cy3) [$n = 9$, biologically independent samples]. **k** Quantification of CircRreb1 staining in cartilage tissue of wide type and CircRreb1 gKO mice ($n = 9$, biologically independent samples). **l** Safranin O/Fast Green staining of 18-month-old wide type male mice and CircRreb1 gKO male mice ($n = 6$, biologically independent samples). **m** Molecular detection of MMP13, p16, p21, and Aggrecan in Wide type and CircRreb1 gKO mice indicated by IHC staining ($n = 6$, biologically independent samples). **n, o** Osteoarthritis Research Society International (OARSI) and cartilage thickness evaluation in wide type and CircRreb1 gKO mice ($n = 6$, biologically independent samples). **p** Quantification of MMP13, p16, p21, and Aggrecan in wide type and CircRreb1 gKO mice ($n = 6$, biologically independent samples). One-way analysis of variance (ANOVA) followed by Tukey's HSD test is used for (**b, c**, and **g**). Two-sided Student's *t* test is used for (**k, o**, and **p**). Quantitative data is shown as mean ± s.d. The Mann–Whitney *U* test (two sided) is used for *n*. Quantitative data is shown as mean ± 95% CI. Exact *p*-values are shown in the figures. Scar bar for (**f**): SA-β-Gal staining (100 μm, amplification: 50 μm), MMP13, p16, Sox9 staining (50 μm). Scar bar for (**j**): 50 μm, amplification: 10 μm. Scar bar for (**l**): 200 μm, amplification: 50 μm. Scar bar for (**m**): 50 μm. Source data are provided as a Source Data file.

change of CircRreb1 after CircRreb1 AAV infection is not in a large excess of the increases that are seen in the aged mice (Supplementary Fig. 5b, c, and d). DMM surgery caused proteoglycan loss in the femur and tibia cartilage, however, the loss was more prominent after CircRreb1 overexpression (Fig. 3c). Furthermore, the OARSI score showed severe cartilage loss in 18-month-old mice compared to 3-month-old mice after CircRreb1 overexpression (Fig. 3e). Hematoxylin and eosin (H&E) staining showed that CircRreb1 aggravated synovial hyperplasia in 18-month-old mice compared to that in 3-month-old mice (Fig. 3b), as well as molecular detection of fibroblast-like synoviocytes (FLS) marker Vimentin and macrophage marker F4/80 (Supplementary Fig. 6b and c), suggesting aging-related OA severity (Fig. 3d). 3D micro-CT analysis of osteophytes indirectly indicated that CircRreb1 aggravated the severity of aging-related OA (Fig. 3f and g). For molecular detection, representative immunohistochemistry (IHC) images showed aggrecan, chemokine ligand 1 (CXCL1), and p16 staining in the femur and tibia articular chondrocytes between 3-month-old and 18-month-old mice (Fig. 3h). CircRreb1 overexpression after DMM increased CXCL1, p16 expression and decreased aggrecan expression in 18-month-old mice compared to that in 3-month-old mice. Collectively, CircRreb1 aggravated aging-related OA phenotypes in aged mice.

## CircRREB1 mediates lipid metabolism participates in senescent progression in chondrocyte

Metabolic pathway shifts play a vital role in developing various diseases, including OA[6]. Hence, we performed an overall metabolomic analysis between P0 and P2 generation chondrocytes to verify the metabolic pathway in senescent progression (Fig. 4a, b, and d). The Kyoto Encyclopedia of Gene and Genomes (KEGG) enrichment analysis identified lipid metabolism as dominant among other metabolic pathways (Fig. 4d). Numerous clinical studies have revealed an association between lipid accumulation and age-related OA pathogenesis[13,31]. Hence, a heat map showing upregulated different lipid types including fatty acid (FA), glyceride (GL), glycerophospholipids (GP), and glycosphingolipid (SP) between P0 generation chondrocyte group and P2 generation chondrocyte group was showed in Supplementary Fig. 7a. To further confirm lipid metabolism involved in aging, we performed lipid metabolomics in old mice, CircRREB1 overexpression system, and CircRreb1 gKO mice. Most types of lipids including FA, GL (DAG), GP (LPC, LPE, PC, PE), and SP (Cer, HexCer) were increased in 18-month-old aging mice (Supplementary Fig. 7b, c, and d). After CircRREB1 overexpression, most types of lipids were also increased in human chondrocytes (Supplementary Fig. 7e, f, and g). Cartilage samples from WT mice and CircRreb1 gKO mice were

collected and lipid metabolomics showed that some lipids (FA, LPC, LPE, PC, and PE) were decreased after CircRreb1 knockout (Supplementary Fig. 7h, i, and j). As age was one of the most closely related factors for OA development and lipid metabolism was associated with OA, we examined the relationship between lipid synthesis and aging. IHC staining of FASN, ELOVL fatty acid elongase (ELOVL) 5, ELOVL6, and Stearoyl-CoA Desaturase (SCD) 1 in AC (articular chondrocyte) and CCZ (calcified cartilage zone) between younger (50–65 y) and older (70–85 y) were performed. We found that the expression of FASN, ELOVL5, ELOVL6, and SCD1 were increased in older groups, as well as quantifications of the relative intensity of these proteins (Fig. 4e, Supplementary Fig. 7k).

As more lipids were observed in older individuals and older chondrocytes, next, we investigated the relationship between CircRREB1 and lipid synthesis (Fig. 4c). At the mRNA level, *ELOVL5, ELOVL6,* and *SCD1* expression decreased after CircRREB1 knockdown and increased after CircRREB1 overexpression in the chondrocytes (Fig. 4f and g). Protein expression analysis of chondrocytes from WT and CircRreb1 gKO mice showed decreased FASN, ELOVL5, ELOVL6, and SCD1 in CircRreb1 gKO chondrocytes (Fig. 4h). At the pathological level, representative immunohistochemistry images showed that the articular chondrocytes in CircRreb1 gKO mice (18 months) exhibited lower FASN, ELOVL5, ELOVL6, and SCD1 expression (Fig. 4i and j). Chondrocytes from CircRreb1 gKO mice showed few Nile red-positive sections, therefore lower lipid accumulation, suggesting the role of CircRREB1 in lipid synthesis (Fig. 4k and l).

## FASN interacts with CircRREB1 to mediate chondrocyte senescence and OA progression

Cytoplasm-localized circRNAs act as competing endogenous (ceRNAs), coding proteins, or binding proteins, and contribute to disease progression. However, an anti-argonaute RNA immunoprecipitation (AGO2 RIP) assay showed that CircRREB1 did not bind to the AGO2 antibody (Supplementary Fig. 8a). No internal ribosome entry site (IRES) element was found in the CircRREB1 sequence (Supplementary Fig. 8b). Bioinformatic analysis identified open reading frames (ORFs) in CircRREB1 (Supplementary Fig. 8c and d). Full-length ORFs were cloned into the pcDNA vector; however, CircRREB1 did not encode proteins, as indicated by western blotting (Supplementary Fig. 8e). We performed RNA pull-down and RPD-MS to explore and identify proteins interacting with CircRREB1 (Fig. 5a). Based on the sum of pep_scores among the RPD-MS results, we listed the top 20 proteins interacting with CircRREB1 (Supplementary Table 2). As shown in previous results, CircRREB1 mediates lipid metabolism and participates in the senescent progression of chondrocytes. We speculated

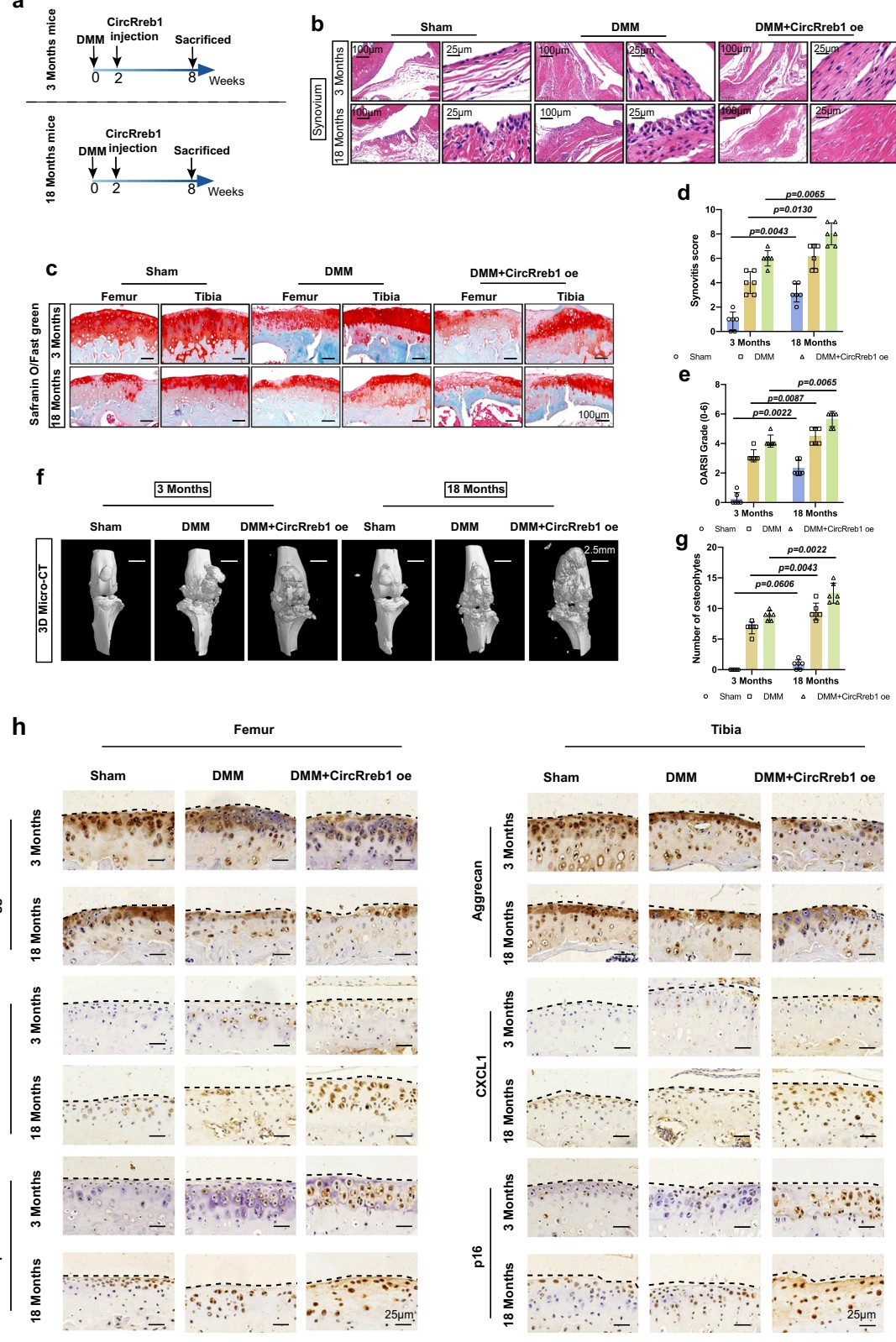

that lipid metabolism-related proteins along with CircRREB1 participate in OA and senescent regulation. We found that FASN was among the top 20 proteins that interacted with CircRREB1 (Supplementary Table 2). The interaction between FASN and CircRREB1 was identified by western blotting in HCs (Fig. 5b) and MCs (Supplementary Fig. 8h). FISH staining showed that CircRREB1 co-localized with FASN in the cytoplasm (Fig. 5c). CatRAPID (http://service.tartaglialab.com/page/

catrapid_group) was used to predict the binding sites of CircRREB1 and FASN (Supplementary Fig. 8g), suggesting that position (300–400) was combined with FASN. CircRNA loops are critical for RNA interactom[32]. We then predicted the loop structure of CircRREB1 using *RNAfold* WebServer (http://rna.tbi.univie.ac.at//cgi-bin/RNAWebSuite/RNAfold.cgi). To determine which structure is required for the interaction between CircRREB1 and FASN, we truncated full-length

**Fig. 3 | CircRreb1 aggravates age-related OA degree in aging mice. a** Illustration of DMM operation and CircRreb1 intra-articular infection in 3-month-old and 18-month-old male C57BL/6 J mice. 3-month-old mice: sham group ($n = 6$), DMM group ($n = 6$), DMM+CircRreb1 oe group ($n = 6$); 8-month-old mice: sham group ($n = 6$), DMM group ($n = 6$), DMM+CircRreb1 oe group ($n = 6$). **b** Synovial hyperplasia post-DMM operation and DMM added with CircRreb1 injection in 3-month-old and 18-month-old mice. **c** Safranin O/Fast Green staining of femur and tibia post-DMM operation, and DMM added with CircRreb1 injection in 3-months-old and 18-month-old mice. **d** Evaluations of synovial hyperplasia by synovitis score ($n = 6$, biologically independent samples). **e** OARSI evaluation in 3-month-old and 18-month-old mice ($n = 6$, biologically independent samples). **f** Representative images of knee joint osteophytes indicated by 3D micro-CT analysis. **g** Quantification of osteophyte number ($n = 6$, biologically independent samples). **h** Molecular detection of Aggrecan, CXCL1, and p16 in articular cartilage (Femur and Tibia) post-DMM operation and DMM added with CircRreb1 overexpression indicated by representative immunohistochemistry (IHC) images in 3-month-old and 18-month-old mice. $n = 6$ mice (sham, DMM, and DMM+Circrreb1 oe) for 3-month-old and 18-month-old mice. The Mann–Whitney $U$ test (two sided) is used for statistical analysis (**d**, **e**, and **g**). Quantitative data is shown as mean ± 95% CI. Exact $p$-values are shown in the figures. Scar bar for (**b**): 100 μm, amplification: 25 μm. Scar bar for (**c**): 100 μm. Scar bar for (**f**): 2.5 mm. Scar bar for (**h**): 25 μm. Source data are provided as a Source Data file.

CircRREB1 into five fragments (Supplementary Fig. 8f). RIP assay revealed that full-length CircRREB1 and fragment 2 were enriched by FASN (Fig. 5d), furthermore, fragment 2 of CircRREB1 is consistent with the position (300–400) predicted by CatRAPID software (Supplementary Fig. 8g). Next, the direct interaction between CircRREB1 and FASN was further validated by electrophoretic mobility shift assay (EMSA) (Fig. 5f). To detect which domain of FASN combined with CircRREB1, the full-length FASN was truncated into seven fragments (Fig. 5e). Full-length FASN and fragment 2 were enriched by CircRREB1 (Fig. 5g). Collectively, CircRREB1 directly interacts with FASN.

To further investigate the function of FASN in ECM metabolism and senescent phenotypes in chondrocytes, HCs, and MCs were infected with three FASN short hairpin (sh) RNA adenoviruses, which significantly decreased *FASN* or *Fasn* mRNA expression (Supplementary Fig. 9a and c). FASN or Fasn knockdown in HC/MCs increased the expression of ECM components and decreased ECM-degrading enzymes, and senescence-related genes (Supplementary Fig. 9b, d). Western blot analysis of FASN or Fasn knockdown in HC/MCs also showed similar results (Supplementary Fig. 9e and f). Furthermore, FASN or Fasn knockdown in HC/MCs enhanced extracellular matrix formation (Supplementary Fig. 9g). To investigate the senescent phenotypes in HCs/MCs after FASN or Fasn knockdown, SA-β-Gal assay was performed. Representative images of SA-β-Gal after FASN or Fasn knockdown in HCs/MCs (Supplementary Fig. 9h and i), and quantification of SA-β-Gal positive area (Supplementary Fig. 9j and k), revealed that FASN or Fasn knockdown significantly decreased senescent phenotypes in chondrocytes. Subsequently, gain-of-function experiments were performed. CRISPR/Cas9 synergistic activation mediator (SAM) was used to overexpress FASN in HCs. Three sgRNAs that infected HCs significantly upregulated *FASN* mRNA expression (Supplementary Fig. 9l). Overexpression of FASN in HCs decreased *Col2*, and *Sox9* expression and increased *MMP13*, *ADAMTS5*, *p16*, and *p21* expression (Supplementary Fig. 9m and n). Meanwhile, FASN overexpression increased FASN, p21, p16, p53, ADAMTS5, and MMP13 proteins expression, and decreased Col2 and Sox9 proteins expression (Supplementary Fig. 9o). Next, the senescent phenotype after FASN overexpression was evaluated using SA-β-Gal staining, and the results indicated that FASN overexpression accelerated senescent phenotype formation (Supplementary Fig. 9p).

After evaluating the functions of FASN in vitro, we investigated the impact of FASN on OA and senescent phenotypes in an in vivo study. Animals underwent DMM operation, followed by Fasn AAV injections (approximately $1 \times 10^{12}$ vg/mL) two weeks post-injury (Supplementary Fig. 10a). Immunofluorescence staining showed the Fasn expression levels among these four groups, as well as the aging mice model (Supplementary Fig. 5e and f), however, the fold change of Fasn after Fasn AAV infection (Supplementary Fig. 5h) is not in large excess of the increases that are seen in the aged mice (Supplementary Fig. 5g). Safranin O/Fast green and Alcian blue staining showed that Fasn overexpression after DMM operation accelerated cartilage loss (Supplementary Fig. 10b), which was also indicated by the OARSI score evaluation and cartilage thickness of the knee joint (Supplementary Fig. 10c and d). In addition, H&E staining of the synovium indicated that Fasn overexpression enhanced synovial hyperplasia (Supplementary Fig. 10e), as indicated by the synovitis score (Supplementary Fig. 10f). In addition, we also observed that Fasn overexpression caused the upregulation of Vimentin positive fibroblast-like synoviocytes (FLS) and F4/80 positive macrophage in synovium significantly (Supplementary Fig. 6a). Osteophytes in the knee joint were evaluated by micro-CT analysis (Supplementary Fig. 10g). The number of osteophytes in the knee joint showed that Fasn overexpression after DMM surgery enhanced the degree of OA (Supplementary Fig. 10h). For molecular level detection, representative images illustrated that Fasn overexpression increased p16 and p21 expression and decreased Aggrecan expression (Supplementary Fig. 10i), as well as quantification of p16-, p21-, and aggrecan-positive chondrocytes (Supplementary Fig. 10j), accelerating senescent and OA phenotypes. Collectively, Fasn plays an important role in aggravating senescence and OA phenotypes.

To further explore the mechanism underlying how FASN mediated lipid metabolism causes cellular senescence, we performed FASN SiRNAs transfection in HCs. FASN silencing in chondrocytes significantly decreased FASN expression and palmitic acid production (Supplementary Fig. 11a), thereby reducing lipid accumulation indicated by Nile red staining (Supplementary Fig. 11b). A large number of evidence suggested that lipid accumulation was observed in OA cartilage[33], followed by reactive oxygen species (ROS) generation by lipid peroxidation[34]. Herein, we investigated whether lipid accumulation mediated by FASN cause chondrocyte damage by ROS pathway. Flow cytometry showed that total ROS levels were significantly upregulated in TBHP-stimulated chondrocytes, while decreased after FASN knockdown (Supplementary Fig. 11c and d). Representative images of FITC-ROS in chondrocytes showed the same result with flow cytometry assay (Supplementary Fig. 11e and g). Moreover, 4-hydroxynonenal (4-HNE), a product of lipid peroxidation, was also decreased when FASN silencing in TBHP-induced chondrocytes (Supplementary Fig. 11f). Next, we performed mRNA-seq (Majorbio Bio-pharm Technology Co., Ltd, Shanghai, China) to detect the downstream genes after FASN knockdown. Data were analyzed using the Majorbio Cloud Platform (www.majorbio.com). KEGG enrichment indicated that cellular senescence was highly associated FASN (Supplementary Fig. 11h). A heat map showed differentially expressed genes associated with cellular senescence when FASN knockdown (Supplementary Fig. 11i). To verify whether FASN knockdown also regulates cellular senescence in TBHP-induced chondrocytes, RT-qPCR also demonstrated that FASN inhibition in TBHP-induced chondrocytes regulated these genes including *RASSF5*, *MAPK3*, *IL6*, *CDK2*, *NFKB1*, *CXCL8*, *ZFP36L2*, *MAPK14*, *NRAS*, *RBL2*, and *LIN9* (Supplementary Fig. 11k), suggesting FASN regulated chondrocyte senescence after TBHP treatment. Furthermore, we observed that the expressions of senescence markers were decreased when FASN knockdown in TBHP-induced chondrocytes indicated by immunofluorescence of p16, p21, and gama-H2AX nuclear foci (Supplementary Fig. 11j), western blot analysis of p16 and p21 (Supplementary Fig. 11l), and SA-β-Gal staining (Supplementary Fig. 11m).

To examine the effects of the CircRREB1-FASN axis on senescence and OA progression, a rescue experiment was performed. In brief, CircRREB1 was knockdown in chondrocytes, followed by FASN

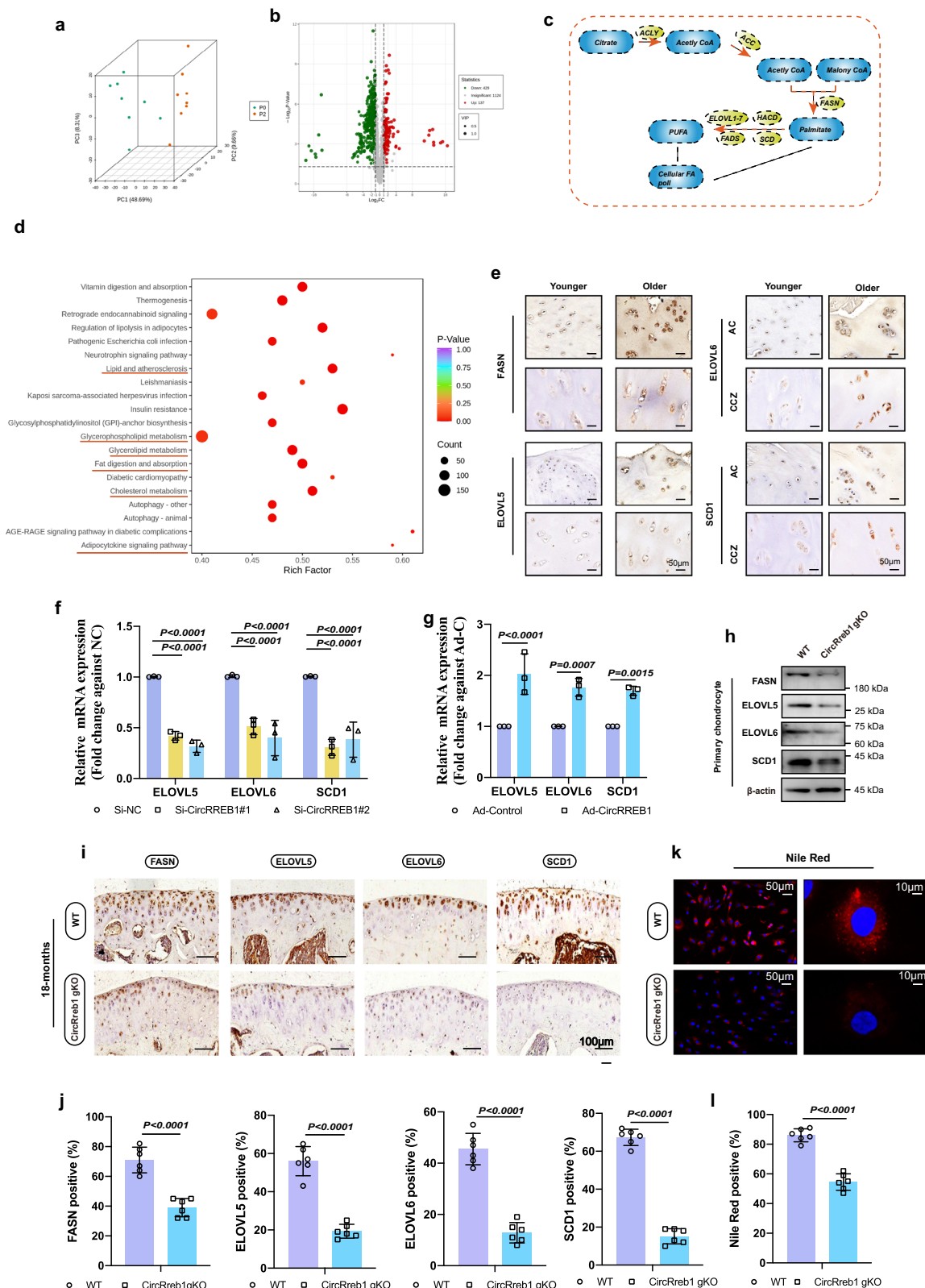

overexpression. RT-qPCR and western blot analysis indicated that matrix degrading enzymes like MMP13 and ADAMTS5 and senescence associated markers p16, p21, and p53 were increased, while Col2 and Sox9 were decreased after FASN overexpression, suggesting FASN was the downstream molecule of CircRREB1 (Supplementary Fig. 12a and c). We next investigated whether FASN was required for the role of CircRREB1 in age-related OA development. The effects of CircRREB1 on

age-related OA development were evaluated when FASN was silenced. The result showed that CircRREB1 overexpression did not change the expression of Col2, Sox9, ADAMTS5, p16, p21, and p53 when FASN silenced (Supplementary Fig. 12b and d), suggesting that FASN mediates the function of CircRREB1 in OA. The role of the CircRREB1-FASN axis in promoting age-related OA progression was confirmed using the above data.

**Fig. 4 | CircRREB1/CircRreb1 mediates lipid metabolism in aging development.** **a** Quality control between P0 and P2 generation chondrocytes in overall metabolomics experiment ($n = 8$, biologically independent samples). **b** differentially expressed metabolites in a volcano map. **c** Schematic illustration of de novo lipogenesis. **d** KEGG enrichment between P0 and P2 generation chondrocytes in an overall metabolomics experiment. **e** FASN, ELOVL5, ELOVL6, and SCD1 expression in younger articular and older articular tissue shown in IHC images. **f, g** *ELOVL5, ELOVL6,* and *SCD1* mRNA expression in HCs infected with CircRREB1 SiRNAs or Ad-CircRreb1 indicated by RT-qPCR ($n = 3$, biologically independent samples). **h** FASN, ELOVL5, ELOVL6, and SCD1 expression in mouse chondrocytes from wide type or CircRreb1 gKO mice. Blots are representative of three independent experiments. **i** FASN, ELOVL5, ELOVL6, and SCD1 expression in wide type or CircRreb1gKO mice.

(18-month-old, $n = 6$, biologically independent samples). **j** Quantification of FASN, ELOVL5, ELOVL6, and SCD1 expression levels in chondrocytes in wide type or CircRreb1 gKO mice ($n = 6$, biologically independent samples). **k** Lipid accumulation in chondrocytes from wide type or CircRreb1gKO mice indicated by Nile red staining. ($n = 6$, biologically independent samples). **l** Quantification of Nile red positive staining ($n = 6$, biologically independent samples). One-way analysis of variance (ANOVA) followed by Tukey's HSD test is used for (**f**). Two-sided Student's *t* test is used for (**g, j,** and **l**) panels. Quantitative data are shown as mean ± s.d. Exact *p*-values are shown in the figures. Scar bar for (**e**): 50 μm. Scar bar for (**i**): 100 μm. Scar bar for (**k**): 50 μm, amplification: 10 μm. Source data are provided as a Source Data file.

## CircRREB1 stabilizes FASN and maintains its function by inhibiting proteasome-mediated degradation

Our investigation indicated that knockdown or overexpression of CircRREB1 regulated FASN at the protein level but not at the mRNA level in HCs/MCs (Fig. 5h and i, Supplementary Fig. 8i and j), suggesting that CircRREB1 post-transcriptionally regulates FASN expression in chondrocytes. Next, we examined the FASN degradation pathway in HCs. Chondrocytes were treated with cycloheximide (CHX) for 2, 4, and 8 h to block FASN protein synthesis. Western blot analysis indicated that FASN protein levels gradually decreased after CHX treatment (Fig. 5j). The main pathways of protein degradation include the proteasome and lysosomal degradation pathways. To determine which degradation pathway is involved in FASN degradation, we used the lysosomal degradation pathway inhibitor, chloroquine (CQ), and the proteasome degradation pathway inhibitor, bortezomib (Borz). After treated with CQ or Borz for 2, 4, and 8 h, western blot analysis indicated that FASN protein levels increased gradually in HCs treated with Borz, whereas no noticeable difference was observed in HCs treated with CQ (Fig. 5k and l), suggesting that proteasomal pathway predominantly degrades FASN. Apparent differences in FASN half-life were observed between the Si-negative control (NC) and Si-CircRREB1 HCs after CHX treatment, while CircRREB1 overexpression showed the opposite result (Fig. 5m–p). These data suggest that CircRREB1 enhances the stabilization of FASN. Next, our investigation indicated that Borz treatment increased FASN protein levels after CircRREB1 knockdown in HC/MCs (Fig. 5q, Supplementary Fig. 8k). Next, we investigated the ubiquitination level of FASN. Western blot analysis showed that the ubiquitination level of FASN was upregulated after CircRREB1 knockdown and downregulated following CircRREB1 overexpression in the endogenous pathway (Fig. 5r). Cumulatively, CircRREB1 stabilized FASN by inhibiting the ubiquitin-proteasome degradation pathway.

## CircRREB1 mediates post-translational modifications to regulate FASN function

Chondrocytes treated with nicotinamide (NAM), an inhibitor of the SIRT family deacetylases, and trichostatin A (TSA), an inhibitor of histone deacetylase (HDAC) I and II, respectively. Western blot analysis showed that TSA treatment instead of NAM affected FASN protein level (Fig. 6a). Next, Co-IP assay result indicated that the acetylation level of FASN was increased by TSA, but not by NAM (Fig. 6b and c). Moreover, TSA treatment enhanced the ubiquitination level of FASN (Fig. 6b). Due affected of TSA treatment, we would like to find out which specific deacetylase is involved in FASN acetylation. In detail, Flag-FASN and different HDAC family deacetylase plasmids were co-transfected into chondrocytes, Co-IP result showed that HDAC3 interacted with FASN (Fig. 6d). To visualize the effect that CircRREB1 recruit FASN and HDAC3, we performed molecular docking (Fig. 6h). The molecular docking showed the interaction between CircRREB1, FASN and HDAC3. To further confirmed the function of CircRREB1 on this complex (CircRREB1, FASN, and HDAC3), we performed RIP assay and confirmed the interaction between CircRREB1 and HDAC3 (Fig. 6f).

To further investigate the relationship between CircRREB1, FASN and HDAC3, CircRREB1 knockdown or overexpression were performed respectively in chondrocytes and we found that CircRREB1 overexpression enhanced the interaction between FASN and HDAC3, and then decreased FASN acetylation, thereby decreased FASN ubiquitination (Fig. 6g). However, CircRREB1 knockdown showed the opposite result (Fig. 6g). Above all, CircRREB1 recruited FASN and HDAC3 to impede the acetylation of FASN.

Due to the increased ubiquitination level of FASN mediated by TSA treatment, we determined to find out which E3 ligases participated in FASN proteasome degradation. E3 ligase prediction was performed using the UbiBrowser (http://ubibrowser.ncpsb.org.cn). We selected STUB1 and MDM2 as potential E3 ligases involved in FASN degradation (Fig. 6i). To further explore the effects of STUB1 and MDM2 on FASN protein level, we overexpressed STUB1 and MDM2 in chondrocytes respectively and western blot analysis indicated that MDM2 significantly reduced FASN protein level compared to STUB1 overexpression (Fig. 6j). Flag-FASN and Myc-MDM2 plasmids were co-transfected into chondrocytes, and a co-immunoprecipitation (Co-IP) assay was performed. The results revealed an interaction between Flag-FASN and Myc-MDM2 (Fig. 6k); consistently, FASN co-localized with MDM2 in the cytoplasm (Fig. 6l). The effects of MDM2 knockdown and overexpression were also investigated in the next step. Two MDM2 SiRNAs and an MDM2 plasmid were transfected into HCs, which significantly decreased *MDM2* mRNA expression or increased *MDM2* expression but did not affect *FASN* mRNA levels (Supplementary Fig. 13a), suggesting MDM2 regulates FASN post-translationally.

To date, post-translational regulation of FASN has been the primary focus of research. Crosstalk between ubiquitination and other modifications plays an essential role in protein function regulation[35,36]. To determine which acetylation sites of FASN, we performed UPLC-MS/MS, which indicated two acetylation sites, K673 and K1065, in FASN (Fig. 6e). We then mutated K673 and K1065 from lysine (K) to arginine (R) to determine the acetylation site of FASN. Flag-FASN-WT, Flag-FASN-K673R, and Flag-FASN-K1065R plasmids were co-transfected into chondrocytes. After treatment with MG-132 and TSA, the Co-IP assay indicated that K673 and K1065 mutants decreased acetylation of FASN, with a more prominent effect of the K673 mutant (Fig. 6m), indicating that K673 is the primary acetylation site on FASN. The K673 mutant on FASN also decreased the ubiquitination level of FASN (Fig. 6m). Furthermore, K673 on FASN was highly conserved among the different species (Fig. 6n). Next, we investigated the effect of CircRREB1 on FASN acetylation. CircRREB1 overexpression enhanced the interaction between FASN and HDAC3, while decreased after K673 mutated (Fig. 6o). CircRREB1 knockdown increased FASN acetylation, followed by enhanced interaction with MDM2 and increased ubiquitination of FASN (Fig. 6p). However, after K673 on FASN mutated, CircRREB1 knockdown showed no effect on FASN acetylation, or FASN ubiquitination (Fig. 6p). Gain-of-function experiments were then performed. CircRREB1 overexpression decreased FASN acetylation, followed by FASN ubiquitination reduction, whereas no noticeable difference was observed after the K673 mutation

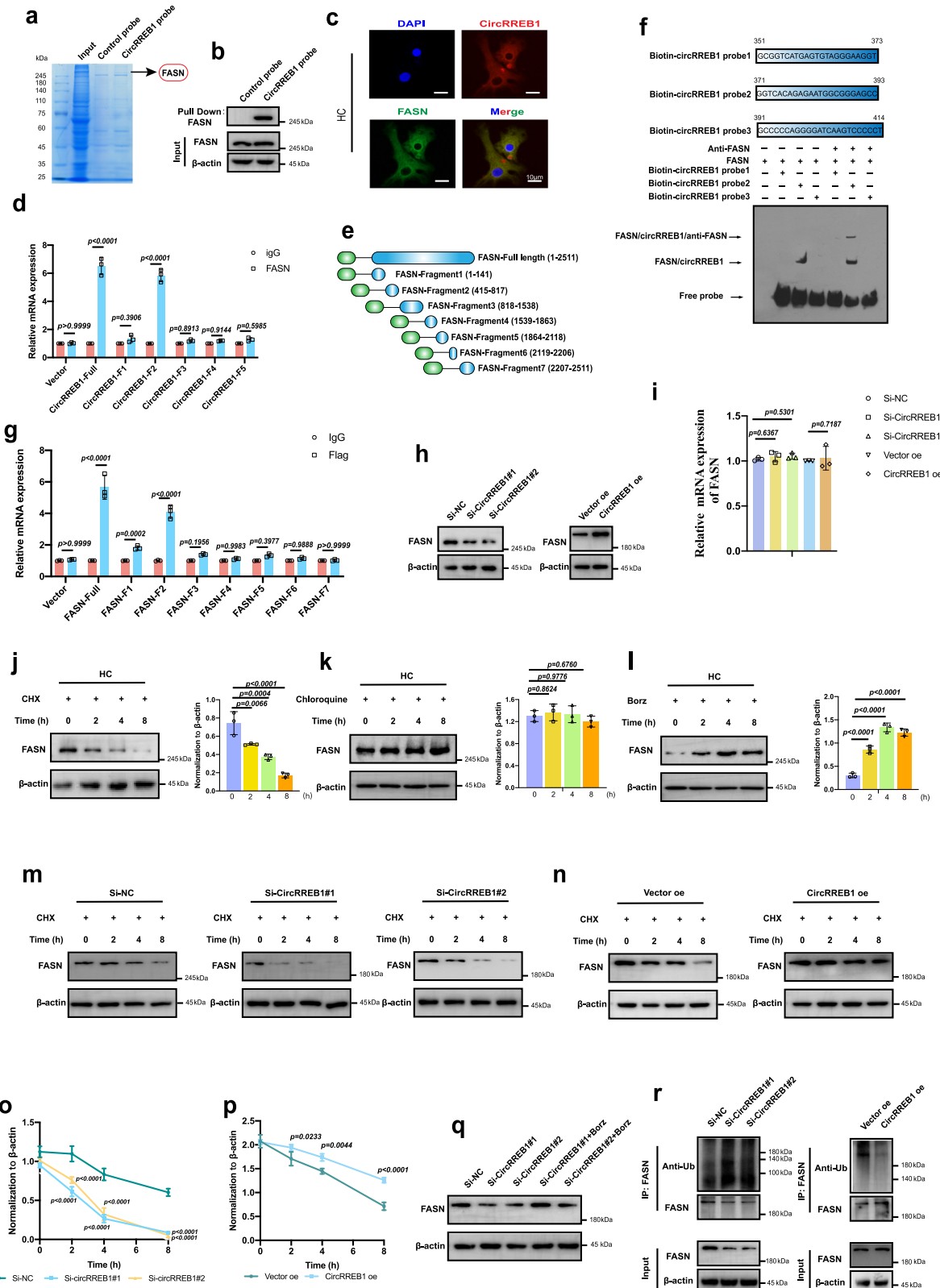

(Fig. 6p). Meanwhile, CircRREB1 knockdown and overexpression did not affect MDM2 protein expression (Fig. 6p). K48 specific type of ubiquitination is widely involved in protein degradation[37,38]. Herein, we would like to investigate whether CircRREB1 regulates FASN ubiquitination via K48 specific ubiquitination. Co-IP result indicated that CircRREB1 overexpression decreased MDM2 mediated K48 specific type ubiquitination (Supplementary Fig. 13b). Collectively, CircRREB1

inhibited acetylation-mediated ubiquitination of FASN, thereby stabilizing FASN and regulating its function.

SUMOylation has been reported to prevent proteasome degradation and maintain protein function[39,40]. Here, we found that E3 SUMO-protein ligase RanBP2, which was the intersection of CircRREB1 pulldown result and Flag-FASN MS result (Fig. 6q), Flag-FASN MS result shown in Supplementary Table 3. Two RanBP2 siRNAs were

**Fig. 5 | CircRREB1 promotes FASN stability and inhibits its degradation via proteasome. a** Coomassie Blue stain of protein bound to CircRREB1. Blots are representative of three independent experiments. **b** CircRREB1 interacts with FASN after a pull-down assay, indicated by western blot analysis. Blots are representative of three independent experiments. **c** CircRREB1 is colocalized with FASN in HC. **d** Binding sequence of CircRREB1 interacts with FASN identified by RNA immunoprecipitation (RIP) assay ($n = 3$, biologically independent samples). **e** Schematic illustration of FASN according to FASN domain. **f** Direct interaction between CircRREB1 and FASN identified by Electrophoretic mobility shift assay (EMSA). **g** Binding domain of FASN interacts with CircRREB1, identified by RIP assay ($n = 3$, biologically independent samples). **h** FASN protein expression in HCs infected with CircRREB1 siRNAs or Ad-CircRreb1. **i** *FASN* mRNA expression in HCs infected with CircRREB1 siRNAs or Ad-CircRreb1 ($n = 3$, biologically independent samples). **j** the half-life of FASN protein in HCs treated with Cycloheximide (CHX) for 2, 4, and 8 h

($n = 3$, biological independent samples). **k** The half-life of FASN protein in HCs treated with chloroquine (CQ) for 2, 4, and 8 h. $n = 3$, biological independent samples. **l** The half-life of FASN protein in HCs treated with Bortezomib (Borz) for 2, 4, and 8 h ($n = 3$, biological independent samples). **m, n, o, p** The half-life of FASN in HCs infected with CircRREB1 siRNAs or Ad-CircRreb1 ($n = 3$, biological independent samples). Blots are representative of three independent experiments. **q** The effect of Borz on FASN protein level alteration mediated by CircRREB1 knockdown. Blots are representative of three independent experiments. **r** Ubiquitinated FASN after CircRREB1 knockdown and overexpression, indicated by immunoprecipitation. Blots are representative of three independent experiments. One-way analysis of variance (ANOVA) followed by Tukey's HSD test is used for panels (**i, j, k, l, o**, and **p**). Quantitative data shown as mean ± s.d. Two-sided Student's $t$ test is used for (**d, g**). Quantitative data are shown as mean ± s.d. Exact $p$-values are shown in the figures. Scar bar for (**b**): 10 μm. Source data are provided as a Source Data file.

transfected into HCs, which significantly decreased *RanBP2* mRNA expression, but did not affect *FASN* mRNA levels (Supplementary Fig. 13c). After RanBP2 knockdown, western blot analysis showed that FASN protein levels significantly decreased (Supplementary Fig. 13d), which suggested that RanBP2 mediated post-translational modification of FASN protein levels. Furthermore, RanBP2 knockdown did not affect ubiquitin-conjugating enzyme 9 (UBC9) or free SUMO2/3 protein levels (Supplementary Fig. 13d). Co-IP result indicated that RanBP2 knockdown decreased the interaction between FASN and UBC9, followed by a reduction of FASN SUMOylation (Supplementary Fig. 13e). We then investigated how CircRREB1 affected FASN SUMOylation. To evaluate the stability of FASN, chondrocytes with or without RanBP2 knockdown were treated with CHX for 2, 4, or 8 h. Obvious differences in FASN half-life were observed between Si-NC and Si-CircRREB1 HCs after CHX treatment, suggesting that RanBP2 knockdown weakened the stability of FASN (Fig. 6r–t). After CircRREB1 knockdown, the Co-IP assay showed that the interaction between FASN and RanBP2 decreased, as well as the interaction between FASN and UBC9, followed by FASN SUMOylation downregulation (Fig. 6u). CircRREB1 overexpression produced the opposite result (Fig. 6v).

Overall, CircRREB1 inhibited acetylation-mediated ubiquitination of FASN and promoted RanBP2-mediated SUMOylation of FASN to stabilize and maintain its functions.

### CircRREB1-FASN axis regulates FGFR3 and FGF18-related PI3K-AKT signaling pathway in chondrocyte

Next, we performed mRNA-seq (Majorbio Bio-pharm Technology Co., Ltd, Shanghai, China) to detect the downstream genes after CircRREB1 knockdown. Data were analyzed using the Majorbio Cloud Platform (www.majorbio.com). Among the differentially expressed genes in Si-NC and Si-CircRREB1, 253 were upregulated, and 273 were down-regulated (Supplementary Fig. 14a). GO enrichment analysis and KEGG pathway analysis revealed different functions after CircRREB1 knockdown (Supplementary Fig. 14b and d). KEGG enrichment analysis also identified PI3K-AKT signal transduction as a significant downstream signaling pathway (Supplementary Fig. 14c). Among the different genes related to the PI3K-AKT signaling pathway, *FGFR3* and *FGF18* were significantly upregulated following CircRREB1 knockdown (Supplementary Fig. 9e, Supplementary Table 4). Therefore, we speculated whether CircRREB1 and FASN also regulate *FGFR3* and *FGF18* expression. RT-qPCR analysis revealed that CircRREB1 or FASN knockdown also significantly increased *FGFR3* and *FGF18* mRNA expression, whereas CircRREB1 or FASN overexpression decreased their expression (Supplementary Fig. 14f, and g). Western blot analysis also indicated this result (Supplementary Fig. 14h). This result confirmed *FGFR3* and *FGF18* as the downstream targets of the CircRREB1-FASN axis. Next, we investigated how the CircRREB1-FASN axis regulates FGFR3, FGF18, and the PI3K-AKT signaling pathway. CircRREB1 knockdown upregulated FGFR3, FGF18, phospho-PI3K (p-PI3K), and phospho-AKT (p-AKT) protein levels, FASN overexpression reversed this effect,

including the signal transduction molecule induced by CircRREB1 knockdown (Supplementary Fig. 14i). AKT activation is a core link in the PI3K-AKT signaling pathway. Representative immunofluorescence images show that CircRREB1 and FASN knockdown induced p-AKT nuclear translocation, whereas FASN overexpression after CircRREB1 knockdown prevented this phenomenon (Supplementary Fig. 14j).

Then, a PI3K inhibitor (pictilisib) and an AKT inhibitor (MK-2206) were used to perform rescue experiments. Treating chondrocytes with pictilisib and MK-2206 decreased p-PI3K, PI3K, p-AKT, and AKT levels, a combination of PI3K inhibitor and AKT inhibitor decreased AKT activation significantly (Supplementary Fig. 15a). Chondrocytes were treated with a combination of PI3K inhibitor and AKT inhibitor, and senescence markers p16, and p21 were increased after treatment (Supplementary Fig. 15b). PI3K-AKT-mTOR is a classic pathway, herein, we designed to investigate whether CircRREB1 or FASN knockdown affects mTOR pathway. OA chondrocytes were treated with CircRREB1 and FASN SiRNAs. Western bolt analysis indicated that CircRREB1 or FASN knockdown together promoted PI3K and AKT activation but showed little effect on mTOR activation (Supplementary Fig. 15c and d), suggesting CircRREB1 or FASN did not affect mTOR associated autophagy. Then, chondrocytes with CircRREB1 knockdown treated with PI3K inhibitor and AKT inhibitor showed that the effects of Col2, MMP13, p16, and p21 expression induced by CircRREB1 knockdown were reversed (Supplementary Fig. 15e). Representative immunofluorescence images of Col2, aggrecan, MMP13, and p16 also revealed that combined inhibition reversed CircRREB1 knockdown effects (Supplementary Fig. 10f). Overall, the CircRREB1-FASN axis regulated the FGFR3 and FGF18-related PI3K-AKT signaling pathway to regulate senescence associated CDK inhibitors to participate in OA and senescent progression in chondrocytes.

### CircRreb1 overexpression reverses the alleviation of OA and senescent phenotypes in CircRreb1 cKO mice

We performed rescue studies to investigate the role of CircRreb1 overexpression in OA and senescent phenotypes in cKO mice. CircRreb1 cKO mice underwent DMM and received TAM injections 5 times one week post-injury, followed by CircRreb1 intra-articular infection on day 14 post DMM operation. (Fig. 7a). CircRreb1 cKO mice exhibited more safranin O positive areas than did control mice; however, animals that received DMM showed cartilage loss and disruption, which was more prominent after CircRreb1 overexpression (Fig. 7b). The OARSI grade also indicated that CircRreb1 accelerated cartilage damage in CircRreb1 cKO mice compared to that in animals that received only DMM operation (Fig. 7d). Representative H&E images show that CircRreb1 aggravated the severity of OA. Fewer synovitis cases were observed in CircRreb1 cKO mice than in control mice (Fig. 7c). However, animals that received DMM and CircRreb1 AAV injection showed more severe synovial hyperplasia than the group that received DMM only (Fig. 7c, e). Furthermore, more Vimentin positive FLSs and F4/80 positive macrophages were observed after CircReeb1 infection

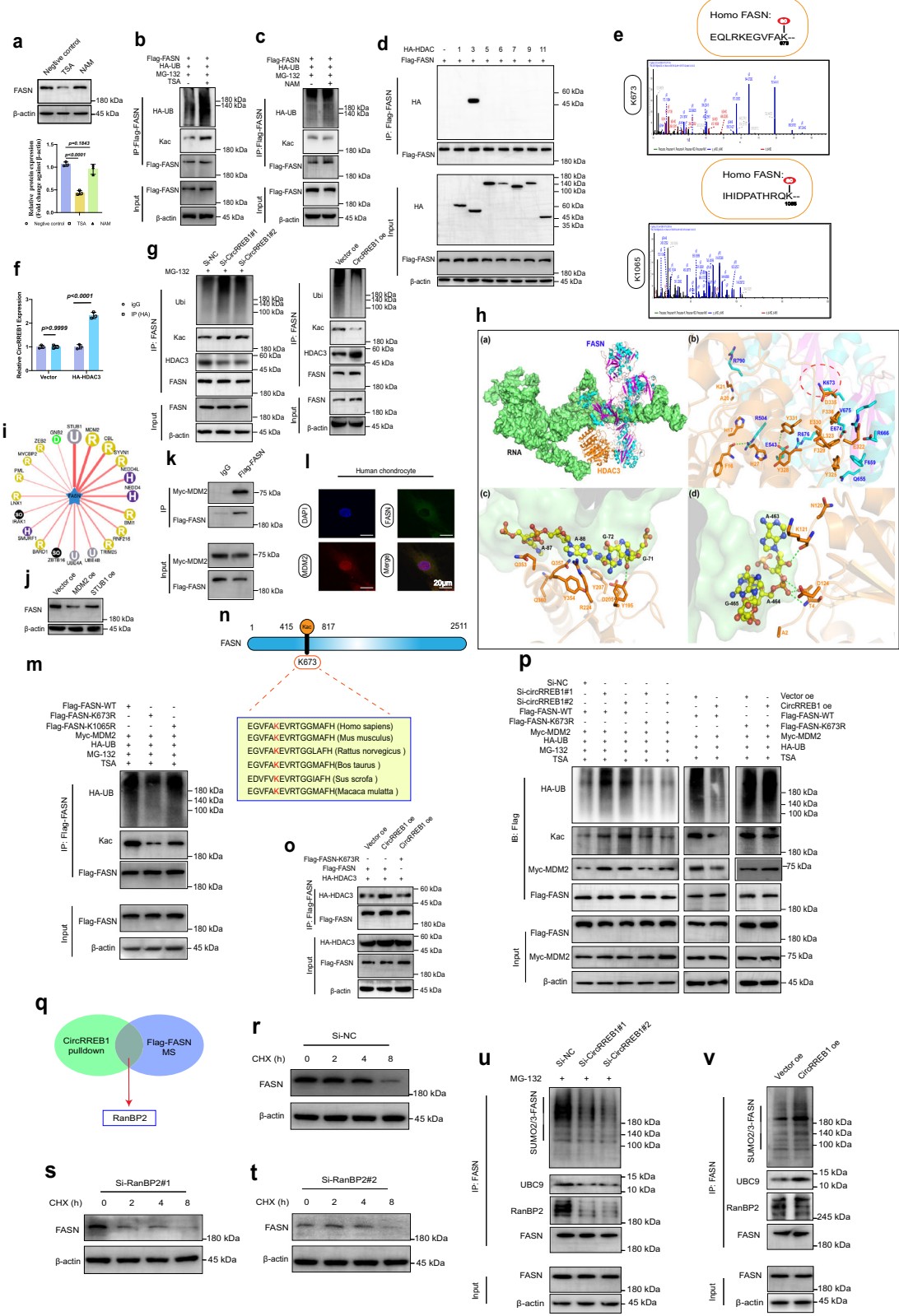

compared to the group that received DMM operation only (Supplementary Fig. 6d). These data indicated that OA and senescent phenotypes were reversed after CircRreb1 overexpression in CircRreb1 cKO mice. At the molecular level, SASP molecules, such as CXCL1, IL-6, and MMP13, were downregulated in CircRreb1 cKO mice compared to those in control mice (Fig. 7f). However, CircRreb1 cKO mice that received DMM surgery and CircRreb1 overexpression showed more

SASP positive cells than animals that received only DMM (Fig. 7f). Furthermore, CircRreb1 overexpression increased the expression of senescence-related indicators p16 and p21 after DMM (Fig. 7g). These senescent phenotypes accelerated OA phenotypes, such as a decreased aggrecan-positive area in CircRreb1 cKO mice (Fig. 7g). We also investigated the downstream effects of CircRreb1 in CircRreb1 cKO mice. FASN levels were decreased in CircRreb1 cKO mice;

**Fig. 6 | CircRREB1 mediates post-translational modifications of FASN to maintain its function. a** Upper panel, Chondrocytes treated with trichostatin (TSA) or nicotinamide (NAM), FASN protein expression is examined by western blot analysis. Lower panel, quantifications of FASN protein expression after TSA or NAM treatment (n = 3, biological independent samples). **b, c** TSA treatment, but not NAM, increases FASN acetylation level and ubiquitination level. Blots are representative of three independent experiments. **d** Co-IP assay is performed and HDAC3 is confirmed as a specific deacetylase involved in FASN acetylation. Blots are representative of three independent experiments. **e** Ultra-performance liquid chromatography-tandem mass spectrometry (UPLC-MS/MS) is performed to find out the acetylation site on FASN. **f** HDAC3 interacts with CircRREB1 confirmed by RIP assay (n = 3, biological independent samples). **g** CircRREB1 knockdown or overexpression affects FASN acetylation, ubiquitination level, and interaction between FASN and HDAC3. Blots are representative of three independent experiments. **h** Molecular docking of CircRREB1 complex (CircRREB1, HDAC3, and FASN). **i** Prediction of a potential interaction between E3 ligase and FASN. **j** Effect of STUB1 and MDM2 overexpression on FASN protein level. Blots are representative of three independent experiments. **k** The interaction between FASN and MDM2 was confirmed by co-IP assay. Images are representative of three independent experiments.

**l** FASN co-localized with MDM2 in chondrocyte cytoplasm. **m** The effects of the K673 and K1065 mutants on FASN acetylation and ubiquitination, K673 is a primary acetylation site on FASN. Blots are representative of three independent experiments. **n** Conservation of the K673 site on FASN. **o** The effect of CircRREB1 overexpression on the interaction between K673R FASN and HDAC3. Blots are representative of three independent experiments. **p** the effect of CircRREB1 and overexpression on K673R FASN acetylation and ubiquitination levels, detected by IP assay. Blots are representative of three independent experiments. **q** Schematic illustration of the intersection between CircRREB1 pulldown mass spectrometry (MS) and Flag-FASN MS. **r–t** FASN protein levels in HCs infected with RanBP2 siRNAs and a negative control after treatment with CHX for 2, 4, and 8 h. Blots are representative of three independent experiments. **u, v** The effects of CircRREB1 knockdown and overexpression on FASN SUMOylation, indicated by IP assay. Blots are representative of three independent experiments. One-way analysis of variance (ANOVA) followed by Tukey's HSD test is used for a. Two-sided Student's t test is used for (**f**). Quantitative data are shown as mean ± s.d. Exact p-values are shown in the figures. The scar for l: 20 μm. Source data are provided as a Source Data file.

---

however, FASN was upregulated following DMM surgery and was more prominent with CircRreb1 overexpression, followed by FGFR3, FGF18, p-PI3K, and p-AKT (Fig. 7h). The role of CircRreb1 in lipid metabolism was investigated in CircRreb1 cKO mice. Fewer Nile red-positive cells were observed in CircRreb1 cKO mice than in control mice (Fig. 7i). However, the DMM surgery reversed this phenomenon. Greater lipid accumulation was observed in the CircRreb1 cKO mice + DMM group than in the CircRreb1 cKO group, which was more prominent after CircRreb1 overexpression (Fig. 7i, j).

## Discussion

We identified that CircRREB1, which is highly expressed in senescent chondrocytes, mediates lipid metabolism-related senescence-associated secretory and OA phenotypes in chondrocytes. FASN is a vital factor in the DNL process that negatively impacts the pathogenesis of many diseases. FASN is upregulated in the articular cartilage of older patients, and we demonstrated the role of FASN in ECM and senescence phenotypes regulation in vitro and in vivo. Notably, in this study, FASN was a candidate CircRREB1-binding protein. Herein, we demonstrate that the CircRREB1-FASN axis is a key regulator in age-related OA pathogenesis by mediating lipid metabolism-related senescent phenotypes.

In this study, we have demonstrated the roles of CircRREB1 on aging related OA progression through in vitro and in vivo studies. Due to the deviation of sample size, there is no significant difference between control and OA tissues according to our previous OA related CircRNA deep sequence, but we find that CircRREB1 is upregulated in OA tissue. Hence, we performed CircRREB1 FISH staining with different ages (n = 24) and a linear regression analysis showed a positive correlation between age and CircRREB1 expression. CircRNA expression levels do not have to reflect biological importance, we still need to identify the expressions of CircRREB1 in larger human samples. Furthermore, in this study, we used three P0 generation chondrocytes and three P2 generation chondrocytes to perform CircRNA sequence, which differed from sequencing used in our previous work, and we found that CircRREB1 was significantly upregulated in P2 generation chondrocytes. CircRNAs are a class of non-coding RNA characterized by a closed-loop structure[41]. When exposed to RNA exonuclease, the loop structure in the circRNA enables them to have a longer half-life than other linear RNAs, suggesting their indispensable role in organisms. Even though circRNAs are usually non-coding, increasing evidence has identified their role in regulating different biological processes[42,43]. In our study, we have identified that CircRREBB1 did not involve in RNA encoding. During OA pathogenesis, circRNAs act as sponges for miRNAs and regulate downstream gene expression[44]. We previously reported that CircSERPINE2 acts as a ceRNA to inhibit OA development[26,41]. Moreover, a few circRNAs located in the nucleus can promote gene transcription[45]. CircRNAs also act as scaffolds to form complex formations within proteins[46]. Recently, we described that CircPDE4B acts as a scaffold to promote RIC8A-MID1 binding to prevent OA progression[28]. This study demonstrates significant upregulation of CircRREB1 in senescent chondrocytes and identifies their role in promoting and stabilizing FASN to govern lipid metabolism-related senescence phenotype and age-related OA progression.

Metabolic pathway shifts play a vital role in OA development[9,47]. Increasing evidence has indicated lipid accumulation in chondrocytes as a significant factor affecting OA initiation and progression. Dysregulation of lipid metabolism causes lipid accumulation, accelerating cartilage degradation[48], resulting in the upregulation of cartilage-degrading enzymes. In this study, lipid metabolic analysis indicated that most types of lipids were increased in aging mice and CircRREB1 overexpressed chondrocytes. Hence, an in-depth understanding of the mechanisms governing lipid metabolism in chondrocytes is essential. FASN, ELOVL5, ELOVL6, and SCD1 are the key factors involved in DNL. We found that the expression of FASN, ELOVL5, ELOVL6, and SCD1 was significantly upregulated in chondrocytes from older individuals, suggesting enhanced lipogenesis during aging. CircRREB1 is highly expressed in P2 generation chondrocytes; knockdown of CircRREB1 decreased *FASN, ELOVL5, ELOVL6,* and *SCD1* mRNA expression, suggesting the role of CircRREB1 in mediating aging-related lipogenesis. In CircRreb1 deficiency mice, we observed downregulation of Fasn, Elovl5, Elovl6, and Scd1 in the chondrocytes. Decreased lipid accumulation in chondrocytes was also observed in CircRreb1gKO mice. These findings indicate that the upregulation of CircRREB1 in chondrocytes is a significant factor promoting lipid metabolism-related OA development.

Next, we investigated how CircRREB1 regulates lipid metabolism to accelerate the development of age-related OA. FASN is highly expressed in proliferating cells and acts as a catalyst during the synthesis of palmitate from acetyl-CoA and malonyl-CoA[36]. Inhibition of FASN exerts therapeutic efficacy in many proliferation cells[49,50]. FASN is commonly expressed at low levels in normal cells; however, upregulation of FASN in aging chondrocytes was observed in this study and could be due to the shift in the metabolic patterns of senescent chondrocytes. Overexpression of FASN in chondrocytes drives senescence and OA phenotypes. However, knockdown of FASN in chondrocytes downregulated the cartilage-degrading enzymes, including MMP13 and ADAMTS5, and protected chondrocytes against matrix degradation. Moreover, senescent phenotypes decreased after FASN

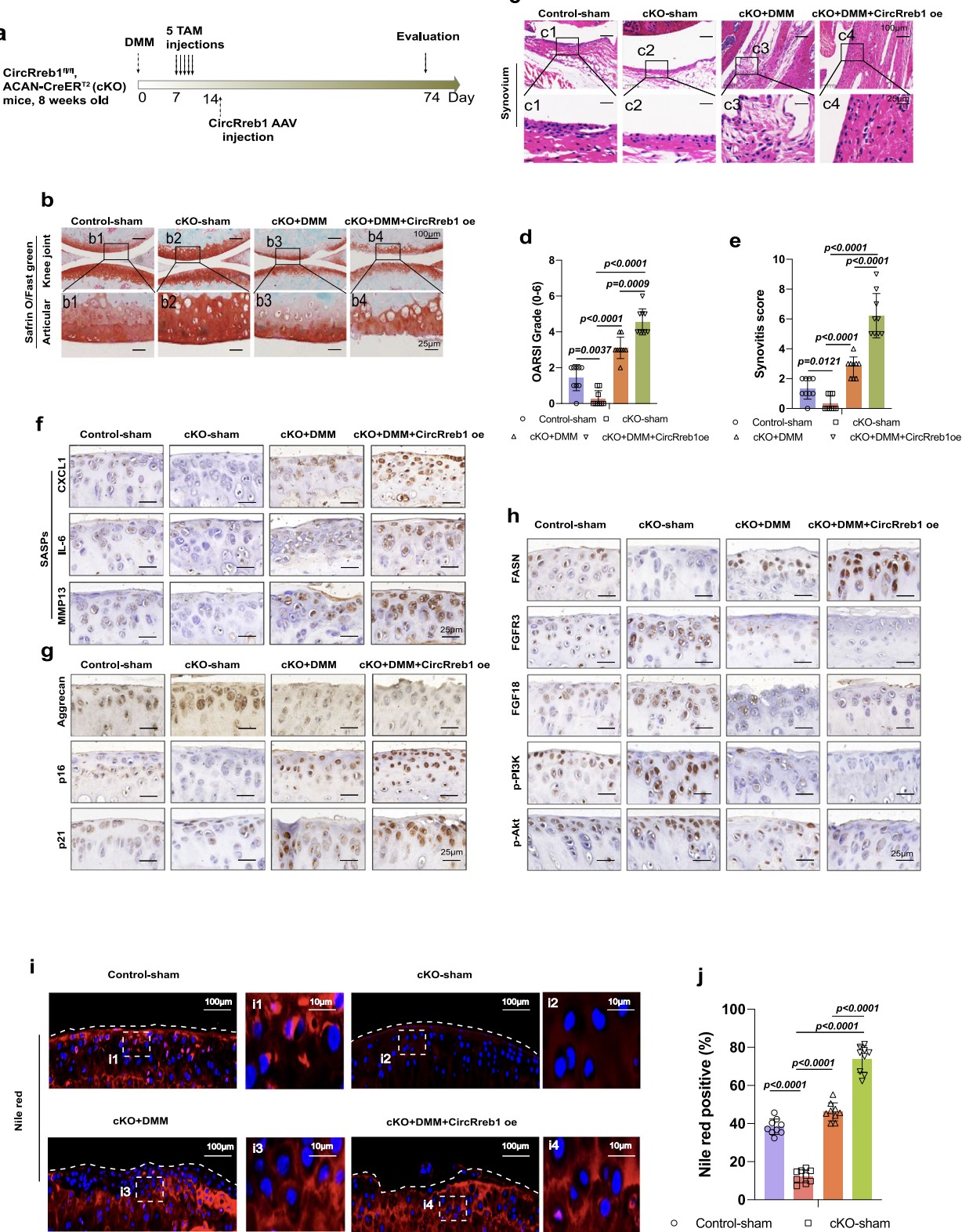

knockdown. These data indicate that FASN may be a critical factor in age-related OA development. It should be noted that FASN is an RNA-binding protein of CircRREB1. This study found that CircRREB1 knockdown affects FASN protein levels instead of *FASN* mRNA levels, suggesting the involvement of other mechanisms (s) in FASN protein regulation. Post-translational modifications (PTMs) are regulators of signal transduction and regulate protein function[51]. PTMs such as

ubiquitylation, acetylation, and succinylation are highly associated with disease development[35,52,53]. Herein, we speculated that CircRREB1 influences FASN stabilization via PTMs. We found that FASN protein level was significantly decreased after TSA treatment instead of NAM through western blot analysis, suggesting FASN acetylation by TSA treatment may promote FASN protein degradation. Moreover, TSA treatment enhanced the ubiquitination level of FASN. This result

**Fig. 7 | Intra-injection of CircRreb1 AAV reverses the effects of CircRreb1 knockout in CircRreb1 cKO mice. a** Schematic illustration of DMM operation and intra-injection of CircRreb1 AAV in 8-week-old CircRreb1 cKO mice. Animal number: control-sham group ($n$ = 9), cKO-sham group ($n$ = 9), cKO+DMM group ($n$ = 9), and cKO+DMM+CircRreb1 oe group ($n$ = 9). **b** SO/FG staining in Control-sham, CircRreb1 cKO, CircRreb1 cKO + DMM, and CircRreb1 cKO + DMM + CircRreb1 AAV mice. Cartilage damage is observed post-DMM operation, which is more prominent after CircRreb1 overexpression. $n$ = 9 mice per group. **c** The degrees of synovial hyperplasia among four groups represented by H&E staining. **d** Cartilage damage indicated by OARSI evaluation ($n$ = 9, biological independent samples). **e** OA severity indicated by synovitis score ($n$ = 9, biological independent samples). **f** SASPs (CXCL1, IL-6, and MMP13) detected in four groups. $n$ = 9 mice per group.

**g** Aggrecan, p16, and p21 expression in cartilage among four groups. $n$ = 9 mice per group. **h** Expression of FASN and its downstream factors (FGFR3, FGF18, p-PI3K, and P-AKT) in chondrocytes among four groups. $n$ = 9 mice per group. **i** Lipid accumulation in chondrocytes among four groups, indicated by Nile red staining. **j** Quantification of Nile red positive cells in four groups ($n$ = 9, biological independent samples). The Mann–Whitney $U$ test is used for (**d**, **e**) (two sided). Quantitative data is shown as mean ± 95% CI. Exact *p-value*s are shown in the figures. One-way analysis of variance (ANOVA) followed by Tukey's HSD test is used for (**j**). Quantitative data are shown as mean ± s.d. Exact *p-value*s are shown in the figures. Scar bar for (**b**, **c**): 100 μm, amplification: 25 μm. Scar bar for (**f**, **g**, and **h**): 25 μm. Scar bar for (**i**): 100 μm, amplification: 10 μm. Source data are provided as a Source Data file.

suggests that acetylation of FASN promotes FASN degradation through ubiquitin-proteasome pathway. CircRREB1 knockdown or overexpression affected the interaction between FASN and E3 ligase MDM2, followed by FASN ubiquitination or deubiquitylation, suggesting the role of CircRREB1 mediated proteasome-ubiquitin pathway in FASN regulation. TSA treatment of chondrocytes caused more FASN ubiquitination and may be explained by the crosstalk between acetylation and ubiquitination. In addition, employing acetyl-deficient (K to R) mutants, K673 was identified as the major acetylation site of FASN, and CircRREB1 overexpression or knockdown did not affect FASN K673R ubiquitination. To find out which specific deacetylase was involved in FASN acetylation, Co-IP assay was performed and HDAC3 was identified that interacted with FASN. Then, HDACs interacted with CircRREB1 were also identified by RIP assay. A complex including CircRREB1, FASN, and HDAC3 was confirmed. We found that CircRREB1 overexpression enhanced the interaction between FASN and HDAC3, and then decreased FASN acetylation, thereby decreasing FASN ubiquitination. However, CircRREB1 knockdown showed the opposite result. These data suggest that CircRREB1 acts as a scaffold to recruit specific deacetylase HDAC3 to affect FASN acetylation.

Recently, small ubiquitin-like modifier (SUMO) protein-conjugated modifications, termed SUMOylation, have been identified in many pathogenesis[54,55]. E1-activating enzymes, E2 ligase UBC9, and E3 SUMO ligases participate in SUMOylation[56]. Unlike ubiquitination, SUMOylation enhances protein stability[56]. Interestingly, RanBP2 is also an RNA-binding protein of CircRREB1 (Supplementary Table 2). A previous study showed that RanBP2 mediates SUMOylation of small heterodimer partners to maintain bile acid homeostasis[39]. Herein, we investigated whether RanBP2 also mediates the SUMOylation of FASN. We found that CircRREB1 overexpression enhanced the interaction between FASN and RanBP2, followed by enhanced RanBP2-mediated SUMOylation of FASN. These data suggested that CircRREB1 enhances FASN stability to prevent proteasomal degradation. In this study, we clarify two mechanisms that together increase the stability of FASN to be involved in lipid metabolism-related OA development.

Fibroblast growth factor signaling is essential for normal cell proliferation, including FGFR3 and FGF18[57]. FGFR3 deficiency causes cartilage damage by enhancing CXCL12-dependent chemotaxis[58], while FGF18 has been reported to attenuate cartilage degradation through the PI3K-AKT signaling pathway[59]. In our study, we found that FGFR3 and FGF18 and its related PI3K-AKT transduction were upregulated due to CircRREB1 knockdown. This result has consisted of mRNA-Seq after CircRREB1 knockdown. FASN protein stability was enhanced by CircRREB1 through FASN PTMs, then, we also confirmed that FGFR3 and FGF18 and its related PI3K-AKT transduction were also upregulated when FASN silenced through RT-qPCR. Then, CircRREB1 was knockdown in the chondrocyte, followed by FASN overexpression. Western blot analysis showed that FASN could reverse the effects mediated by CircRREB1 knockdown (Supplementary Fig. 12a and c, Supplementary Fig. 14i), suggesting FASN is the downstream molecule of CircRREB1. We next investigated whether FASN was required for the

role of CircRREB1 in age-related OA development. The effects of CircRREB1 on age-related OA development have been evaluated when FASN is silenced. The result showed that CircRREB1 overexpression did not change the expression of Col2, Sox9, ADAMTS5, p16, p21, and p53 when FASN silenced (Supplementary Fig. 12b and d), suggesting that FASN mediates the function of CircRREB1 in OA. Therefore, we confirm that CircRREB1-FASN axis regulates FGFR3 and FGF18, and PI3K-AKT signaling transduction. FGF18 was reported to be anti osteoarthritis by promoting PI3K-AKT signaling pathway[59]. Although FGFR3 and its related PI3K-AKT transduction are activated in tumor models, we speculate that the function of PI3K-AKT pathway in chondrocyte is different due its multiple biology. PI3K-AKT-mTOR is a classic pathway, which is associated with autophagy[60]. In this study, we verified whether CircRREB1 affect autophagy in OA chondrocyte. CircRREB1 knockdown did not affect autophagy associated proteins LC3B and p62 indicated by western blot analysis. Moreover, autophagy flux showed no obvious difference in chondrocytes after CircRREB1 knockdown, suggesting CircRREB1 exerts little effect on lysosome activity. These data demonstrate that CircRREB1 regulates senescence and OA phenotypes without affecting autophagy. CircRREB1 and FASN knockdown in OA chondrocyte significantly promoted PI3K and AKT activation, however, mTOR pathway was not activated. In this study, we consider that CircRREB1 knockdown activates PI3K-AKT pathway to regulate another pathway instead of mTOR pathway. PI3K-AKT signal pathway is a classic pathway which is wildly used in different disease. Except promotes mTOR and NF-kB pathways, p-AKT also inhibits CDK inhibitor p21[61], and promotes p-MDM2 to regulate p53 pathway[62]. P21 and p53 are now considered senescence markers. Moreover, the combined effect of PI3K and AKT inhibitors (PI3Ki and AKTi) promoted CDK inhibitors expression, which is strongly associated with chondrocyte senescence. CircRREB1 knockdown decreased p21 expression, however, its expression was reversed by PI3K-AKT inhibition, suggesting that CircRREB1 regulate PI3K-AKT signaling pathway to regulate p21 pathway. Hence, we consider that the CircRREB1-FASN axis regulates the FGFR3 and FGF18-related PI3K-AKT signaling pathways to promote dysregulation of cartilage homeostasis via regulating CDK inhibitors expression.

Cumulatively, we propose a circRNA-mediated mechanism in lipid metabolism-associated senescence regulation. We confirmed that CircRREB1 was highly expressed in P2 generation chondrocytes, verified the enhanced lipogenesis process in senescent progression, and confirmed the CircRREB1-mediated lipogenesis process. CircRREB1 enhances lipogenesis-associated factor FASN stability via two pathways: inhibiting FASN acetylation-mediated ubiquitination and promoting RanBP2 mediated SUMOylation of FASN to prevent its proteasomal degradation. We believe lipid metabolism-targeting CircRREB1 could be developed as a promising therapeutic strategy for age-related OA development. Furthermore, in this study, human cartilages samples were collected and assessed according to age instead of the OA severity. We found that lipid accumulation in human cartilage was upregulated during aging (Fig. 4, Supplementary Fig. 7k). CircRREB1 is associated with aging (Supplementary Fig. 1c) and

CircRreb1 deficiency in mice exhibits lower lipogenesis (Supplementary Fig. 4k and Supplementary Fig. 7j), indicating CircRREB1 mediates lipid metabolism related senescent phenotype during aging. Although aging is highly associated with OA, different severity of OA samples is a limitation of this manuscript, hence, more clinical samples should be investigated to verify the relationship between CircRREB1 and age-related OA.

## Methods

### Mice

All animal experiments were approved by the Institute of Health Sciences Institutional Animal Care and Use Committee of Zhejiang university (Zhejiang, China). All animals were housed in SPF environment under constant temperature (23–25 °C) and humidity (45–65 °C) with a 12-hour light/12-hour dark circadian cycle. They were fed with a standard laboratory diet (Research Diets, D12450K, USA) and had free access to water. CircRreb1 global knockout mice (CircRreb1 gKO) were purchased from Cyagen Biosciences (Suzhou, China). In detail, the gRNA to the mouse Mmu_circ_0000454 gene and Cas9 mRNA were co-injected into fertilized mouse eggs to generate targeted knockout offspring. The absence of CircRreb1 in the mutants was confirmed using routine tail DNA genotyping. The deficiency of CircRreb1 was confirmed by standard PCR genotyping. Genotyping primers were designed to distinguish CircRreb1$^{+/+}$, CircRreb1$^{+/-}$, or CircRreb1$^{-/-}$ alleles: The forward primer was 5′-AAAGTATTTGCCCACACAGGCTTC-3′, and the reverse primer was 5′-CCATGTAATGACAACAGCAACTGA-3′. To observe the senescent and OA phenotypes in mice cartilage, 18-month-old wide type male mice and CircRreb1 gKO male mice were used in this study. CircRreb1$^{fl/fl}$ mice were purchased from Cyagen Biosciences (Suzhou, China). To generate CircRreb1 conditional knockout mice, we bred CircRreb1$^{fl/fl}$ mice with ACAN-CreER$^{T2}$ knock in mice (CircRreb1$^{fl/fl}$, ACAN-CreER$^{T2}$). For inducible deletion of CircRreb1, animals (8-week-old) received 5 times injections of tamoxifen (TAM, Selleck, cat. no. S1238, China) at a dosage 100 mg kg$^{-1}$ body weight. Corn oil injections in mice were used as a control group. Male C57BL/6 J mice (wide type, 8-weeks-old, 3-month-old, and 18-month-old) were purchased from Slac Laboratory Animal Co. Ltd (Shanghai, China). The number of animals in each group was specified in figure legends. For animal euthanasia practice, we used 1% sodium pentobarbital to reduce the pain of animals.

### Experimental OA in mice

3-month and 18-month-old C57BL/6 J male mice, and 8-week-old CircRreb1 cKO male mice were used for experimental OA studies. All animal experiments were approved by the Institute of Health Sciences Institutional Animal Care and Use Committee (Zhejiang, China). For destabilization of the medial meniscus (DMM) operation, animals were anesthetized using 1% sodium pentobarbital. After exposing the knee joint, the medial meniscus was carefully removed under a microscope. Then, the wounds were closed, and all animals received buprenorphine (0.05 mg/kg) and penicillin for 5 days. 10 µL of CircRreb1 or Fasn AAV (approximately $1 \times 10^{12}$ vg/mL) was injected into the knee joint post-DMM operation. Mice were sacrificed 8 weeks post-DMM operation.

### Anesthesia of experimental animals

All experimental animals in this study were anesthetized by intraperitoneal injection of 1% sodium pentobarbital before DMM surgery or sacrifice to reduce the pain. The specific dosage is 0.1 ml/100 g for mice.

### Monitoring of experimental animals

All experimental animal monitoring in this study was conducted by the Department of Laboratory Animal Science of Zhejiang University Health Sciences Institutional Animal Care.

### Human cartilage samples

Human knee joint samples of different ages were collected from patients who had undergone total knee arthroplasty. For histological research, participants (50–65 years old) were included in the younger group ($n = 12$), and participants (70–85 years old) were included in the older group ($n = 12$). All samples were fixed in 4% paraformaldehyde (PFA) and decalcified in 15% EDTA. After decalcification, samples were cut into sections for histological analyses. Human chondrocyte isolated from cartilage samples were also used for western blot, RT-qPCR, immunofluorescence staining, SA-β-Gal staining, and Alcian blue staining. The collection of human cartilage samples was approved by the Ethics Committee of the Sir Run Run Shaw Hospital (Zhejiang, China), and the methods followed the guidelines set by the Declaration of Helsinki. Written informed consent was obtained from participants. Descriptive characteristics of human cartilage samples were listed in Supplementary Table 5.

### Cell culture

Knee cartilage tissues were collected, and it were cut appropriately under sterile conditions and washed thrice with PBS. The cartilage samples were then digested in collagenase (Sigma, cat. no. 1148090, USA) at 37 °C overnight. After digestion, the cell solution was filtered through a 200 µm mesh, followed by centrifugation at 100 g for 5 min. The cell suspension was cultured in high glucose DMEM (Gibco, cat. no. 11965092, Australia) supplemented with 10% FBS (Gibco, cat. no. 10099141 C, Australia), and 1× antibiotic-antimycotic (Sigma, cat. no. 15070063, USA). Chondrocytes were kept in an incubator containing 5% $CO_2$ at 37 °C. After two passages (P2), Chondrocytes were cultured in an incubator containing 5% $CO_2$, and 3% $O_2$ at 37 °C (hypoxia condition). Chondrocytes cultured in normoxia and hypoxia conditions were then used for 4-hydroxynonenal (4-HNE) and p16 staining.

5-day-old C57BL/6 J mice or CircRreb1gKO mice were sacrificed for mouse chondrocyte extraction. Briefly, the hindlimbs were dissected and washed thrice with PBS. Skin and soft tissues were removed to expose the articular tissues. The articular cartilage on the femoral condyle and tibial plateau was then removed by ophthalmic microsurgery and cut into small pieces, followed by digestion in collagenase at 37 °C overnight. The culture conditions of mouse chondrocytes were the same as those of human chondrocytes.

### RNA sequencing

To generate a circRNA profiling database, we performed RNA-seq analyses of ribosomal RNA-depleted total RNA from three P0 generation chondrocytes and three P2 chondrocytes. Knee joint samples from three participants were collected and chondrocytes were isolated and cultured in high glucose DMDM supplemented with 10% FBS, 1× antibiotic-antimycotic in an incubator containing 5% $CO_2$ at 37 °C. P0 generation chondrocytes were collected, at the same culture condition, the remaining were cultured for two passages in monolayer (P2 generation). The information of the three participants was listed in Supplementary Table 5. RNA sequencing was supported by Yuan Shen Technology (Shanghai, China). DESeq2 uses a median of ratios method, which is performed by dividing each raw count value in the sample by that sample's normalization factor to generate normalized count values. The normalization factor is determined by the median value of all ratios for a single sample, which is calculated as the gene counts divided by the geometric mean of each gene. Significant differences were defined as a | log2FC(older/younger) | >1 and FDR ≤ 0.05.

### Cell treatment

Doxorubicin (100 nM, Selleck, cat. no. S1208, China) was used to induce senescence in the chondrocytes. In brief, chondrocytes were treated with doxorubicin for 5 days, and the doxorubicin-added media was refreshed every 2 days. For in vitro experiments, the following reagents were used to treat chondrocytes: bortezomib (250 nM,

Selleck, cat. no. S1013, China), chloroquine (50 μM; Selleck, cat. no. S6999; China), cycloheximide (10 μg/mL; Selleck, cat. no. S7418; China), trichostatin A (5 μM; Selleck, cat. no. S7418, China), and pictilisib (5 nM, Selleck, cat. no. S1065, China), MK-2206 (100 nM, Selleck, cat. no. S1078, China).

### Small interfering RNA transfection
Human and mouse chondrocytes were transfected with specific siRNAs (RiboBio, Guangzhou, China). RNA and protein were extracted 48 and 72 h post-transfection, respectively. Briefly, Lipofectamine RNAiMAX transfection reagent (Thermo Fisher Scientific, cat. no. 113778075, USA) was used for siRNA transfection. All siRNA sequences are listed in Supplementary Table 6.

### Senescence-β-Galactosidase (SA-β-Gal) staining
The SA-β-Gal staining kit (Beyotime, cat. no. C0602, China) was used to evaluate senescent phenotypes in chondrocytes after one-week treatment. In brief, chondrocytes were fixed with 0.2% glutaraldehyde for 15 min at room temperature. The cells were then washed thrice with PBS. SA-β-Gal solutions were incubated with chondrocytes at 37 °C for 12 h. The incubation environment was kept at pH 6.0. An inverted phase contrast microscope (Lecia, Germany) was used to observe SA-β-Gal-positive cells. Total cells and SA-β-Gal-positive cells were counted per culture dish.

### Lysosomal-β-Galactosidase Staining
The Lysosomal-β-Galactosidase Kit (Beyotime, cat. no. C0605, China) was used to detect acid β-Galactosidase as a negative control of SA-β-Gal staining. Chondrocytes after treatment were collected and washed by PBS for three times, followed by fixation with 0.2% glutaraldehyde for 15 min at room temperature. Then, after washing by PBS, cells were incubated with Lysosomal-β-Galactosidase solution at 37 °C for 12 h. An inverted phase contrast microscope (Lecia, Germany) was used to observe Acid-β-Gal-positive cells.

### Reactive oxygen species measurement
Reactive Oxygen Species (ROS) assay kit (Beyotime, cat. no. S0033S, China) was utilized to observe ROS generation in chondrocytes. In brief, chondrocytes after treatment were incubated with DCFH-DA solution (1:1000, 10 μM) at 37 °C for 20 min. Then, ROS generation was acquired by a fluorescence inverted microscope (Leica, Germany). We also measured the intracellular ROS production by using CM-H$_2$DCFDA (Invitrogen™, cat. no. C6827, USA) through a flow cytometer (BD Biosciences, USA). The gating strategy is shown in Supplementary Fig. 16.

### Alcian blue staining
Chondrocytes treated with different stimuli were fixed in 4% PFA (Beytome, cat. no. P0099, China) for 30 min. After washing with PBS three times, chondrocytes were incubated with 1% Alcian blue solution (Sigma, cat. no. A5268, USA) for 1 h. After incubation, the cells were washed three times with PBS. Articular sections were also fixed with 4% PFA, followed by incubation with Alcian blue solution for 1 h. Representative images of Alcian blue staining were obtained using a digital scanner.

### Western blot analysis
Proteins were extracted by lysis in RIPA buffer (Fude Biological Technology Co., Ltd, cat. no. FD009, China) with a mixture of phosphatase inhibitor (Fude Biological Technology Co., Ltd, cat. no. FD1002, China) and protease inhibitor (Fude Biological Technology Co., Ltd, cat. no. FD1001, China) on ice for 30 min. The lysis solutions were then collected and centrifuged at 13400 g for 10 min. The protein solution was collected and quantified using the BCA assay (Beyotime, cat. no. P0010, China). Briefly, protein solutions were separated by sodium

dodecyl sulfate-polyacrylamide gel electrophoresis and transferred to a polyvinylidene fluoride membrane. The membranes were blocked in 10% skim milk for 1 h, followed by incubation with the indicated primary antibody overnight (primary antibodies are listed in Supplementary Table 7). After incubation with the horseradish peroxidase-conjugated anti-rabbit or anti-mouse secondary antibodies (Fude Biological Technology Co., Ltd, car. no. FDM007, and FDR007, China) for 1 h at room temperature, immunoreactive protein bands were detected using an ECL substrate (Fude Biological Technology Co., Ltd, cat. no. FD8020, China) through Chemiluminescent Substrate (Bio-Rad, USA).

### RNA extraction and quantitative real-time PCR
Total cellular RNA was extracted using the AG RNAex Pro reagent (Accurate Biotechnology Co., Ltd, cat. no. AG21101, Changsha, China), and reverse transcription was performed using a PrimeScript TR reagent kit (Accurate Biotechnology Co., Ltd, cat. no. AG11706, Changsha, China). Real-time PCR was performed using an ABI 7500 Sequencing Detection System (Applied Biosystems, Foster City, CA, USA) with UltraSYBR Mixture (Yeason, cat. no. 11204ES, China) to detect the amplification reactions of specific circRNAs or mRNAs. β-Actin was used as the housekeeping gene. The ΔΔCt method was used to normalize the mRNA expression levels. All the primer sequences used in this study are listed in Supplementary Table 8.

### Histology, immunofluorescence, and immunohistochemistry
Human joint samples and mouse articular samples were fixed in 4% PFA. All samples were embedded in paraffin, cut into 5-μm thick sections, and stained with safranin O (Sigma, cat. no. S8884, USA) and Fast Green (Sigma, cat. no. S7258, USA). For immunofluorescence, cells were fixed with 4% PFA for 30 min, followed by incubation with 0.5% Triton X-100 solution (Sigma, cat. no. 9036-19-5, China) at room temperature for 15 min. The cells were blocked with 5% Bovine Serum Albumin (BSA), Fude Biological Technology Co., Ltd, car. no. FD0030, China for 1 h and incubated with primary antibodies (Supplementary Table 7) at 4 °C overnight. After washing thrice with PBS, cells were incubated with Alexa 488-conjugated goat anti-mouse secondary antibody (1:500, Invitrogen, Cat. no. A11001, USA) and Alexa 555-conjugated donkey anti-rabbit secondary antibody (1:500, Beyotime, cat. no. A0453, China) for 1 h at room temperature. Nuclei were stained with 4′,6-diamidino-2-phenylindole (DAPI) (Invitrogen, cat. no. F6057, China). The cells were observed using a fluorescence inverted microscope (Leica, Germany). For immunohistochemistry, sections were incubated with 3% H2O2 (Boster technology, cat. no. SV0004, China) to remove endogenous peroxidase. Antigen retrieval was performed with 0.1% trypsin without EDTA for 2 h at 37 °C. After blocking with 5% BSA for 1 h at room temperature, sections were incubated with primary antibodies (Supplementary Table 7) at 4 °C overnight. The next day, the sections were incubated with peroxidase-conjugated secondary antibodies for 30 min (BOSTER, cat. no. SV0004) and detected using DAB substrate (Beyotime, cat. no. P0202, China).

### Histological assessment
After safranin O/Fast green staining and hematoxylin and eosin (H&E) staining. All sections were evaluated by Osteoarthritis Research Society International (OARSI) system and synovitis score to evaluate the OA degree by two independent individuals. OARSI grading and synovitis score were listed in Supplementary Table 9 and Supplementary Table 10.

### RNA fluorescent in situ hybridization
Cy3-labeled CircRREB1 and CircRreb1 probes were designed and synthesized by RibioBio (Guangzhou, China) to detect CircRREB1 and CircRreb1 expression in chondrocytes and knee joint sections. A RNA

fluorescent in situ hybridization (FISH) kit (RiboBio, Guangzhou, China) was used to detect CircRREB1 or CircRreb1 expression.

## Co-immunoprecipitation

Cells were co-transfected with specific plasmids for 48 h, and then harvested. Cells were lysed with IP lysis buffer (composed of 20 mM Tris-HCl, 150 mM NaCl, 1 mM EDTA, 1% Nonidet P40, a phosphatase inhibitor, and a protease inhibitor cocktail). The supernatant was immunoprecipitated with Sepharose beads (Yeason, cat. no. 36417ES, China) and specific IP antibodies overnight at 4 °C. The antibodies used are listed in Supplementary Table 7. After incubation, the immune complex was washed three times with NaCl buffer for 3 times. The IP proteins were boiled from the immune complex with 1 × SDS loading at 95 °C for 5 min.

## RNA pull down assay

A total of $10^7$ human chondrocytes were collected and lysed in a lysis buffer. The supernatant was collected, 100 μL of which was used as input. The remaining supernatant was divided into two parts and incubated with CircRREB1 specific probe streptavidin Dynabeads mixture or control probe streptavidin Dynabeads mixture at 4 °C overnight using an RNA pull-down kit (BersinBio, cat. no. Bes5102, China). The supernatant interacted with CircRREB1 and was used for Mass Spectrometry identification. The sequences of the CircRREB1 probes are listed in Supplementary Table 6.

## RNA binding protein immunoprecipitation

The RIP assay was performed using a RIP RNA-Binding Protein Immunoprecipitation Kit (BersinBio, cat. no. Bes5101, China). Co-precipitated RNAs were detected using RT-qPCR.

## FASN acetylation site detection

Cells overexpressed FASN were collected and cell protein solutions were separated by sodium dodecyl sulfate-polyacrylamide gel electrophoresis, followed by coomassie blue staining. At first, for in-gel tripic digestion, gel pieces were destained in 50 mM $NH_4HCO_3$ in 50% acetonitrile (v/v) until clear. Gel pieces were dehydrated with 100 μl of 100% acetonitrile for 5 min, the liquid removed, and the gel pieces rehydrated in 10 mM dithiothreitol and incubated at 56 °C for 60 min. Gel pieces were again dehydrated in 100% acetonitrile, the liquid was removed and gel pieces were rehydrated with 55 mM iodoacetamide. Samples were incubated at room temperature, in the dark for 45 min. Gel pieces were washed with 50 mM $NH_4HCO_3$ and dehydrated with 100% acetonitrile. Gel pieces were rehydrated with 10 ng/μl trypsin resuspended in 50 mM $NH_4HCO_3$ on ice for 1 h. Excess liquid was removed and gel pieces were digested with trypsin at 37 °C overnight. Peptides were extracted with 50% acetonitrile/5% formic acid, followed by 100% acetonitrile. Peptides were dried to completion and resuspended in 2% acetonitrile/0.1% formic acid.

Then, UPLC-MS/MS method was performed to detect acetylation sites of FASN. Specifically, the tryptic peptides were dissolved in 0.1% formic acid (solvent A), and directly loaded onto a home-made reversed-phase analytical column (15-cm length, 75 μm i.d.). The gradient was comprised of an increase from 6% to 23% solvent B (0.1% formic acid in 98% acetonitrile) over 16 min, 23% to 35% in 8 min, and climbing to 80% in 3 min then holding at 80% for the last 3 min, all at a constant flow rate of 400 nl/min on an EASY-nLC 1000 UPLC system. The peptides were subjected to NSI source followed by tandem mass spectrometry (MS/MS) in Q ExactiveTM Plus (Thermo) coupled online to the UPLC. The electrospray voltage applied was 2.0 kV. The m/z scan range was 350 to 1800 for a full scan and intact peptides were detected in the Orbitrap at a resolution of 70,000. Peptides were then selected for MS/MS using NCE setting 28 and the fragments were detected in the Orbitrap at a resolution of 17,500. A data-dependent procedure that alternated between one MS scan

followed by 20 MS/MS scans with 15.0 s dynamic exclusion. Automatic gain control (AGC) was set at 5E4.

The resulting MS/MS data were processed by using Proteome Discoverer 2.4. The modification set is the acetylation of lysine. Carbamidomethyl on Cys was specified as a fixed modification and oxidation on Met was specified as a variable modification. Peptide confidence was set at high, and peptide ion score was set >20.

## CircRREB1 and FASN overexpression in vitro

Ad-CircRREB1 adenovirus was purchased from HanBio (Shanghai, China). Briefly, human chondrocytes were cultured and infected with the 400 MOIs of control adenovirus and 100, 200, and 400 MOI of CircRREB1 adenovirus for 2 h, and cultured for another 24 h. CRISPR/Cas9 synergistic activation mediator (SAM) was used to overexpress FASN in human chondrocytes. LV-FASN-sgRNA was purchased from GeneChem (Shanghai, China). Human chondrocytes were infected with the three LV-FASN-sgRNAs.

## CircRREB1 and FASN overexpression in vivo

CircRreb1 AAVs were purchased from HanBio (Shanghai, China) and Fasn AAV was purchased from Genechem (Shanghai, China). Briefly, after DMM operation, animals received an articular injection of 10 μL of CircRreb1 and Fasn AAV (approximately $1 \times 10^{12}$ vg/mL) in knee joints.

## Plasmids

For the co-IP assay, human FASN cDNA was synthesized and subcloned into the Plvx vector with a Flag tag, and human MDM2 cDNA was synthesized and subcloned into Plvx vector with a Myc or HA tag. Ubiquitin cDNA was synthesized and subcloned into the Plvx vector with an HA tag. For the RIP assay, cDNAs of CircRREB1 fragments were synthesized and subcloned into the Plvx vector. Fragments of FASN were synthesized and subcloned into the plvx vector with a Flag tag.

## Nile red staining

Cells and tissue sections were fixed with 4% PFA, followed by 1 μM Nile red solution (MedChemExpress, cat. no. HY-D0718; China) for 10 min at room temperature. Nuclei were stained with DAPI. Images were obtained using a fluorescence microscope (Leica, Germany).

## Metabolomics

For overall metabolomics, P0 generation chondrocytes and P2 generation chondrocytes were sent to Metware Biotechnology Co., Ltd (Wuhan, China) for LC–MS/MS analysis ($n = 8$). The sample extracts were analyzed by ultra-performance liquid chromatography (UPLC) and tandem mass spectrometry (MS/MS). For targeted lipomics, we collected cartilage tissues of 3-month-old mice ($n = 4$) and 18- month-old mice ($n = 4$), cartilage tissues of wide type mice ($n = 4$), and CircRreb1 gKO mice ($n = 4$), human chondrocyte transfected with control adenovirus ($n = 4$) and CircRREB1 adenovirus ($n = 4$). All sample preparation and metabolic profiling were performed with standard procedures in cooperation with Core Facility of Instrument of Hangzhou Cosmos Wisdom Biotechnology Co., Ltd. In detail, for cell sample preparation and extraction, the sample stored at −80 °C refrigerator was thawed on ice. The thawed sample was homogenized by a grinder (30 HZ) for 20 s. A 400 μL solution (Methanol: Water = 4:1, V/V) containing internal standard was added into 40 mg of the sample and vortexed for 3 min. The sample was placed in liquid nitrogen for 5 min and on dry ice for 5 min, and then thawed on ice and vortexed for 2 min. This freeze-thaw circle was repeated three times in total. The sample was centrifuged at 13400 g for 10 min (4 °C). A 300 μL of supernatant was collected and placed at −20 °C for 30 min. The sample was then centrifuged at 12000 rpm for 3 min (4 °C). A 200 μL aliquots of supernatant were transferred for LC-MS analysis. For Tissue sample preparation and extraction, the sample stored at −80 °C refrigerator

was thawed on ice. A 50 μL solution (Methanol: Water = 7:3, V/V) containing internal standard was added to the sample and centrifuged at 12000 rpm for 3 s (4 °C). The thawed sample was homogenized by a grinder (30 HZ) for 20 s. A 150 μL solution (Methanol: Water = 7:3, V/V) containing internal standard was added to the grinded sample, and shaked at 500 g for 5 min. After placing on ice for 15 min, the sample was centrifuged at 13400 g for 10 min (4 °C). A 150 μL of supernatant was collected and placed in -20 °C for 30 min. The sample was then centrifuged at 13400 g for 3 min (4 °C). A 120 μL aliquots of supernatant were transferred for LC-MS analysis. Then, the samples were analyzed using Waters ACQUITY Ultra-Performance LC (UPLC) system coupled with a Waters XEVO TQ-S mass spectrometry controlled by MassLynx 4.1 software (Waters, Milford, MA). 40,000 FWHM. Electrospray Ionization (ESI). 1.9Kv(ESI−); 2.8 Kv(ESI+). ACQUITY UPLC BEH Amide 1.7 μM analytical column (2.1 × 100 mm). The gradient was started at a flow rate of 0.6 mL/min, with the injected volume being 1 μL and the column temperature at 45 °C. Mobile phases were A = ACN/water (95/5, 5 mM NH4FA), B = ACN/water (50/50, 5 mM NH4FA), respectively. The lipids were eluted using the following gradients: 0–2 min (0.1–20% B), 2–5 min (20–80% B), 5–5.1 min (80-0.1% B), 5.1–8 min (0.1%B). The raw data file generated by UPLC-TQMS will be processed by MassLynx software (Waters, Milford, MA, USA), and peak extraction, integration, and quantification of each lipid will be performed. The powerful R studio package is used for subsequent statistical analysis. All metabolites were listed at Supplementary Data 1, Supplementary Data 2, Supplementary Data 3. and Supplementary Data 4.

### Statistical analysis

Positive cells in images were analyzed by Image J software (version v. 1.51a) and Flow cytometer data were analyzed by BD FACSDiva (version 9.0). Statistical analysis was performed using GraphPad Prism software (version 8.0). Data from two groups were analyzed by using an unpaired, two-tailed Student's $t$ test for RT-qPCR and the percentages of positive cells. One-way analysis of variance (ANOVA) followed by Tukey's HSD test is used for multigroup comparisons. For nonparametric data, the Mann-Whitney U test was used for OARSI grade, Synovitis score, and osteophytes number. Linear regression analysis was used for CircRREB1 expression in human cartilage samples with aging. Parametric data were presented as mean ± SD and the calculated 95% confidence intervals (CIs) for nonparametric data. Statistical significance was set at $P < 0.05$.

### Reporting summary

Further information on research design is available in the Nature Portfolio Reporting Summary linked to this article.

## Data availability

The authors declare that the data supporting the findings of this study are available with the paper and its Supplementary information files. A reporting summary for this article is available as Supplementary Information file. CircRREB1 was recorded in Circbank database (http://www.circbank.cn). The Circular RNA deep sequencing data generated in this study has been deposited in the GEO database under accession code GSE236856. The mRNA sequencing in human chondrocyte with or without CircRREB1 knockdown has been deposited in the GEO database under accession code GSE237298. The mRNA sequencing in human chondrocyte with or without FASN knockdown has been deposited in the GEO database under accession code GSE237011. Detailed metabolic substances identified by metabolomics in different models can be found in Supplementary data 1, Supplementary data 2, Supplementary data 3, and Supplementary data 4. All raw metabolomic data included in this study are available upon in an unrestricted manner from the corresponding author. Source data are provided with this paper.

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

## Acknowledgements

This work was supported by Major joint project of National Nature Science Fund of China (No. U22A20282), National Key R&D Program of China (No. 2020YFC1107104), the Key research and development plan in zhejiang province (No. 2018C03060、2020C03041); National Nature Science Fund of China (81871797 and 81874045) and Major projects jointly built by Zhejiang Provincial Health Commission (WKJ-ZJ-2006). We would also like to thank Editage (www.editage.com) for English language editing.

## Author contributions

S.W.F., X.Q.F., and S.Y.S. designed this study. Z.G., J.J.Z., J.X.C. performed the majority of the experiments and wrote the paper. F.F., S.H.Y., and H.T.Z. participated in experiments and revised this paper. Z.Y.Z helped preform animal surgery. C.X.S. and K.Y.L. analyzed data and edited figures. Z.G., J.J.Z., J.X.C., and F.F. contributed equally to this work.

## Competing interests

The authors declare no competing interests.
