## [Peer Review File · Nature Communications]

CircRREB1 mediates lipid metabolism-related senescence-associated secretory phenotypes in aging chondrocytes via FASN post-translational modificationsREVIEWER COMMENTS

Reviewer #1 (Remarks to the Author):

This study examines the role of a circRNA in lipid metabolism related cellular senescence and osteoarthritis.

CircRREB1 was found by deep sequencing to be upregulated in aging chondrocytes, and its role in mouse models of aging-related and surgically induced OA was studied. Detailed mechanistic analyses revealed interactions with FASN and PI3k pathway.

The study is novel as it is the first to address CircRreb1 in the context of OA.

Most experimental approaches are suitable, but some have limitations and clarifications and changes in data interpretation are needed.

Suggestions for revision:

Authors refer to 'aging' chondrocytes, but the model is simply for culturing cells for two passages in monolayer. It is not appropriate to refer to this as a model of aging. It is only an effect of monolayer culture. Throughout the text this needs to be changed.

In Fig 1, nearly 100% of chondrocytes were SAbGal positive. Why were cells additionally treated with doxorubicin in Fig 2?

Fig 2C shows reduction of MMP and ADAMTS expression by si CircRREB1. What were the basal levels of these RNAs? Were OA chondrocytes used here?

Would increased expression of these genes, for example by IL-1 also be suppressed by si CircRREB1?

Fig 2d: were the cells treated with doxorubicin?

Fig 3: the intraarticular injection of CircRreb1specific adeno-associated virus (AAV) is missing information about the fold increase of CircRreb1 in the joint tissues. If this is in large excess of the increases that are seen in the aged mice, the results may not be physiologically relevant.

The same concern applies to Extended Data Fig. 6. Intra-articular infection of FASN AAV.

Presumably, the cartilage samples were collected from patients undergoing knee arthroplasty. What type of cartilage (fibrillated or intact surface was analyzed?). Given the heterogeneity of the human OA population, the very small data variability especially in the 'younger group' in Fig 4f is surprising.

Fig 4j-m: indicate age of the mice.

Lines 108-109 and 111-112 are redundant

Line 387 indicate how FASN was overexpressed.

The mRNAseq experiment is a strength as it adds an independent approach to investigate CircRreb1 mechanisms. The main finding that is further investigated is about PI3K-AKT signal Transduction and FGF3 and FGF18. Here it would be important to determine whether CircRreb1 changes mRNA levels of FGF3 and FGF18 directly (independent from FASN).

The discussion should also address whether or how the CircRreb1-regulation of FASN protein stability is also responsible for the changes in PI3K pathway or whether this is an independent mechanisms of CircRreb1.

Methods: for the Ad transduction experiments in vitro and AAV experiments in vivo, the virus titers need to be indicated and the fold differences in the transfected gene need to be shown. Also explain how the virus titers were selected.

Methods for AAV are completely missing.

Reviewer #2 (Remarks to the Author):

In this manuscript circRREB1 is identified to be highly differently regulates in chondrocytes exposed to hypoxia compared to normoxia. circRREB1 were subsequently verified to be higher expressed in cells treated with doxo, an agent used to induce cellular senescence, and in mouse and human cartilage related to ageing or OA, respectively. The cellular and molecular role of circRREB1 is characterized by knockdown and transfection studies, and identified to be related to lipid metabolism.

This group has previously identified differential regulated circRNA in samples from OA patients and normal controls (ref 29); however, circRREB1 is not listed in their findings in this article nor in a similar study based on patients' samples by Lin et al. 2017 to be among the highest expressed circRNAs. Expression levels do not have to reflect biological importance, but there is a lack of discussion related to this lack of identification in larger human patient material.

Not many papers describe the biological role of circRREB1, however, Ji et al. (J Mol Cellul Cardioogy 2020) identified circRNA from Rreb1 to be reduced upon doxo treatment of cells (myocardocytes), which is an opposite effect compared to the effect of doxo on chondrocytes, although myocardocytes and chondrocytes both originates from mesenchymal stem cells.

Line 161-162. Supplementary Table 1 list the 15 top up-and down regulated circRNAs, not 20 as indicated in the text.

It is claimed that analyses of samples from 3 donors, each replicated 3 times gives n=9 (ex Fig 1 c). Multiples measurements of each of the 3 samples in the two groups cannot contribute to n, these are pseudoreplications.

Hypoxia conditions are not described in method part.

Scale bars lacking on in all figures with images of cells.

In the extended data Fig 1 circRREB1 positive cells are quantified and presented in % of circRREB1 positive cells in normo (P0) versus hypoxia (O2), the same calculations are done in doxo versus untreated cells and in tissues of young versus old. I assume that the correct calculation and text related to the y-axis should be '% of total number of cells' not in '% of circRREB1 positive cells'? Presenting circRREB1 positive cells in % of circRREB1 cells will give 100% ..

The rationale for presenting a-d in extended data fig 2 is not clear. The effect of doxo on

chondrocytes are well described by others, also in the concentration chosen in this study. The same goes for the figures in extended data 3 presenting the effect of doxo on murine chondrocytes.

Reviewer #3 (Remarks to the Author):

Major

1. This study utilizes SA- β -gal staining to detect senescent cells. Positive staining is caused by the upregulation of β -galactosidase activity in lysosomes. As the enzymatic activity of lysosomes is regulated by the autophagy pathway, SA- β -gal staining in cultured chondrocytes might represent an increase in autophagy rather than a senescent state. This study should incorporate one or more additional biomarkers of senescence, and careful consideration should be given to how the interventions being applied affects autophagy and the lysosomal activity of cells. In addition, the pH values, which play roles in the results of SA- β -gal staining, should be described in the Method section.

2. This study performs RNA-seq analyses of ribosomal RNA-depleted total RNA from three P0 generation chondrocytes and three P2 chondrocytes. Where are these chondrocytes from? What kinds of culture conditions are these cells received? Please give a detailed description.

3. The human knee joints from younger (50–65y) and older adults (70–85y) but the mice joints from 8w, 3m, and 18m are evaluated. Why are these time points chose?

4. The authors only provide the age of the participants. How about the gender, severity of OA, BMI, and other characteristics related to OA pathogenesis?

5. How about the molecular detection of synovial hyperplasia?

6. The authors show the SO/FG staining of femur and tibia, but only one of them displays the results of the molecular detection. Why?

7. The mean OARSI score of 3-month-old mice in DMM group is about 3, while the mean score of 8-week-old mice in DMM group is about 4 which is similar to that of 18-month-old mice in DMM group. Why?

8. As FASN itself can aggravate age-related OA development, what is the authors understanding on “FASN overexpression reverses the effects induced by circRREB1 knockdown”?

9. The effects of circRREB1 on age-related OA development should be evaluated when FASN is silenced.

Minor

1. The results of figures should be matched with manuscript.

2. How old are the gKO mice used in this study, 18w? Please describe in the Method section.

3. Please add the group name in figure 2I.
4. Please check the magnifications and add the scale bars.

Reviewer #4 (Remarks to the Author):

Gong et al. found that circRNA CircRREB1 expression is upregulated in aging chondrocytes, which in turn affects FASN expression. These events exacerbate senescence phenotypes and osteoarthritis progression. They claimed that CircRREB1 interacts with and stabilizes FASN by inhibiting the proteasomal degradation of FASN. They showed that overexpression of either CircRREB1 or FASN is sufficient to cause senescent phenotypes in chondrocytes while their inhibitions abolish all the senescence-associated changes in chondrocytes. Knockout of CircRREB1 ameliorates OA in various mouse models.

Major concerns:

Although the authors claim that CircRREB1 influences FASN stabilization via two types of PTMs, the underlying mechanisms still remain as a "speculation" as they mentioned in the discussion (line 667). It appears that authors would like to suggest that CircRREB1 serves as a scaffold to recruit both FASN and PTM regulators (what specific deacetylase?). They need to provide more robust biochemical evidence to fully reveal the precise molecular function of CircRREB1.

They did not sufficiently show how mechanistically FASN-mediated lipid metabolism contributes to causing senescence of chondrocytes. This would be the key message that readers from senescence/OA fields would like to know from this paper with the most interest.

Meanwhile, there is a very high chance that investigators from the senescence field would not be fully convinced by their claims about CircRREB1- or FASN-mediated senescence. For instance, they showed that overexpression of either CircRREB1 or FASN is sufficient to cause senescent phenotypes in chondrocytes within "24 h" (as described in figures legends). However, generally speaking, mammalian cells rarely make a commitment to enter senescence within that short time window even after they are exposed to strong DNA-damaging agents. For human cells, it is not even unusual to take more than a week to develop senescence-associated secretory phenotypes (SASP). Therefore, what they observed in 24 h after CircRREB1 or FASN overexpression is unlikely to be a "senescence" response.

Another critical reason why this reviewer does not think what they observed is senescence-associated phenomena is that unlike the general notion that mTOR inhibition suppresses SASP factor expression in senescent cells (Judith Campisi and colleagues, Nature Cell Biology, 2015 volume 17), they report that the combined treatments of PI3K and AKT inhibitors increased SASP factor expression.

Specific comments

1. More description on metabolomics data (in Fig. 4a-d) such as alterations in each lipid type. Metabolite measurement is required to support the significance of FASN in aging and OA progression. Do the lipid levels alter in aged human or mice and CircRREB1 overexpression system? What are the implications of these lipids in OA?
2. How is circRREB1 upregulated in aging chondrocyte in the first place? The upstream regulatory mechanism for circRREB1 production in senescent chondrocytes remains unclear. In Fig. 1f and Extended Fig. 1, the expression of circRREB1 is upregulated in senescent or aged chondrocytes. How are the expressions of the host gene RREB1 in these

conditions?

- 3.** Introduction includes redundant description (for instance, line 108-111 and line 111-115). In line 106-108, is there any previously identified association between FASN and aging-associated diseases other than cancer?
- 4.** In line 161 and Supplementary Table 1, the result of RNA-seq summarized in the table described only 15 up- and down-regulated circRNAs, not 20. For the table, a statistical measurement such as FDR to test the significance of differential expression is missing. What does mean value indicate in the table? Read count?
- 5.** Description of histological assessment method of human and mouse OA is missing, especially OARSI grading and synovitis scoring. How was the thickness of articular cartilage measured?
- 6.** More detailed description of the origin of human cartilage samples is required since it lacks consistency throughout the manuscript. While online methods read "participants (50– 60 years old) were included into the younger group and participants (70–89 years old) were included into the older group", other texts in Result and figures read "younger (50– 65y) and older adults (70–85y)".
- 7.** Related to Fig. 4e, human cartilage shows distinct zonal structure and chondrocytes in each zone exhibits different characteristics. The authors should have discussed the zonal origin of cartilage tissue and section method for the cartilage samples, and compared the protein level of fatty acid metabolism-related proteins within similar regions.
- 8.** Related to Fig. 1c,d, the result of RNA-seq should be addressed in detail. How was total 3800 circRNAs were identified using CIRI2 and CIRC explorer? In the previous article published by the same author (Shen et al., *Ann. Rheum. Dis.* 2019), the author identified 12,738 circRNAs using three OA and control human cartilage samples and the same circRNA explorers. Where did the discrepancy come from? How was the expression of circRREB1 in the previous RNA-seq data?
- 9.** Related to the screening of circRREB1 (hsa_circ_0001573) in Fig. 1d, the rationale for choosing hsa_circ_0001573 for further study among differentially expressed circRNAs is lacking the justification. For instance, in line 171-173, why was the top downregulated hsa_circ_0000211 excluded for further study?
- 10.** Related to Fig. 4l and m, do CircRreb1 gKO mice show reduced synthesis of specific lipids in system level?
- 11.** For Fig. 5d and Extended Fig. 4f,g, binding site prediction of circRREB1 with FASN is not clearly explained. How does the CatRAPID prediction result match with that from RIP assay?
- 12.** Some of the statistical methods they used are inappropriate or inaccurately described. Especially, statistical analysis of nonparametric data such as histological grades should not have been conducted using 'two-side unpaired t-test' throughout the manuscript. The authors described that they used one-way ANOVA with LSD-t by assuming equal variance for multiple group comparisons. However, it says one-way ANOVA with Turkey's multiple comparison, which should be at least 'Tukey's HSD' although LSD and HSD use different significant difference. Overall, this reviewer cannot figure out the statistical method they used.
- 13.** Related to Fig. 6a, the rational for selecting MDM2 as a candidate is insufficiently provided.
- 14.** Related to mass spectrometry analysis of FASN in line 470-473, the method is completely missing and the explanation for MS data is not enough. Was the UPLC-MS/MS method performed focusing on detecting acetylation sites of FASN? Were K673 and K1065 the only acetylation sites on FASN?
- 15.** In Fig. 6k, the author could not find the result of Flag-FASN MS.
- 16.** Related to Extended Fig. 9-10, there are many attempts to reduce OA and SASP expression by inhibiting PI3K-Akt pathway (Wang et al., *Am. J. Transl. Res.* 2022) and the inhibition of this pathway is also known to promote autophagy (Xue et al., *Biomed. Pharmacother.* 2017). PI3K-Akt is generally known to activate mTOR and NF- κ B pathway,

which are considered as a pro-SASP transcription factors (Herranz et al., Nat. Cell. Bio. 2015; Salminen et al., Cell Signal. 2012).

17. In Fig. 7h, the quality of IHC against FGFR3, FGF18, p-PI3K, and p-Akt are low (background levels are very different between samples), making it difficult to compare signals between cKO+DMM and cKO+DMM+CircRreb1 oe groups.

Reviewer #1 (Remarks to the Author):

This study examines the role of a circRNA in lipid metabolism related cellular senescence and osteoarthritis.

CircRREB1 was found by deep sequencing to be upregulated in aging chondrocytes, and its role in mouse models of aging-related and surgically induced OA was studied. Detailed mechanistic analyses revealed interactions with FASN and PI3k pathway.

The study is novel as it is the first to address CircRreb1 in the context of OA.

Most experimental approaches are suitable, but some have limitations and clarifications and changes in data interpretation are needed.

Response: Thank you for your efforts to review our manuscript. We appreciate your helpful comments to improve and correct this manuscript. We are very happy to edit our manuscript according to your constructive suggestions. We have done some experiments carefully to support these concerns and then we revised this manuscript with all changes highlighted in red. Thanks again for your help.

Suggestions for revision:

Authors refer to 'aging' chondrocytes, but the model is simply for culturing cells for two passages in monolayer. It is not appropriate to refer to this as a model of aging. It is only an effect of monolayer culture. Throughout the text this needs to be changed.

Response: Thanks for your constructive suggestions, we are very happy to further edit our manuscript according to your comments. We have changed aging chondrocytes into P2 generation chondrocyte in the revised manuscript (please line 51, line 57, line 141, line 150, line 233, line 239, line 242, line 765, line 870).

In Fig 1, nearly 100% of chondrocytes were SA-b-Gal positive. Why were cells additionally treated with doxorubicin in Fig 2?

Response: Thanks for your comment for our manuscript. We cultured primary chondrocyte and subjected them to two passages (P2). At the station of P2, we found that most chondrocytes were highly expressed SA- β -Gal staining, suggesting P2 generation chondrocytes were at senescence station. A reference published in Science translational medicine by professor Jin-Hong Kim (Stress-activated miR-204 governs senescent phenotypes of chondrocytes to promote osteoarthritis development PMID: 30944169)

confirmed that doxorubicin induced chondrocyte senescence. Only senescent cells expressed SA-J3-Gal and Doxo treatment significantly increased SA-J3-Gal positive staining. In this study, Kim and et al. used different model to investigate the senescence phenotypes. Hence, in our study, we would like to investigate the effects of CircRREB1 in different models. In Figure 2f, we used Doxo to induce senescence of human chondrocyte, followed by CircRREB1 knockdown to observe SA-J3-Gal staining, immunofluorescence staining of MMP13, p16, Sox9, and Alcian blue staining. In Figure 2a-d, for molecular detection, P2 generation OA chondrocytes were used. We have added this information in the figure legend.

Fig 2C shows reduction of MMP and ADAMTS expression by si CircRREB1. What were the basal levels of these RNAs? Were OA chondrocytes used here?

Would increased expression of these genes, for example by IL-1 also be suppressed by si CircRREB1?

Response: Thanks for your constructive suggestion. We used OA chondrocytes to investigate the role of CircRREB1 on MMP13 and ADAMTS5 expression. OA chondrocytes transfected with NC, CircRREB1#1, CircRREB1#2, CircRREB1#3 SiRNAs respectively. The CT value and power value of one sample are listed as follow:

CT value	Si-NC			Si-CircRREB1#1			Si-CircRREB1#2			Si-CircRREB1#3		
MMP13	24.589	24.713	24.801	27.213	27.198	27.205	26.158	26.133	26.133	26.840	26.869	26.838
ADAMTS5	23.342	23.205	23.408	24.496	24.500	24.558	24.107	24.131	24.138	24.063	24.106	24.068
actin	17.134	17.663	17.105	17.538	17.696	17.335	16.918	16.228	16.499	17.577	17.050	17.231

Power value	Si-NC			Si-CircRREB1#1			Si-CircRREB1#2			Si-CircRREB1#3		
MMP13	1.0807398	0.9917283	0.9330097	0.2044036	0.2066525	0.2055458	0.2161749	0.2198766	0.21998	0.2246762	0.220205	0.2249879
ADAMTS5	0.983906	1.081851	0.9394614	0.5156367	0.5140717	0.4940149	0.3434536	0.33794	0.3362816	0.5906786	0.573333	0.588635

We have further performed an experiment to explore the role of CircRREB1 knockdown after IL-113 stimulation. Method: Human chondrocytes were seeded into 6-well plate and cultured with DMEM high glucose media containing IL-113 (10 ng/ml) for 48h, followed by CircRREB1 knockdown. Cells were collected and RNAs were extracted. Next, RT-qPCR was performed. β -actin (*ACTB*) rRNA was chosen as a housekeeping gene. Here, we listed the CT value and power value of MMP13 and ADAMTS5 of one sample as follows:

CT value	con			IL-1 β			IL-1 β +Si-CircRREB1#1			IL-1 β +Si-CircRREB1#2		
MMP13	24.319	24.494	24.433	19.019	19.299	18.944	21.502	21.340	21.095	21.354	20.785	21.206
ADAMTS5	25.477	25.873	25.631	22.837	22.831	22.775	24.135	24.115	24.569	24.274	24.367	24.198
actin	15.921	15.543	15.454	15.346	15.127	15.752	15.320	14.800	14.792	14.977	15.171	14.747

Power value	con			IL-1 β			IL-1 β +Si-CircRREB1#1			IL-1 β +Si-CircRREB1#2		
MMP13	1.0687373	0.9472467	0.9877931	35.872373	29.560247	37.802569	4.7389598	5.3012853	6.2853183	5.2303031	7.7596834	5.7936012
ADAMTS5	1.1355653	0.8631547	1.0202327	6.0316024	6.0545241	6.2951979	1.8103643	1.8358647	1.3405859	1.637974	1.5357065	1.7268463

The result of RT-qPCR was showed as follows. We found that IL-1 β simulation significantly increased *MMP13* and *ADAMTS5* mRNA expression, however, these expressions were decreased after CircRREB1 knockdown.

Fig 2d: were the cells treated with doxorubicin?

Response: In Figure 2d, we utilized P2 generation OA chondrocytes and then performed CircRREB1 knockdown (n=3) to detect the molecular level of ECM degrading enzymes, ECM components, and senescence associated markers.

Fig 3: the intraarticular injection of CircRreb1specific adeno-associated virus (AAV) is missing information about the fold increase of CircRreb1 in the joint tissues. If this is in large excess of the increases that are seen in the aged mice, the results may not be physiologically relevant.

The same concern applies to Extended Data Fig. 6. Intra-articular infection of FASN AAV.

Response: Thanks for your helpful suggestion. We performed CircRreb1 FISH staining in aging mice. Immunofluorescence pictures taken in the same batch of experiments under the same conditions were selected. We selected CircRreb1 immunofluorescence intensity to evaluate the fold increase of CircRreb1 in the joint tissues. The CircRreb1 immunofluorescence intensity was calculated by $\text{IntDen (correction)} = \text{IntDen (cell)} - \text{Area (cell)} * \text{Mean (background)}$ through Image J software. We showed the relative fluorescence intensity as follows. The fold increase of CircRreb1 in 3M mice and 18M mice were showed in Extended Data Fig 5a-d. The result indicated that fold change of CircRreb1 after CircRreb1 AAV infection are not in large excess of the increases that are seen in the aged mice (please see line 322- 327).

Then, we also performed FASN fluorescence staining. The fold increase of FASN expression in knee joints after Fasn overexpression was showed in Extended Data Fig. 5

e-h. The result also showed that fold change of Fasn after Fasn overexpression are not in large excess of the increases that are seen in the aged mice (please see line 448-454).

Extended Data Fig. 5 a, CircRreb1 FISH staining in sham, DMM, DMM+CircRreb1 group between 3-month-old mice and 18-month-old mice. b, Fluorescence intensity of CircRreb1 in 3-month-old mice and 18-month-old mice. c, Fluorescence intensity of CircRreb1 in sham, DMM, DMM+CircRreb1 group in 3-month-old mice. d, Fluorescence intensity of CircRreb1 in sham, DMM, DMM+CircRreb1 group in 18-month-old mice. e, Representative immunofluorescence images of FASN in sham, DMM, DMM+vector, and DMM+Fasn group. f, Representative immunofluorescence images of FASN in 3-month-old mice and 18-month-old mice. g, Fluorescence intensity of FASN in 3-month-old mice and 18-month-old mice. h, Fluorescence intensity of FASN in sham, DMM, DMM+vector, and DMM+Fasn group. Two-tailed Student's t test is used for b and g. One-way analysis of variance (ANOVA) followed by a post hoc test is used for c, d, and h. Quantitative data shown as mean \pm s.d. Exact p values are shown in figures.

Presumably, the cartilage samples were collected from patients undergoing knee arthroplasty. What type of cartilage (fibrillated or intact surface was analyzed?). Given the heterogeneity of the human OA population, the very small data variability especially in the 'younger group' in Fig 4f is surprising.

Response: Thanks for your constructive suggestion. In human normal cartilage, the

superficial zone was defined as the first 10% of the tissue thickness, the translational zone as the next 10% and the radial zone as the remaining 80% of the tissue thickness (PMID:12127837). Hence, according to the distinct zonal structure of cartilage, we would like to explore the superficial zone (articular cartilage, AC) and the radial zone (calcified cartilage zone, CCZ) of tibia plateau according to this reference. According to your suggestion and another reviewer's comments, we showed the FASN, ELOVL5, ELOVL6, and SCD1 expression in articular cartilage (AC) and calcified cartilage zone (CCZ) respectively between younger individuals and older individuals (Fig. 4e). Higher expression of FASN, ELOVL5, ELOVL6, and SCD1 were observed in older samples, suggesting more de novo lipogenesis happened in joint tissues in older individuals (please see line 366371).

Fig 4j-m: indicate age of the mice.

Response: Thanks for your efforts to our manuscript. The animals used in Fig 4j-m were 18 months old. We have described it in the revised manuscript (please see 379-380).

Lines 108-109 and 111-112 are redundant

Response: Thanks for your helpful suggestion. We have re-edit this part in the revised manuscript (please see line 127-130).

Line 387 indicate how FASN was overexpressed.

Response: After evaluating the functions of Fasn in vitro, we investigated the impact of Fasn on OA and senescent phenotypes in an in vivo study. Animals underwent DMM operation, followed by Fasn AAV (approximately 1×10^{12} vg/mL). injections two weeks post-injury (please see line 448-454, line 1107-1110).

The mRNA seq experiment is a strength as it adds an independent approach to investigate CircRreb1 mechanisms. The main finding that is further investigated is about PI3K-AKT signal Transduction and FGF3 and FGF18. Here it would be important to determine whether CircRreb1 changes mRNA levels of FGF3 and FGF18 directly (independent from FASN).

Response: Thanks for your constructive suggestion. We performed knockdown and gain of function experiment. We knockdown and overexpress CircRREB1 in HCs, then, RNA was extracted. RT-qPCR result confirmed that CircRREB1 knockdown upregulated FGFR3 and FGF18 mRNA expression, while CircRREB1 overexpression downregulated these expressions (Extended Data Fig. 14f). The result of RT-qPCR was showed as follows:

The address of is also

discussion should also whether or how the CircRreb1 -regulation FASN protein stability responsible for the changes in PI3K pathway or whether an independent

this is mechanism of CircRreb1.

Response: Thanks for your suggestion. It is our pleasure to further edit our manuscript according to your advice. In our study, we found that FGFR3 and FGF18 and its related PI3K-AKT transduction were upregulated due to CircRREB1 knockdown. This result is consisted with mRNA-Seq after CircRREB1 knockdown. FASN protein stability was enhanced by CircRREB1 through FASN PTMs, then, we also confirmed that FGFR3 and FGF18 and its related PI3K-AKT transduction were also upregulated when FASN silenced through RT-qPCR. Then, CircRREB1 was knockdown in chondrocyte, followed by FASN overexpression. Western blot analysis showed that FASN could reverse the effects mediated by CircRREB1 knockdown (Extended Data Fig.12a and c, Extended Data Fig. 14i), suggesting FASN is the downstream molecule of CircRREB1. We next investigated whether FASN was required for the role of CircRREB1 in age-related OA development. The effects of CircRREB1 on age-related OA development has been evaluated when FASN is silenced. The result showed that CircRREB1 overexpression did not change the expression of Col2, Sox9, ADAMTS5, p16, p21, and p53 when FASN silenced (Extended Data Fig. 12b and d), suggesting that FASN mediates the function of CircRREB1 in OA. Therefore, we confirm that CircRREB1-FASN axis regulates FGFR3 and FGF18 and PI3K-AKT signaling transduction. We have discussed in the discussion part. (please see line 827-843).

Methods: for the Ad transduction experiments in vitro and AAV experiments in vivo, the virus titers need to be indicated and the fold differences in the transfected gene need to be shown.

Also explain how the virus titers were selected.

Response: We apologize for the poor demonstration of these transduction experiments. We have explained in the revised manuscript. For CircRREB1 adenovirus transduction, briefly, human chondrocytes were cultured and infected with the 400 MOIs of control adenovirus and 100, 200, and 400 MOI of CircRREB1 adenovirus respectively for 2h, and cultured for another 24h. Then, chondrocytes were collected and total RNAs were extracted. Quantitative RT-PCR (qRT-PCR) analysis further confirmed that CircRREB1 was significantly increased in 400 MOI group compared to control group (please see line 1099-1102). The result was showed in Fig 2g. Furthermore, we observed that the shape and state of chondrocytes when exposed to 400 MOI adenoviruses are still good, hence, we selected 400 MOI of CircRREB1 adenovirus as the best virus titer. 400 MOI of CircRREB1 adenovirus was also used for the gain of function experiments.

For CircRREB1 and Fasn AAV infection, after DMM operation, animals received articular injection of 10 μ L of CircRreb1 or Fasn AAV (approximately 1×10^{12} vg/mL). After 2 months post AAV injection, knee joint articular were collected under microscope (please see line 1108-1111). To show the fold change of CircRreb1 and Fasn expression in vivo, we performed FISH staining and immunofluorescence staining of Fasn, and calculated the CircRreb1 or Fasn immunofluorescence intensity. The final fluorescence intensity was calculated by $\text{IntDen (correction)} = \text{IntDen (cell)} - \text{Area (cell)} * \text{Mean (background)}$ through Image J software. We selected sham group as the control and showed the relative CircRreb1 fluorescence intensity after CircRreb1 infection in 3-month-old mice and 18 month-old mice. Meanwhile, we also selected sham group as the control after Fasn overexpression. The fold change of CircRREB1 and Fasn in vivo were showed as follow.

Methods for AAV are completely missing.

Response: Thanks for your comment to our manuscript. We are sorry that we worked on the manuscript for a long time and the repeated addition and removal of sentences. We are very sorry that AAV infection is missing. Herein, we have put it into method part.

CircRreb1 and Fasn AAVs were purchased from HanBio (Shanghai, China) and Genechem (Shanghai, China) respectively. Briefly, after DMM operation, animals received articular injection of 10 μ L of CircRreb1 or Fasn AAV (approximately 1×10^{12} vg/mL) in knee joints (please see line 1107-1110).

Reviewer #2 (Remarks to the Author):

In this manuscript circRREB1 is identified to be highly differently regulates in chondrocytes exposed to hypoxia compared to normoxia. circRREB1 were subsequently verified to be higher expressed in cells treated with doxo, an agent used to induce cellular senescence, and in mouse and human cartilage related to ageing or OA, respectively. The cellular and molecular role of circRREB1 is characterized by knockdown and transfection studies, and identified to be related to lipid metabolism.

Response: Thank you very much for your time and effort that you have put into reviewing our manuscript submitted in Nature Communications. We appreciated that you give us this chance to better edit our manuscript according to your suggestions and comments. Based on the comments, we carefully revised the manuscript with all changes highlighted in red. This group has previously identified differential regulated circRNA in samples from OA patients and normal controls (ref 29); however, circRREB1 is not listed in their findings in this article nor in a similar study based on patients' samples by Lin et al. 2017 to be among the highest expressed circRNAs. Expression levels do not have to reflect biological importance, but there is a lack of discussion related to this lack of identification in larger human patient material.

Response: Thanks for your comment to our manuscript. Your advice is beneficial to improve the quality and rationality of our study. We previously used 3 paired clinical OA and control tissues to perform CircRNA deep sequencing. There were no obvious difference of age, height, weight, and BMI between 3 paired clinical OA and control tissues. The difference between two groups is whether is OA. The information of 3 paired clinical OA and control tissues showed as follow:

3 paired clinical OA tissues for circRNA deep sequencing				
Gender	Age(year)	Height(cm)	Weight(kg)	BMI
Male	61	170	72	24.9135
Male	61	168	58	20.5499
Female	60	160	65	25.3906
3 paired control tissues for circRNA deep sequencing				
Gender	Age(year)	Height(cm)	Weight(kg)	BMI

Male	60	169	78	27.3099
Male	62	170	62	21.4532
Female	62	160	60	23.4375

We found out the CircRREB1 expression in this CircRNA deep sequencing. The CircRREB1 expression value in control tissue is 30.1, 29.1, and 21.3 (fpkm value). The expression in OA tissue is 37.9, 26.6, and 8.3 (fpkm value). Due to the deviation of sample size, there is no significant difference between control and OA tissues, but we find that CircRREB1 is upregulated in OA tissue and we need investigate the expression of CircRREB1 in larger human patient material. Furthermore, in this manuscript, we used three P0 generation chondrocytes and three P2 generation chondrocytes to perform CircRNA deep sequencing. Hence, the samples used for CircRNA sequencing is differ from sequencing used in our previous work. We have discussed in the discussion part (please see line 727-738).

To further observe the relationship between CircRREB1 and age, we used sections of knee joints (Medial tibial plateau) from different age who received total knee joints replacement to perform CircRREB1 FISH staining. We collected 24 samples: 50-54y(n=4), 55-60y(n=3), 61-64y (n=4), 65-70y (n=3), 71-74y (n=3), 75-80y (n=4), 81-85y (n=3). CircRREB1 was labeled by Cy3. We showed the representative images of CircRREB1 FISH staining. We observed that CircRREB1 increased with aging. Next, we quantified the relative CircRREB1 intensity among these samples (intensity of CircRREB1 in 50y sample was used as control). The CircRREB1 immunofluorescence intensity was calculated by $\text{IntDen (correction)} = \text{IntDen (cell)} - \text{Area (cell)} * \text{Mean (background)}$ through Image J software. The Reletive CircRREB1 intensity was showed as follow. Linear regression analysis between age and CircRREB1 staining was provided. Linear regression analysis showed a positive correlation between age and CircRREB1. But we still need identify the expression of CircRREB1 in larger human patient material. We have discussed in the discussion part (please see line 727-738).

FISH: CircRREB1(Cy3)

Not many papers describe the biological role of circRREB1, however, Ji et al. (J Mol Cellul Cardiology 2020) identified circRNA from Rreb1 to be reduced upon doxo treatment of cells (myocardocytes), which is an opposite effect compared to the effect of doxo on chondrocytes, although myocardocytes and chondrocytes both originates from mesenchymal stem cells.

Response: Thanks for your careful review. We used Doxorubicin to induce chondrocyte senescence according to a previous study by professor Jin-Hong Kim (Stress-activated miR-204 governs senescent phenotypes of chondrocytes to promote osteoarthritis development PMID: 30944169). Doxorubicin could significantly promote p16, p21 and other senescence phenotypes in primary human or mouse chondrocytes. Although myocardocytes and chondrocytes both originates from mesenchymal stem cells, we verified many times and confirmed that CircRREB1 was upregulated after doxo stimulation in human chondrocytes indicated by RT-qPCR (n=3, 3 donors for three replicates). Hence,

we suggested that the effect of Doxo is different in chondrocytes and myocardiocytes.

Line 161-162. Supplementary Table 1 list the 15 top up-and down regulated circRNAs, not 20 as indicated in the text.

Response: We are very sorry for this description by mistake. We have changed this sentence in the revised manuscript (please see line 176-177). Thanks again for your careful review.

It is claimed that analyses of samples from 3 donors, each replicated 3 times gives $n=9$ (ex Fig 1 c). Multiples measurements of each of the 3 samples in the two groups cannot contribute to n , these are pseudoreplications.

Response: We are very sorry for this description by mistake. We have changed this description in figure legends in the revised manuscript. Thanks again for your careful review (please see line 1375, line 1381 line 1383, line 1394, line1397, line1403, line 1441, line 1455, line 1458, line 1461, line 1539 line 1540, line 1542 line 1543-1545, line 1561, line 1565, line 1567, line 1631-1632, line1637-1638, line 1640, line 1671, line 1680-1682, line 1692, line 1694-1695, line 1708).

Hypoxia conditions are not described in method part.

Response: We are sorry for this missing part. We have added it into the method part in the revised manuscript. Chondrocytes were cultured in an incubator containing 5% CO₂, and

3% O₂ at 37°C (hypoxia condition). Please see line 921-924.

Scale bars lacking on in all figures with images of cells.

Response: We are thanks for your careful review. We are very pleasure to further edit our manuscript according to your advice. We have added scale bars in the revised manuscript. In the extended data Fig 1 circRREB1 positive cells are quantified and presented in % of circRREB1 positive cells in normo (P0) versus hypoxia (O2), the same calculations are done in doxo versus untreated cells and in tissues of young versus old. I assume that the correct calculation and text related to the y-axis should be '% of total number of cells' not in '% of circRREB1 positive cells'? Presenting circRREB1 positive cells in % of circRREB1 cells will give 100% ..

Response: Thanks for your constructive suggestion. We are very agreed to your concerns. We have changed '% of circRREB1 cells' into '% of total number of cells'. Please see new extended data 1d-h. Thanks again for your help.

The rationale for presenting a-d in extended data fig 2 is not clear. The effect of doxo on chondrocytes are well described by others, also in the concentration chosen in this study. The same goes for the figures in extended data 3 presenting the effect of doxo on murine chondrocytes.

Response: Thanks for your suggestions. we used Doxo to induce chondrocytes senescence according to a study by professor Jin-Hong Kim (Stress-activated miR-204 governs senescent phenotypes of chondrocytes to promote osteoarthritis development PMID: 30944169). We noticed that the results of Doxo stimulation in chondrocytes have been studied. Hence, we removed this part in the extended data Fig. 2 and extended data Fig. 3.

Reviewer #3 (Remarks to the Author):

Major

1. This study utilizes SA- β -gal staining to detect senescent cells. Positive staining is caused by the upregulation of β -galactosidase activity in lysosomes. As the enzymatic activity of lysosomes is regulated by the autophagy pathway, SA- β -gal staining in cultured chondrocytes might represent an increase in autophagy rather than a senescent state. This study should incorporate one or more additional biomarkers of senescence, and careful consideration should be given to how the interventions being applied effects autophagy and the lysosomal activity of cells.

In addition, the pH values, which play roles in the results of SA- β -gal staining, should be described in the Method section.

Response: Thanks for your constructive suggestion. According to your suggestion, we performed some experiments to explore the roles of CircRREB1 on autophagy and the lysosomal activity (please see line 256-273). We collected and cultured human chondrocytes from OA individuals. Then, CircRREB1 was knockdown. Western blot analysis showed that CircRREB1 knockdown did not affect autophagy associated proteins such as p62 and LC3 (Extended Data Fig. 3c, d, and e). Although CircRREB1 shows no influence on autophagy pathway, we also explore the role of CircRREB1 on lysosomal activity by using chloroquine (Cq). After NC or CircRREB1 SiRNA transfection, chondrocytes were treated with or without Cq (5uM) for another 2 days. Western blot showed the LC3 and p62 expression with or without Cq stimulation. Autophagy flux index was used to evaluate the impact of CircRREB1 on lysosomal activity. Autophagy flux= $(LC3B-II + CQ/\beta\text{-actin})/(LC3B-II-CQ/\beta\text{-actin})$. The result of Autophagy flux showed no difference between Si-NC and Si-CircRREB1 group (Extended Data Fig. 3f and g). Furthermore, we used another senescence biomarkers such as p21 and γ -H2AX nuclear foci to support the anti-senescence role of CircRREB1 knockdown. CircRREB1 knockdown in chondrocytes decreased p21 and γ -H2AX nuclear foci expression (Extended Data Fig. 3a). Meanwhile, we have described the pH of SA- β -gal staining in the method part (please see 963).

We thank for this concerns to our manuscript, we also would like to explain that the kit used in this study is Senescence 13-Galactosidase Staining Kit. Senescent cells usually become larger and express high 13-Galactosidase enzyme activity at pH6.0, which suggests that senescent cells can be marked by this kit. Specific SA-13-gal staining kit has been widely used by many previous study (PMID: 28043053, 29941930, and 30944169). We have also noticed Lysosomal 13-Galactosidase Staining Kit, which is used for acid 13-Galactosidase detection in lysosomal. We have also noticed that most normal cells express a high level of lysosome 13- Galactosidase activity level, which can be a negative control of Senescence 13-Galactosidase Staining. Hence, we performed experiment again to detect acid 13-Galactosidase after CircRREB1 knockdown in chondrocytes. The result of Lysosomal 13-Galactosidase Staining confirmed that no obvious difference was showed between Si-NC and Si-CircRREB1 group. All these results were showed as follows.

Extended Data Fig. 3 a, Immunofluorescence images of p21 and gamma-H2AX nuclear foci expression after CircRREB1 knockdown. b, Acid β-Galactosidase expression after

CircRREB1 knockdown. c, LC3B and p62 proteins expression after CircRREB1 knockdown. d, Quantification of LC3 II. e, Quantification of p62. f, Chondrocytes transfected with NC or CircRREB1 SiRNA treated with or with CQ. LC3B and p62 expression are examined. g, Quantification of autophagy flux. Autophagy flux= (LC3B-II + CQ/13-actin)/(LC3B-II-CQ/13-actin). Two-tailed Student's t test used for statistical analysis. Quantitative data shown as mean \pm s.d. Exact p values are shown in figures.

2. This study performs RNA-seq analyses of ribosomal RNA-depleted total RNA from three P0 generation chondrocytes and three P2 chondrocytes. Where are these chondrocytes from? What kinds of culture conditions are these cells received? Please give a detailed description.

Response: The statement of CircRNA deep sequence has been supplemented in the revised manuscript, we will be happy to edit the text further based on helpful comments from the reviewers. We collected knee joints samples from three participants. Chondrocytes from lateral plateau were isolated and cultured in Dulbecco's modified Eagle's medium (DMEM) containing 4.5 g/L d-glucose, L-glutamine, and 110 mg/L sodium pyruvate (Gibco, cat. no. 11965092), supplemented with 1% penicillin/streptomycin (Thermo Fisher, cat. no. 15140122), and 10% fetal bovine serum (FBS) (Gibco, cat. no. 10099141C). P0 generation chondrocytes were collected and the remain were subjected to two passage (P2 generation). Herein, we successfully collected three P0 generation chondrocytes and three P2 generation chondrocytes. All chondrocytes were incubated in an incubator containing 5% CO₂ at 37°C (please see line 933-942, line 915-924).

3. The human knee joints from younger (50–65y) and older adults (70–85y) but the mice joints from 8w, 3m, and 18m are evaluated. Why are these time points chose?

Response: The incidence rate of osteoarthritis gradually increases among people over 50 years old. We collected human knee joints samples from participants with different age. The topic in this study is aging associated CircRNA, then, we would like to distinguish samples by age and classified them into two groups according to our previous study (PMID: 33408787). Aging mice were used in this study. CircRREB1 is highly associated with aging identified by our CircRNA sequence, hence, we would like to investigate the role CircRREB1 in young mice and old mice. Previous study has reported the age of young mice (3month) and old mice (18month) (PMID: 35501923, 34480023).

In order to investigate the role of CircRREB1 on senescence and OA progression in a DMM model, 8-week-old mice were used because these were adult mice, which also

according to a previous study (<https://doi.org/10.1038/s43587-021-00165-w>, published in Nature Aging) and our previous work (PMID:34039624).

4. The authors only provide the age of the participants. How about the gender, severity of OA, BMI, and other characteristics related to OA pathogenesis?

Reponses: We are sorry for the missing information you referred. Hence, we have added a table of these information in the Supplementary table S5.

5. How about the molecular detection of synovial hyperplasia?

Response: Thanks for your helpful suggestion. To detect the molecular level of synovial hyperplasia, we selected two markers Vimentin and F4/80. Vimentin is the marker for fibroblast-like synoviocytes (FLS) and F4/80 is the marker for macrophage. These two labels can be the candidates for the synovial hyperplasia. We then performed immunofluorescence staining of Vimentin and F4/80 in Fasn oe model, aging mice model and cKO mice model (please see line 333-334, line 460-462, line 694-696). The result has been listed as follow.

Extended Data Fig. 6 a, Representative immunofluorescence images F4/80 and Vimentin in Fasn AAV based in vivo model. b, c, Representative immunofluorescence images F4/80 and Vimentin in aging mice model. d, Representative immunofluorescence images F4/80 and Vimentin in CircRreb1 cKO mice model.

6. The authors show the SO/FG staining of femur and tibia, but only one of them displays the results of the molecular detection. Why?

Response: Thanks for your constructive suggestion. We showed the SO/FG staining of femur and tibia, hence, according to your suggestion, we showed the molecular detection (Aggrecan, CXCL1, and p16) of femur and tibia respectively. Please see the revised Fig. 3. Thanks again for your support.

7. The mean OARSI score of 3-month-old mice in DMM group is about 3, while the mean score of 8-week-old mice in DMM group is about 4 which is similar to that of 18-month-old mice in DMM group. Why?

Response: Thanks for your constructive comments. Your suggestions are very important to improve our manuscript. There may be some certain errors between different individuals for this score evaluation, therefore, we further invited another two individuals who were blinded to our experiment to perform OARSI evaluation again in cKO mice and Fasn oe model (Fig. 7d, Extended Data Fig. 10 a).

8. As FASN itself can aggravate age-related OA development, what is the authors understanding on “FASN overexpression reverses the effects induced by circRREB1 knockdown”?

Response: We are sorry for the poor explanation of this sentence. Here, we would like to explain the aim of these experiments. In this study, we confirmed that CircRREB1 caused senescence by binding FASN through regulating FASN PTMs. We next investigate whether FASN is the downstream molecule of CircRREB1. In brief, CircRREB1 was knockdown in chondrocytes, followed by FASN overexpression. RT-qPCR and western blot analysis indicated that matrix degrading enzymes like MMP13 and ADAMTS5 and senescence associated markers p16, p21, and p53 were increased, while Col2 and Sox9 were decreased after FASN overexpression, suggesting FASN was the downstream molecule of CircRREB1. We next investigated whether FASN was required for the role of CircRREB1 in age-related OA development. The effects of CircRREB1 on age-related OA development were evaluated when FASN was silenced. The result showed that CircRREB1 overexpression did not change the expression of Col2, Sox9, ADAMTS5, p16, p21, and p53 when FASN silenced (Extended Data Fig. 12b and d), suggesting that FASN mediates the function of CircRREB1 in OA. Above all, the role of the CircRREB1-FASN axis in promoting age-related OA progression was confirmed (please see line 500-513). 19. The effects of circRREB1 on age-related OA development should be evaluated when FASN is silenced.

Response: Thanks for your suggestion for our manuscript. We are very happy to further edit our manuscript according to your advice. FASN combined with CircRREB1 which was identified by RNA pulldown and RIP assay. Hence, we suggested that the progress of chondrocyte senescence and osteoarthritis was mediated by CircRREB1-FASN axis. We investigated role of CircRREB1 on ECM and senescence phenotypes when FASN silenced through western blot analysis and RT-qPCR. FASN silencing in chondrocyte could protect chondrocyte and decrease senescence markers, indicated by upregulation of Col2, Sox9 expression, downregulation of ECM degrading enzymes MMP13 and ADAMTS5, and

senescence markers p16 and p21. However, when we overexpressed CircRREB1, these effects could not be changed (Extended Data Fig. 12b and d), suggesting that FASN mediates the function of CircRREB1 in age-related OA (please see line 500-513).

Extended Data Fig. 12 **a**, Col2, Sox9, MMP13, ADAMTS5, p16, p21, and p53 mRNA expression in HCs with CircRREB1 knockdown followed by FASN overexpression (n=3, 3 donors for three replicates). **b**, Col2, Sox9, ADAMTS5, p16, p21, and p53 mRNA

expression in HCs with FASN knockdown followed by CircRREB1 overexpression (n=3, 3 donors for three replicates). **c**, FASN, p16, p21, p53, Col2, Sox9, ADAMTS5, and MMP13 expression in HCs with CircRREB1 knockdown followed by FASN overexpression. **d**, FASN, p16, p21, p53, Col2, Sox9, and ADAMTS5 expression in HCs with FASN knockdown followed by CircRREB1 overexpression. One-way analysis of variance (ANOVA) followed by a post hoc test is used for statistical analysis. Quantitative data shown as mean \pm s.d. Exact p values are shown in figures.

Minor

1. The results of figures should be matched with manuscript.

Response: Thanks for your suggestion. We have edited and corrected figures and words in the manuscript.

2. How old are the gKO mice used in this study, 18w? Please describe in the Method section.

Response: The gKO and WT mice used in this study were 18-month-year old. We have added this information into the method and result part (please see line 896-897).

3. Please add the group name in figure 2I.

Response: Thanks for your suggestion. We have added the name in the Fig. 2I

4. Please check the magnifications and add the scale bars.

Response: Thanks for your suggestion. All scale bars have added.

Reviewer #4 (Remarks to the Author):

Gong et al. found that circRNA CircRREB1 expression is upregulated in aging chondrocytes, which in turn affects FASN expression. These events exacerbate senescence phenotypes and osteoarthritis progression. They claimed that CircRREB1 interacts with and stabilizes FASN by inhibiting the proteasomal degradation of FASN. They showed that overexpression of either CircRREB1 or FASN is sufficient to cause senescent phenotypes in chondrocytes while their inhibitions abolish all the senescence-associated changes in chondrocytes. Knockout of CircRREB1 ameliorates OA in various mouse models.

Response: We appreciated for your efforts put into reviewing our manuscript. Thanks for your constructive suggestions to improve the quality of our manuscript. We are very pleasure to further edit this manuscript according to your suggestions and comments. Based on the comments, we carefully revised the manuscript with all changes highlighted in red. Thanks again for giving us this chance to revise our study.

Major concerns:

Although the authors claim that CircRREB1 influences FASN stabilization via two types of PTMs, the underlying mechanisms still remain as a “speculation” as they mentioned in the discussion (line 667). It appears that authors would like to suggest that CircRREB1 serves as a scaffold to recruit both FASN and PTM regulators (what specific deacetylase?). They need to provide more robust biochemical evidence to fully reveal the precise molecular function of CircRREB1.

Response: Thanks for your constructive suggestions to improve the quality of our manuscript. According to your suggestion, we have performed some experiments to explain your concerns. The results relevant to this point have been showed as follow.

First of all, we would like to explore how acetylation affect FASN protein level. Herein, we treated human chondrocytes with nicotinamide (NAM), an inhibitor of the SIRT family deacetylases, and trichostatin A (TSA), an inhibitor of histone deacetylase (HDAC) I and II respectively. We found that FASN protein level was significantly decreased after TSA treatment instead of NAM through western blot analysis and FASN protein quantification (Fig. 6a). This result suggested that acetylation by TSA may promote FASN protein degradation.

Secondly, chondrocytes were co-transfected with Flag-FASN and HA-UB plasmids and then treated with or without TSA and NAM. The acetylation level of FASN was increased by TSA (Fig. 6b), but not by NAM (Fig. 6c). Moreover, TSA treatment enhanced ubiquitination of FASN (Fig. 6b), thereby promoted FASN degradation. This result suggested that acetylation promoted FASN degradation through the ubiquitin-proteasome pathway.

Thirdly, we would like to find out which specific deacetylase involved in FASN acetylation because the acetylation level of FASN was affected by an HDAC family deacetylase. In detail, we co-transfected Flag-FASN with different HDAC family deacetylase. Co-IP result showed that HDAC3 combined with FASN (Fig. 6d). Herein, we confirmed that HDAC3 is the specific deacetylase involved in FASN acetylation.

Next, we performed molecular docking to visualize the combination of CircRREB1, FASN, and HDAC3. We would like to explain that CircRREB1 serves as a scaffold to recruit HDAC3 to stabilize FASN protein level (Fig. 6h). In Fig. 6, part (b) showed the interaction sites between CircRREB1 and FASN, (c) and (d) showed the interaction sites between CircRREB1 and HDAC3. We found the K673 site on FASN, which is also the acetylation site on FASN detected by UPLC-MS/MS method (Fig. 6e). To confirm this conclusion, we also performed some experiments. We have confirmed that the interaction between CircRREB1 and FASN through RNA pulldown assay and RNA immunoprecipitation (RIP) assay. We also performed RIP assay to confirmed the interaction between CircRREB1 and HDAC3 (Fig. 6f). RIP result showed that CircRREB1 also interacted with HDAC3 (Fig. 6f). To further investigate the relationship between CircRREB1, FASN, and HDAC3, CircRREB1 knockdown and overexpression were performed respectively in chondrocyte and we found that CircRREB1 overexpression enhanced the interaction between FASN and HDAC3, and then decreased FASN acetylation, thereby decreased FASN ubiquitination (Fig. 6g). However, CircRREB1 knockdown showed the opposite result (Fig. 6g).

Overall, we performed some molecular experiments to elucidate that CircRREB1 recruited FASN and HDAC3 to decrease the acetylation of FASN, thereby reduced proteasome degradation of FASN.

Fig. 6. CircRREB1 mediates PTMs of FASN to maintain its function. a, Chondrocytes treated with TSA or NAM, FASN protein expression is examined by western blot analysis. b, c, TSA treatment, but not NAM, increases FASN acetylation level and ubiquitination

level. d, Co-IP assay is performed and HDAC3 is confirmed as a specific deacetylase involved in FASN acetylation. e, UPLC-MS/MS is performed to find out acetylation site on FASN. f, HDAC3 interacts with CircRREB1 confirmed by RIP assay. g, CircRREB1 knockdown or overexpression affects FASN acetylation, ubiquitination level and interaction between FASN and HDAC3. h, Molecular docking of CircRREB1 complex (CircRREB1, HDAC3, and FASN). i, Prediction of a potential interaction between E3 ligase and FASN. j, Effect of STUB1 and MDM2 overexpression on FASN protein level. k, The interaction between FASN and MDM2 confirmed by co-IP assay. l, FASN co-localized with MDM2 in chondrocyte cytoplasm. m, The effects of the K673 and K1065 mutants on FASN acetylation and ubiquitination, K673 is a primary acetylation site on FASN. n, Conservation of the K673 site on FASN. o, The effect of CircRREB1 overexpression on the interaction between K673R FASN and HDAC3. p, The effect of CircRREB1 and overexpression on K673R FASN acetylation and ubiquitination levels, detected by IP assay. q, Schematic illustration of the intersection between CircRREB1 pulldown MS and Flag-FASN MS. r-t, FASN protein levels in HCs infected with RanBP2 siRNAs and a negative control after treatment with CHX for 2, 4, and 8 h. u,v, The effects of CircRREB1 knockdown and overexpression on FASN SUMOylation, indicated by IP assay. One-way analysis of variance (ANOVA) followed by a post hoc test is used for a. Two-tailed Student's t test is used for f. Quantitative data shown as mean \pm s.d. Exact p-values are shown in figures. They did not sufficiently show how mechanistically FASN-mediated lipid metabolism contributes to causing senescence of chondrocytes. This would be the key message that readers from senescence/OA fields would like to know from this paper with the most interest.

Response: Thanks for your constructive suggestions for our manuscript. It is our pleasure to perform more experimental evidence according to your suggestions. Herein, we are ready to explain how mechanistically FASN mediated chondrocyte senescence (please line 471-499).

First of all, we identified that FASN knockdown significantly decreased FASN protein expression and FASN mediated palmitic acid production. In addition, we also observed that lipid accumulation in chondrocytes was significantly decreased after FASN silencing indicated by Nile Red staining. A lot of studies have focused on the relationship between lipid metabolism and osteoarthritis. For some examples, PPAR α -ACOT12 axis mediated de novo lipogenesis in chondrocyte caused OA progression (PMID:34987154). Another

study showed that lipid accumulation in chondrocyte mediated by PGAM1 overexpression also caused osteoarthritis (PMID: 30143643). Hence, we would like to explore FASN mediated lipid metabolism in aging-related osteoarthritis is of great significance. Lipid peroxidation can generate a lot of ROS, thereby causing cellular senescence and osteoarthritis. Herein, we investigated whether FASN mediated lipid metabolism caused ROS environment to induce chondrocyte senescence. We performed flow cytometry assay and found that the total ROS levels were increased in TBHP-induced chondrocyte, while decreased followed by FASN knockdown in chondrocyte. FITC-ROS staining were strong in TBHP-induced groups, while significantly decreased after FASN knockdown. Moreover, fewer amount of 4-hydroxynonenal (4-HNE), a lipid peroxidation product, was observed in FASN knockdown after TBHP stimulation.

For the next step, we performed mRNA-seq (Majorbio Bio-pharm Technology Co., Ltd, Shanghai, China) to detect the downstream genes after FASN knockdown. KEGG analysis showed that FASN knockdown was highly associated with cellular senescence. This result is also consisted with our preliminary research basis. Then, we listed cellular senescence associated genes which is differentially expressed between Si-NC groups and Si-FASN groups. Next, we would like to explore whether FASN silencing also regulates cellular senescence through this pathway in TBHP-induced chondrocytes. We found that cellular senescence associated genes and oxidative response genes (RASSF5, MAPK3, IL6, CDK2, NFKB1, CXCL8, ZFP36L2, MAPK14, NRAS, RBL2, and LIN9) were upregulated after TBHP stimulation, however, FASN silencing could reduce these genes expression indicated by RT-qPCR. This result showed that ROS environment induced by FASN mediated lipid metabolism is associated with chondrocyte senescence.

At last, we explored the senescent phenotypes in TBHP-induced chondrocytes when FASN silencing. Chondrocytes treated with TBHP and FASN SiRNAs for one week. Representative immunofluorescence staining showed that FASN knockdown reduced senescent markers gamma-H2AX nuclear foci, p21, and p16, as well as SA- β -gal staining. Senescence associated markers p16 and p21 were reduced after FASN knockdown under ROS environment. In conclusion, we demonstrated that FASN mediated lipid metabolism caused chondrocyte senescence by generating ROS environment.

Extended Data Fig. 11 FASN caused lipid metabolism mediates cellular senescence by regulating ROS microenvironment. **a**, FASN protein expression and palmitic acid production after FASN knockdown. **b**, lipid accumulation in chondrocyte indicated by Nile Red staining after FASN knockdown. **c**, Flow cytometry of CM-H₂DCFDA. **d**, Quantification of FITC ROS level (n=3). **e**, ROS expression in chondrocyte treated with TBHP alone or FASN knockdown after TBHP treatment. **f**, Detection of a product of lipid peroxidation, 4-HNE. **g**, Quantification of ROS intensity. **h**, KEGG enrichment after FASN knockdown. **i**, A

heat map of cellular senescence associated genes. **j**, Representative immunofluorescence images of gamma-H2AX nuclear foci, p21, and p16. **k**, Cellular senescence associated genes expression in chondrocyte with FASN knockdown after TBHP treatment indicated by RT-qPCR (n=3, 3 donors for three replicates). **l**, P16 and p21 expression in in chondrocyte with FASN knockdown after TBHP treatment. **m**, SA- β -Gal staining in chondrocyte with FASN knockdown after TBHP treatment. One-way analysis of variance (ANOVA) followed by a post hoc test is used for statistical analysis. Quantitative data shown as mean \pm s.d. Exact p values are shown in figures.

Meanwhile, there is a very high chance that investigators from the senescence field would not be fully convinced by their claims about CircRREB1- or FASN-mediated senescence. For instance, they showed that overexpression of either CircRREB1 or FASN is sufficient to cause senescent phenotypes in chondrocytes within “24 h” (as described in figures legends). However, generally speaking, mammalian cells rarely make a commitment to enter senescence within that short time window even after they are exposed to strong DNA-damaging agents. For human cells, it is not even unusual to take more than a week to develop senescence-associated secretory phenotypes (SASP). Therefore, what they observed in 24 h after CircRREB1 or FASN overexpression is unlikely to be a “senescence” response.

Response: We are very sorry for this statement. There must be a mistake because we worked on the manuscript for a long time and the repeated addition and removal of sentences and sections obviously led to this false statement. We are so appreciated your suggestions and we have performed and corrected SA- β -gal staining again according to your advice. Chondrocytes received different treatments and were stained by SA- β -gal one week after treatment. Representative images showed in the revised manuscript. Please see Fig. 2f, Extended Data Fig. 3b, Extended Data Fig. 4b and h, Extended Data Fig. 9h, l, and p, and Extended Data Fig. 11m.

Another critical reason why this reviewer does not think what they observed is senescence-associated phenomena is that unlike the general notion that mTOR inhibition suppresses SASP factor expression in senescent cells (Judith Campisi and colleagues, Nature Cell Biology, 2015 volume 17), they report that the combined treatments of PI3K and AKT inhibitors increased SASP factor expression.

Response: Thank you for your concerns for our manuscript, we are pleasure to do more experiments according to your helpful suggestion and explain the result of this part. We

would like to explain it through 4 points. We have discussed in the discussion part (please see line 843-867).

First of all, we performed mRNA-seq (Si-CircRREB1 vs Si-NC) and found that CircRREB1 knockdown was highly associated with PI3K-AKT signaling transduction by regulating FGFR3 and FGF18 expression (Extended Data Fig. 14c). Among differentially expressed genes related to PI3K-AKT pathway, we ranked the genes by Log2FC(Si1573/NC), which was showed as follow:

Gene_id	Gene name	Log2FC(Si1573/NC)	Significant	Regulate
ENSG00000156427	FGF18	5.426075799	yes	up
ENSG00000124253	PCK1	2.660034001	yes	up
ENSG00000068078	FGFR3	2.26032005	yes	up
ENSG00000092758	COL9A3	2.170689355	yes	up
ENSG00000144668	ITGA9	2.00779687	yes	up
ENSG00000091879	ANGPT2	1.943152397	yes	up
ENSG00000143127	ITGA10	1.59357179	yes	up
ENSG00000163235	TGFA	1.57400881	yes	up
ENSG00000113296	THBS4	1.480506754	yes	up
ENSG00000168477	TNXB	1.203756117	yes	up
ENSG00000259207	ITGB3	-1.001691826	yes	down
ENSG00000162409	PRKAA2	-1.007385624	yes	down
ENSG00000113494	PRLR	-1.081970797	yes	down
ENSG00000184371	CSF1	-1.093148067	yes	down
ENSG00000108821	COL1A1	-1.181189271	yes	down
ENSG00000029559	IBSP	-3.120891266	yes	down

FGF18 and FGFR3 are listed at top3. Hence, we consider that FGF18 and FGFR3 are the downstream of CircRREB1. FGF18 and FGFR3 are protective and indispensable for chondrocyte growth and viability, and FGF18 is the only FGF-based drug currently in clinical trials for anti-osteoarthritis (PMID: 32807927). FGFR3 deficiency also enhanced cartilage damage (PMID: 31662319). FGF18 promoted PI3K-AKT signaling pathway to anti osteoarthritis (PMID: 30273654). In our manuscript, after CircRREB1 knockdown, FGF18 and FGFR3 were upregulated (Extended Data Fig. 14h), as well as downstream

PI3K-AKT molecule (Extended Data Fig. 15c). However, we also found that FGFR3 and its related PI3K-AKT pathway are activated in many tumor models (PMID: 21078999, 24519156, 29299828,16767162). Hence, we would like to explain that FGFR3 and FGF18 and PI3K-AKT transduction play different roles in chondrocyte model and tumor model. Hence, we further acquired another three samples from OA patients, and CircRREB1 and FASN were knockdown respectively to explore the expression of PI3K, p-PI3K, AKT, p-AKT, mTOR, and p-mTOR expression. CircRREB1 and FASN knockdown in OA chondrocyte could upregulate the expression of p-PI3K and p-AKT, however, no obvious difference was observed of mTOR and p-mTOR. The result showed that CircRREB1 upregulated PI3K-AKT signal transduction, but showed little influence on mTOR activation. The result was showed as follows.

Secondly, we are very appreciated that you provided a meaningful study to us in your comments. We are agreed to your comment that PI3K/AKT/mTOR is a classic signal pathway. We have studied this reference you referred that mTOR promoted SASP in senescent tumor cells. We notice that PI3K-AKT pathway exerts big functions, which is an

intracellular signal transduction pathway that responds to extracellular signals and promotes metabolism, proliferation, cell survival, growth and angiogenesis. In our manuscript, we found that CircRREB1 or FASN knockdown activated PI3K and AKT, but showing little influence on mTOR pathway. Hence, we would like to explain that there may exist other pathway after AKT activation. We have noticed that AKT activation could mediate different pathway.

Except the classic PI3K/AKT/mTOR, p-AKT inhibits p27 and p21 to regulate cell survival, and promotes MDM2 to regulate p53 pathway. To date, p21, p53, as well as p16 are the classic senescent markers. Hence, we applied combination of PI3Ki and AKTi, which significantly reduced AKT activation (Extended Data Fig. 15a). AKT inhibition also increased CDK inhibitors p16 and p21 in OA chondrocytes (Extended Data Fig. 15b). We speculated that CircRREB1 regulated AKT activation to regulate p21 pathway. CircRREB1 knockdown, followed by AKT inhibition, Col2, MMP13, p16, and p21 expression were partially reversed, which suggested that CircRREB1 regulate p21 through PI3K-AKT signal pathway (Extended Data Fig. 15e).

Extended Data Fig. 15 PI3K inhibitor and AKT inhibitor reverses the effects of CircRREB1 knockdown. **a**, P-PI3K, PI3K, p-AKT, and AKT expression in HCs treated with PI3K inhibitor (PI3Ki), AKT inhibitor (AKTi) or combined treatment. **b**, p16 and p21 in HCs treated with combination of PI3K inhibitor (PI3Ki) and AKT inhibitor (AKTi). **c**, **d**, P-PI3K, PI3K, p-AKT, AKT, mTOR, and p-mTOR expression in HCs after CircRREB1 and FASN knockdown. **e**, PI3K added with AKTi reverse the protective effects mediated by CircRREB1 knockdown. **f**, Representative immunofluorescence images of Col2, Aggrecan, MMP13, and p16 in three groups.

Thirdly, in our manuscript, we found that CircRREB1 and FASN knockdown enhanced p-PI3K and p-AKT expression, but not p-mTOR expression. Because mTOR pathway is highly associated with autophagy. Herein, we have done some more experiments to explore whether CircRREB1 regulate autophagy and lysosome activity in chondrocyte. We showed the Extended Data Fig.3 as follows. We collected and cultured human chondrocytes from OA individuals. Then, CircRREB1 was knockdown. Western blot analysis showed that CircRREB1 knockdown did not affect autophagy associated proteins such as p62 and LC3 (Extended Data Fig. 3c, d, and e). Although CircRREB1 shows no influence on autophagy pathway, we also explore the role of CircRREB1 on lysosomal

activity by using chloroquine (Cq). After NC or CircRREB1 SiRNA transfection, chondrocytes were treated with or without Cq (5uM) for another 2 days. Western blot showed the LC3 and p62 expression with or without Cq stimulation (Extended Data Fig. 3f). Autophagy flux index was used to evaluate the impact of CircRREB1 on lysosomal activity. Autophagy flux= (LC3B-II + CQ/ β -actin)/(LC3B-II-CQ/ β -actin). The result of Autophagy flux showed no difference between Si-NC and Si-CircRREB1 group (Extended Data Fig. 3g). CircRREB1 knockdown also reduced p21 and gamma-H2AX nuclear foci expression. Above all, we confirmed that CircRREB1 regulate chondrocyte senescence indicated by p21 and other senescence markers instead of regulating autophagy and lysosome activity. This result is consisted with the result that CircRREB1 did not affect mTOR pathway.

Extended Data Fig. 3 **a**, Immunofluorescence images of p21 and gamma-H2AX nuclear foci expression after CircRREB1 knockdown. **b**, Acid β -Galactosidase expression after

CircRREB1 knockdown. **c**, LC3B and p62 proteins expression after CircRREB1 knockdown. **d**, Quantification of LC3 II. **e**, Quantification of p62. **f**, Chondrocytes transfected with NC or CircRREB1 SiRNA treated with or with CQ. LC3B and p62 expression are examined. **g**, Quantification of autophagy flux. Autophagy flux= (LC3B-II + CQ/13-actin)/(LC3B-II-CQ/13-actin). Two-tailed Student's *t* test used for statistical analysis. Quantitative data shown as mean ± s.d. Exact p values are shown in figures.

At last, we found that CircRREB1 knockdown decreased p21 expression and overexpression enhanced p21 expression (Fig. 2a and h). We applied combination of PI3Ki and AKTi in OA chondrocytes and found that p21, p16 was upregulated. Then we performed rescue assay, combination treatment in chondrocyte with CircRREB1 knockdown and found that p21, Col3, MMP13, and p16 were rescued, suggesting CircRREB1 regulate p21 pathway through PI3K-AKT transduction. Although PI3K-AKT-mTOR pathway caused SASP in many senescent tumor cells, we would like to discuss it in the discussion part. CircRREB1 knockdown in chondrocyte showed little effect on autophagy, as well as little effect on mTOR activation. However, CircRREB1 knockdown actually activated AKT activation in OA chondrocytes and regulated senescence associated p21, hence, we discussed that the effect of CircRREB1 on regulating p21 pathway larger than mTOR pathway(please see line 843-867).

Specific comments

1. More description on metabolomics data (in Fig. 4a-d) such as alterations in each lipid type. Metabolite measurement is required to support the significance of FASN in aging and OA progression. Do the lipid levels alter in aged human or mice and CircRREB1 overexpression system? What are the implications of these lipids in OA?

Response: Thanks for your constructive suggestions. The aim of metabolomics measurements is to find out which metabolic pathway is the main metabolic pathway in chondrocyte senescence progression. The result of overall metabolomics showed that lipid metabolism is the most relevant metabolic pathway in P0 group and P2 group. Hence, we suggested that lipid metabolism involved in aging progression. Numerous clinical studies have revealed an association between lipid accumulation and age-related OA pathogenesis, here, we showed the alterations of upregulated lipid type including fatty acid (FA), glyceride (GL), glycerophospholipids (GP), and glycosphingolipid (SP) between P0 generation chondrocyte group and P2 generation chondrocyte group (Extended Data Fig.

7a). Next, in order to investigate whether lipid metabolism also increased in aged mice, CircRREB1 overexpressed human chondrocytes or CircRreb1 gKO mice, we performed lipid metabolomics. In 18-month-old mice cartilage tissues, most types of lipid including FA, GL (DAG), GP (LPC, LPE, PC, PE), and SP (Cer, HexCer) were increased compared to 3-month-old mice, suggesting that these lipids involved in aging mice (Extended Data Fig. 7b, c, and d). Human chondrocytes treated with Ad-CircRREB1 and Ad-vector and metabolic analysis showed that most types of lipid were also increased after CircRREB1 overexpression (Extended Data Fig. 7e, f, and g). Furthermore, cartilage samples from WT mice and CircRreb1 gKO mice were collected and lipid metabolomics showed that some lipids (FA, LPC, LPE, PC, and PE) were decreased after CircRreb1 knockout (Extended Data Fig. 7h, i, and j). Above all, we found that CircRREB1 mediated lipid metabolism in chondrocytes with aging (please see line 352-364).

As age is the most risk factor for OA. Next, we would like to explain the application of these lipid in OA. The current existing evidence suggests that OA is associated with obesity-related chronic inflammation as well as abnormal lipid metabolism in obesity, such as fatty acids (FA) and triglycerides (GL) (PMID: 35510046). Rosendaal et al. showed that FA level was increased in OA chondrocytes (PMID: 31629023), causing matrix degrading enzymes upregulation (PMID: 29852453, 31849953, 28418007). 15-hydroxyeicosatetraenoic acid (15-HETE), one component of FA, was increased in plasma in OA patients (PMID: 26195278). Hence, FA is the factor causing OA. Epidemiological studies indicated that GL is a risk factor for the progression of OA (PMID: 32606898). GL can promote NF- κ B nuclear translocation by inducing endoplasmic reticulum stress, which leads to the release of pro-inflammatory factors, thereby enhancing the catabolism of the ECM (PMID: 29136765, 29434185). GP is also associated with OA. For some examples, an increased plasma ratio of LPC to phosphatidylcholine (PC) was associated with advanced knee OA (PMID: 27160277). Upregulated PC was observed in OA tissues (PMID: 33582239, 31273319). One study showed that ceramide and sphingomyelins were increased in OA patients (PMID: 35644463). Oleic acid/palmitic (O/P) acid caused ceramide accumulation, thereby causing OA progression (PMID: 33778044). Overall, we found that lipid metabolism is higher expressed in senescent chondrocyte, which can be a biomarker for OA. We demonstrated that CircRREB1 mediated OA progression by regulating lipid metabolism.

Extended Data Fig. 7 **a**, A heat map showed the upregulated lipid types in P0 generation chondrocytes and P2 generation chondrocytes. **b**, Quality control of cartilage tissues from 3-month-old mice and 18-month-old mice (n=4). **c**, Proportion of various lipids in 3-month-old mice and 18-month-old mice. **d**, Differentially expressed lipid types in 3-month-old mice and 18-month-old mice. **e**, Quality control of CircRREB1 overexpressed and vector

overexpressed chondrocytes (n=4). **f**, Proportion of various lipids in CircRREB1 overexpressed chondrocyte and vector overexpressed chondrocyte. **g**, Differentially expressed lipid types in CircRREB1 overexpressed chondrocyte and vector overexpressed chondrocyte. **h**, Quality control of cartilage tissues from WT mice and CircRreb1 gKO mice (n=4). **i**, Proportion of various lipids in WT mice and CircRreb1 gKO mice. **j**, Differentially expressed lipid types in WT mice and CircRreb1 gKO mice. **k**, Quantifications of the relative intensity of FASN, ELOVL5, RLOVL6, and SCD1 in articular chondrocyte (AC) and CCZ (calcified cartilage zone) between younger group and older group. Two-tailed Student's *t* test is used for statistical analysis. Quantitative data shown as mean \pm s.d. Exact p values are shown in figures.

2. How is circRREB1 upregulated in aging chondrocyte in the first place? The upstream regulatory mechanism for circRREB1 production in senescent chondrocytes remains unclear. In Fig. 1f and Extended Fig. 1, the expression of circRREB1 is upregulated in senescent or aged chondrocytes. How are the expressions of the host gene RREB1 in these conditions?

Response: Thanks for your suggestion to improve the quality of our manuscript. We have performed some experiments to explain the upstream regulatory mechanism for CircRREB1 production. First of all, we performed Rt-qPCR to detect host gene RREB1 expression in two senescence models of chondrocyte. One is P2 generation chondrocyte compared with P0 generation, the other is Doxo induced senescence model. The result of RT-qPCR showed that host gene RREB1 were not changed in two models. As RNA-binding proteins could govern CircRNAs post-transcriptionally, we suggested that CircRREB1, but not RREB1 mRNA, was governed by specific RNA-binding proteins post-transcriptionally during chondrocyte senescence. To validate our hypothesis, we measured CircRREB1 and host gene RREB1 expression after knockdown of an RNA-binding protein DExH-Box Helicase 9 (DHX9), which was reported governing the CircRNAs biogenesis broadly. After DHX9 knockdown, CircRREB1 was upregulated, while host gene RREB1 did not show significant changes. Notably, we also found that DHX9 was downregulated in P2 generation chondrocyte. DHX9 knockdown in chondrocyte also increased p16 and p21 expression indicated by RT-qPCR. Together, the downregulation of DHX9 in aging

chondrocyte is at least partially responsible for the overexpression of CircRREB1 (please see line 231-243).

Extended Data Fig. 2 CircRREB1 is regulated by DHX9. **a**, *RREB1* mRNA expression in P0 generation chondrocyte and P2 generation chondrocyte (n=3, 3 donors for three replicates). **b**, *RREB1* expression in Doxo stimulated model (n=3, 3 donors for three replicates). **c**, *CircRREB1* and *RREB1* expression after DHX9 knockdown in chondrocyte (n=3, 3 donors for three replicates). **d**, *DHX9* expression in P0 generation chondrocyte and P2 generation chondrocyte (n=3, 3 donors for three replicates). **e**, Senescence associated *p16* and *p21* expression after DHX9 knockdown (n=3, 3 donors for three replicates). Two-tailed Student's *t* test used for a, b, d, e. One-way analysis of variance (ANOVA) followed by a post hoc test is used for c, e. Quantitative data shown as mean \pm s.d. Exact p values are shown in figures.

3. Introduction includes redundant description (for instance, line 108-111 and line 111-115). In line 106-108, is there any previously identified association between FASN and aging-associated diseases other than cancer?

Response: Thanks for your suggestions. We have simplified this description in the introduction part. A previous study has reported that FASN was important for the initial stage of the induction of senescence in mouse hepatic stellate cells and human primary fibroblasts (PMID:30962418). This study indicates FASN is vital during aging. However,

the underlying relationship between FASN and age-related OA is not well studied. Hence, we consider that it is important to study FASN viability during aging in chondrocytes.

4. In line 161 and Supplementary Table 1, the result of RNA-seq summarized in the table described only 15 up- and down-regulated circRNAs, not 20. For the table, a statistical measurement such as FDR to test the significance of differential expression is missing. What does mean value indicate in the table? Read count?

Response: Thanks for your careful review to our manuscript. We are sorry for the wrong description. We have corrected it (please see line 177-178). For CircRNA data, we have added FDR value which indicating significance in the new edited Supplementary Table 1. Mean value of P0 and P2 were averaged by fpkm value.

5. Description of histological assessment method of human and mouse OA is missing, especially OARSI grading and synovitis scoring. How was the thickness of articular cartilage measured?

Response: We are sorry for the missing part of description of histological assessment method in this manuscript. Here we would like to explain how we perform histological assessment. After safranin O/Fast green staining and hematoxylin and eosin (H&E) staining. All sections were evaluated by Osteoarthritis Research Society International (OARSI) system and synovitis score to evaluate the OA degree by two independent individuals. OARSI grading and synovitis score were listed in Supplementary Table S9 and Supplementary Table S10.

For the thickness of articular cartilage measurement, maximal cartilage thickness measured from at least 5 sections (80 um apart) of each joint by two intendent individuals. Then, measurements of each section of each joint were averaged to give an average joint thickness measure.

6. More detailed description of the origin of human cartilage samples is required since it lacks consistency throughout the manuscript. While online methods read “participants (50– 60 years old) were included into the younger group and participants (70–89 years old) were included into the older group”, other texts in Result and figures read “younger (50–65y) and older adults (70–85y)”.

Response: We are sorry for the missing part of the description of human cartilage samples and spelling mistake. Herein, we have corrected and added this part in the supplementary table S5.

7. Related to Fig. 4e, human cartilage shows distinct zonal structure and chondrocytes in

each zone exhibits different characteristics. The authors should have discussed the zonal origin of cartilage tissue and section method for the cartilage samples, and compared the protein level of fatty acid metabolism-related proteins within similar regions.

Response: Thanks for your suggestions for our manuscript. In human normal cartilage, the superficial zone was defined as the first 10% of the tissue thickness, the translational zone as the next 10% and the radial zone as the remaining 80% of the tissue thickness (PMID:12127837). Hence, according to the distinct zonal structure of cartilage, we would like to explore the superficial zone (articular cartilage, AC) and the radial zone (calcified cartilage zone, CCZ) of tibia plateau according to this reference (please see line 365-372).

Section method for the cartilage samples: Tibia plateau tissues were collected and fixed in 4% formaldehyde for 3 days and decalcified in 10% EDTA, pH=7.5 for one month. Tissues were dehydrated with a graded series of ethanol washed and embedded in paraffin and sectioned to 5-um slices.

According to advice, we performed IHC staining of FASN, ELOVL5, ELOVL6, and SCD1 again in the superficial zone (articular cartilage) and the radial zone (calcified cartilage zone) of tibia plateau respectively (Fig. 4e). We also compared the relative expression of FASN, ELOVL5, ELOVL6, and SCD1 of two distinct zones respectively (Extended Data Fig. 7k).

8. Related to Fig. 1c,d, the result of RNA-seq should be addressed in detail. How was total 3800 circRNAs were identified using CIRI2 and CIRC explorer? In the previous article published by the same author (Shen et al., Ann. Rheum. Dis. 2019), the author identified 12,738 circRNAs using three OA and control human cartilage samples and the same circRNA explorers. Where did the discrepancy come from? How was the expression of circRREB1 in the previous RNA-seq data?

Response: Thanks for your comments for our manuscript. We used CIRI2 and CIRCexplorer2 software to identify and analyze the circular RNA, and selected the circular RNA predicted by the two software as the final prediction result. The simultaneous prediction of these two software can significantly reduce the false positive results predicted by the software and more accurately identify the circular RNA. CIRI2 software uses the result file of the genome compared by bwa mem software as the input file. For each sample, CIRI2 software predicts the position information before and after the formation of circular

RNA, and further identifies the candidate circRNA. The specific analysis process is as follows: after the bwa mem comparison, CIRI2 processes the sam file twice. For the first time, the junction reads are detected through the PCC (paired chiastic clipping) signal. The candidate circRNAs were obtained by preliminary filtration using PEM (paired-end mapping) and GT-AG sequence characteristics. The second time, detect additional junction reads again and further filter out false positive candidate circRNAs. The schematic diagram of CIRI2 software is as follows:

CIRC explorer 2 analyzes the reverse splicing information from the Tophat-Fusion comparison results, and annotates the prediction results of circular RNA according to the comparison results. The specific analysis process is as follow: first of all, we filter out the reads that cannot be mapped by Tophat-Fusion, and then use Tophat-Fusion mapping to map these reads to the genome. Secondly, the nonlinear candidate reads in the Tophat-Fusion GT-AG comparison results are re-compared to the genome, and the position of the junction site is more accurately determined with the help of the gene annotation file. The schematic diagram of CIRC explorer 2 software is as follows:

Hence, the above are the method we identify the CircRNAs.

Next, we would like to explain that the CircRNAs sequencing we published in Ann. Rheum. Dis (2019). We used 3 paired clinical OA and control tissues to perform CircRNA deep sequencing. There were no obvious difference of Age, height, weight, and BMI between 3 paired clinical OA and control tissues. Actually, the difference between two groups is whether is OA. The information of 3 paired clinical OA and control tissues showed as follow:

3 paired clinical OA tissues for circRNA deep sequencing				
Gender	Age(year)	Height(cm)	Weight(kg)	BMI
Male	61	170	72	24.9135
Male	61	168	58	20.5499
Female	60	160	65	25.3906

3 paired control tissues for circRNA deep sequencing				
Gender	Age(year)	Height(cm)	Weight(kg)	BMI
Male	60	169	78	27.3099
Male	62	170	62	21.4532
Female	62	160	60	23.4375

At last, we found out the CircRREB1 expression in this CircRNA deep sequencing. The CircRREB1 expression value in control tissue is 30.1, 29.1, and 21.3 (fpkm value). The expression in OA tissue is 37.9, 26.6, and 8.3 (fpkm value). Due to the deviation of sample size, there is no significant difference between control and OA tissues, but we find that CircRREB1 is upregulated in OA tissue and we need investigate the expression of CircRREB1 in larger human patient material. Furthermore, in this manuscript, we used three P0 generation chondrocytes and three P2 generation chondrocytes to perform CircRNA deep sequencing. Hence, the samples used for CircRNA sequencing is differ from sequencing used in our previous work. We have discussed in the discussion part (please see line 727-738).

9. Related to the screening of circRREB1 (hsa_circ_0001573) in Fig. 1d, the rationale for choosing hyfor further study among differentially expressed circRNAs is lacking the justification. For instance, in line 171-173, why was the top downregulated hsa_circ_0000211 excluded for further study?

Response: Thanks for your comments for our manuscript. We performed CircRNA deep sequence from three P0 generation chondrocytes and three P2 generation chondrocytes.

We selected the top 10 upregulated CircRNAs according to $\log_2FC(P2/P0) > 1$ and mean P2 value > 40 (hsa_circ_0001181, hsa_circ_0002153, hsa_circ_0000711, hsa_circ_0001756, hsa_circ_0000284, hsa_circ_0003611, hsa_circ_0001461, hsa_circ_0001573, hsa_circ_0002132, and hsa_circ_0006935) and top 10 downregulated CircRNAs according to $|\log_2FC(P2/P0)| > 1$ and mean P2 value < 2 (hsa_circ_0000211, hsa_circ_0007068, hsa_circ_0046263, hsa_circ_0001781, hsa_circ_0004792, hsa_circ_0079672, hsa_circ_0038005, hsa_circ_0042079, hsa_circ_0005230, hsa_circ_0006208). After evaluation in Fig. 1f, we further selected top 5 upregulated CircRNAs (hsa_circ_0003611, hsa_circ_0006935, hsa_circ_0001181, hsa_circ_0001573, hsa_circ_0000711) and top5 downregulated CircRNAs (hsa_circ_0007068, hsa_circ_0004792, hsa_circ_0079672, hsa_circ_0042079, hsa_circ_0000211) to perform

RNA knockdown to detect the expression of senescent marker p21 and SASPs factor MMP13 via RT-qPCR analysis. For upregulated CircRNAs, we found that hsa_Circ_0001573 significantly decreased the expression of p21 and MMP13. For downregulated CircRNAs, the effects of knockdown were no more obvious than that of the upregulated CircRNAs. Hsa_Circ_0001573 showed the best effects among these CircRNAs, hence, we further selected hsa_Circ_0001573 as our target in this study (please see line 189-196). The result of RT-qPCR showed as follows, also in the Extended Data Fig. 1d and e.

10. Related to Fig. 4l and m, do CircRreb1 gKO mice show reduced synthesis of specific lipids in system level?

Response: Cartilage samples from WT mice and CircRreb1 gKO mice were collected and lipid metabolomics showed that some lipids (FA, LPC, LPE, PC, and PE) were decreased after CircRreb1 knockout (Extended Data Fig. 7h, I, and j). It suggests that CircRreb1 mediated lipid metabolism in OA and senescence progression.

11. For Fig. 5d and Extended Fig. 4f,g, binding site prediction of circRREB1 with FASN is not clearly explained. How does the CatRAPID prediction result match with that from RIP assay?

Response: Thank for your helpful comment and we are very pleasure to explain the result of RIP assay and CatRAPID prediction. We used CatRAPID software to predict the nucleotide position of CircRREB1 which interacted with FASN protein. The result showed in Extended Fig. 8g indicates that the position (300nn-400nn) of CircRREB1 has great potential to combine with FASN. RNA loop structure is important for RNA interaction. So, we used RNAfold software and visualized CircRREB1 loop structure (Extended Fig. 8f). We divided CircRREB1 sequence into 5 fragments and RIP result showed that fragment 2 of CircRREB1 combined with FASN. The sequence of Fragment2 exactly corresponds to the 300nn-400nn binding site in the prediction result. Furthermore, EMSA assay confirmed

the direct interaction between CircRREB1 and FASN, the sequence of Biotin-CircRREB1 probe2 is according to Fragment2. Above all, we confirmed that RIP assay is matched with CatRAPID prediction.

12. Some of the statistical methods they used are inappropriate or inaccurately described. Especially, statistical analysis of nonparametric data such as histological grades should not have been conducted using 'two-side unpaired t-test' throughout the manuscript. The authors described that they used one-way ANOVA with LSD-t by assuming equal variance for multiple group comparisons. However, it says one-way ANOVA with Turkey's multiple comparison, which should be at least 'Tukey's HSD' although LSD and HSD use different significant difference. Overall, this reviewer cannot figure out the statistical method they used.

Response: Thanks for your suggestions to our manuscript. We have corrected the statistical method according to your advice.

Statistical analysis was performed using GraphPad Prism software (version 8.0). Data from two groups were analyzed by using unpaired, two-tailed Student's *t* test for RT-qPCR and the percentages of positive cells. One-way analysis of variance (ANOVA) followed by Tukey's HSD test is used for multigroup comparisons. For nonparametric data, the Mann-Whitney U test was used for OARSI grade, Synovitis score, and osteophytes number. Parametric data were presented as mean \pm SD and the calculated 95% confidence intervals (CIs) for nonparametric data. Statistical significance was set at $P < 0.05$ (please see line 1130-1138).

We also described the method in the figure legends. Thanks again for your helpful comment.

13. Related to Fig. 6a, the rational for selecting MDM2 as a candidate is insufficiently provided.

Response: We are very sorry for the missing part of MDM2 selection. Herein, we appreciated this chance and are very pleasure to explain why we select MDM2 as the candidate. First of all, we utilized the UbiBrowser (<http://ubibrowser.ncpsb.org.cn>) to predict E3 ligase of FASN (Fig. 6i). STUB1 and MDM2 are the top2 E3 ligase of FASN (Fig. 6i). To further explore the effects of STUB1 and MDM2 on FASN protein level. We overexpressed STUB1 and MDM2 in chondrocytes respectively and western blot analysis indicated that MDM2 significantly reduced FASN protein level compared to STUB1 overexpression (Fig. 6j). Next, we performed Co-IP assay to confirm the interaction

between FASN and MDM2 (Fig. 6k). FASN and MDM2 co-localized in cytoplasm (Fig. 6l). Hence, above all, we consider that MDM2 as a potential E3 ligase of FASN.

14. Related to mass spectrometry analysis of FASN in line 470-473, the method is completely missing and the explanation for MS data is not enough. Was the UPLC-MS/MS method performed focusing on detecting acetylation sites of FASN? Were K673 and K1065 the only acetylation sites on FASN?

Response: We are appreciated your careful review and we are sorry for the missing part of MS method. Herein, we have described the MS method in the revised manuscript. We used UPLC-MS/MS method to detect acetylation sites of FASN. The method of UPLC-MS/MS is as followed:

Chondrocytes overexpressed FASN were collected and cell protein solutions were separated by sodium dodecyl sulfate-polyacrylamide gel electrophoresis, followed by coomassie blue staining. At first, for in-gel tripic digestion, gel pieces were destained in 50 mM NH₄HCO₃ in 50% acetonitrile (v/v) until clear. Gel pieces were dehydrated with 100 μl of 100% acetonitrile for 5 min, the liquid removed, and the gel pieces rehydrated in 10 mM dithiothreitol and incubated at 56 °C for 60 min. Gel pieces were again dehydrated in 100% acetonitrile, liquid was removed and gel pieces were rehydrated with 55 mM iodoacetamide. Samples were incubated at room temperature, in the dark for 45 min. Gel pieces were washed with 50 mM NH₄HCO₃ and dehydrated with 100% acetonitrile. Gel

pieces were rehydrated with 10 ng/μl trypsin resuspended in 50 mM NH₄HCO₃ on ice for 1 h. Excess liquid was removed and gel pieces were digested with trypsin at 37 °C overnight. Peptides were extracted with 50% acetonitrile/5% formic acid, followed by 100% acetonitrile. Peptides were dried to completion and resuspended in 2% acetonitrile/0.1% formic acid.

Then, UPLC-MS/MS method was performed to detect acetylation sites of FASN. Specifically, the tryptic peptides were dissolved in 0.1% formic acid (solvent A), directly loaded onto a home-made reversed-phase analytical column (15-cm length, 75 μm i.d.). The gradient was comprised of an increase from 6% to 23% solvent B (0.1% formic acid in 98% acetonitrile) over 16 min, 23% to 35% in 8 min and climbing to 80% in 3 min then holding at 80% for the last 3 min, all at a constant flow rate of 400 nl/min on an EASY-nLC 1000 UPLC system. The peptides were subjected to NSI source followed by tandem mass spectrometry (MS/MS) in Q Exactive™ Plus (Thermo) coupled online to the UPLC. The electrospray voltage applied was 2.0 kV. The m/z scan range was 350 to 1800 for full scan, and intact peptides were detected in the Orbitrap at a resolution of 70,000. Peptides were then selected for MS/MS using NCE setting as 28 and the fragments were detected in the Orbitrap at a resolution of 17,500. A data-dependent procedure that alternated between one MS scan followed by 20 MS/MS scans with 15.0s dynamic exclusion. Automatic gain control (AGC) was set at 5E4.

The resulting MS/MS data were processed by using Proteome Discoverer 2.4. The modification setting is acetylation of lysine. Carbamidomethyl on Cys were specified as fixed modification and oxidation on Met was specified as variable modification. Peptide confidence was set at high, and peptide ion score was set > 20.

Next, we would like to explain the result of UPLC-MS/MS result. We matched two acetylation sites of FASN in this assay. One is K673 site and the other is K1065. We should explain that two acetylation sites on FASN were detected in our chondrocytes model.

K673: EGVFAKEVR, 1xAcetyl [K6]

K1065: QKLYTLQDKAQVADVVSRR, 1xAcetyl [K2]

We also mutated K673 and K1065 site on FASN and found that K673 was the primary acetylation site of FASN. The acetylation levels of FASN mediated by CircRREB1 knockdown or overexpression were also affected by K673R, suggesting CircRREB1 affecting K673 acetylation of FASN. This result is accordance with the result showed by molecular docking. Please see the docking result as follows. The part (b) showed that K673 site which is combined with specific deacetylase HDAC3.

15. In Fig. 6k, the author could not find the result of Flag-FASN MS.

Response: We are very sorry for the missing part of Flag-FASN MS. We have loaded the MS result in the Supplementary Table S3.

16. Related to Extended Fig. 9-10, there are many attempts to reduce OA and SASP expression by inhibiting PI3K-Akt pathway (Wang et al., *Am. J. Transl. Res.* 2022) and the inhibition of this pathway is also known to promote autophagy (Xue et al., *Biomed. Pharmacother.* 2017). PI3K-Akt is generally known to activate mTOR and NF- κ B pathway, which are considered as a pro-SASP transcription factors (Herranz et al., *Nat. Cell. Bio.* 2015; Salminen et al., *Cell Signal.* 2012).

Response: Thanks for your comments for our manuscript. We are appreciated this chance to explain concerns to improve the quality of our manuscript. We have read the article you

referred and published in *American journal of translational research* by Wang et al. They focused on Senomorphic agent pterostilbene to ameliorate osteoarthritis through PI3K/AKT/NF- κ B pathway. In figure 7A, they used IL-1 β (10ng/ml) to stimulate chondrocyte and found that p-PI3K and p-AKT were increased, which showed the opposite result in another research by Guo *et al.* published in *Pharmacological research* (PMID:30273654, please see Figure 2A in this study, p-PI3K and p-AKT were decreased after IL-1 β treatment). Hence, we further acquired another three samples from OA patients and CircRREB1 and FASN were knockdown respectively to explore the expression of PI3K, p-PI3K, AKT, p-AKT, mTOR, and p-mTOR expression. Our result showed that CircRREB1 and FASN knockdown in OA chondrocyte could upregulate the expression of p-PI3K and p-AKT, however, no obvious difference was observed of mTOR and p-mTOR. The result showed that CircRREB1 upregulated PI3K-AKT signal transduction, but showed little influence on mTOR activation. The result was showed as follows.

PI3K-Akt is generally known to activate mTOR, which is associated with autophagy. Hence, we do more experiments to verify whether CircRREB1 regulate autophagy in aging chondrocyte. When CircRREB1 was knockdown in chondrocyte, we found that autophagy related proteins LC3B and P62 showed no obvious difference with or without CircRREB1 knockdown (Extended Data Fig. 3c, d, and e). Then, we investigated whether CircRREB1 affect autophagy flux in aging chondrocyte. Chondrocyte transfected with NC or CircRREB1 SiRNA, then treated with or without CQ. Autophagy flux index was used to evaluate the impact of CircRREB1 on lysosomal activity. Autophagy flux= (LC3B-II + CQ/ β -actin)/(LC3B-II-CQ/ β -actin). The result of Autophagy flux showed no difference between Si-NC and Si-CircRREB1 group (Extended Data Fig. 3f and g). These results showed that CircRREB1 did not affect autophagy in aging chondrocyte. Hence, we would like to explain that although CircRREB1 knockdown upregulated PI3K and AKT activation, downstream of p-AKT showed little effect on mTOR activation and autophagy.

We consider that PI3K-AKT activation regulate another pathway instead of mTOR and autophagy pathway. PI3K-AKT signal pathway is a classic pathway which are wildly used in different disease. Except promotes mTOR and NF-kB pathway, p-AKT also inhibits CDK inhibitor p21, promotes p-MDM2 to regulate p53 pathway. P21 and p53 are now considered as senescence markers. CircRREB1 knockdown decreased p21 expression, however, its expression was reversed by PI3K-AKT inhibition, suggesting that CircRREB1 regulate PI3K-AKT signaling pathway to regulate p21 pathway. We also have discussed this part in the discussion part (please see line843-867).

17. In Fig. 7h, the quality of IHC against FGFR3, FGF18, p-PI3K, and p-Akt are low (background levels are very different between samples), making it difficult to compare signals between cKO+DMM and cKO+DMM+CircRreb1 oe groups.

Response: Thanks for your suggestions for our manuscript. We are very pleasure to perform IHC staining again according to your advice. We have showed the representative images in the Figure 7.

REVIEWER COMMENTS

Reviewer #2 (Remarks to the Author):

The authors have satisfactorily addressed the issues I commented on earlier.

Reviewer #3 (Remarks to the Author):

Thanks for the authors' response, but I am still confused by the sample source.

As described in the manu, "knee joints samples from three participants were collected and chondrocytes were isolated and cultured " and then "P0 generation chondrocytes were collected, at the same culture condition, the remain were cultured for two passages in monolayer (P2 generation)". Are these two generations of cell samples from cartilage sufficient to complete relevant experiments? As described in the manu, "Participants (50–65 years old) were included into the younger group (n=12), and participants (70–85 years old) were included into the older group (n=12)", which group should the patients (66-69 years old, for example, the one 66 years old as in Table S5) included in? Are the sample numbers correct? Additionally, why the 3 knee joint tissues for circRNA deep sequencing are from "younger patients (i.e. 50–65 years old)"? Again, how about the severity of OA?

Reviewer #4 (Remarks to the Author):

The authors have elaborately resolved all the issues raised by this reviewer by providing extensive experimental results. This reviewer has no further suggestions.